# Rethinking Associative Memory Mechanism in Induction Head

**Shuo Wang**
The University of Tokyo
wang-shuo3112@g.ecc.u-tokyo.ac.jp

**Issei Sato**
The University of Tokyo
sato@g.ecc.u-tokyo.ac.jp

## Abstract

Induction head mechanism is a part of the computational circuits for in-context learning (ICL) that enable large language models (LLMs) to adapt to new tasks without fine-tuning. Most existing work explains the training dynamics behind acquiring such a powerful mechanism. However, it is unclear how a transformer extract information from long contexts and then use it to coordinate with global knowledge acquired during pretraninig. This paper considers weight matrices as associative memory to investigate how an induction head functions over long contexts and balances in-context and global bigram knowledge in next token prediction. We theoretically analyze the representation of the learned associative memory in attention layers and the resulting logits when a transformer is given prompts generated by a bigram model. In the experiments, we design specific prompts to evaluate whether the outputs of the trained transformer align with the theoretical results.

## 1 Introduction

In recent years, transformer-based models, such as BERT (Devlin et al., 2019) and GPT (Radford, 2018; Radford et al., 2019; Brown et al., 2020), have achieved remarkable success in natural language processing. Especially, in-context learning (ICL) (Brown et al., 2020; Dong et al., 2024) has emerged as a groundbreaking capability within large language models (LLMs), enabling them to adapt to new tasks without traditional fine-tuning. Instead, these models leverage patterns from a prompt or input sequence, effectively learning "in context" by interpreting examples or instructions provided in real-time (Liu et al., 2022; Wu et al., 2023). During a phase change, models acquire the ability to complete complex patterns through induction heads, suggesting that induction heads are a key mechanistic basis of ICL (Olsson et al., 2022). It is known that the induction head mechanism, which is a pattern of attention in a transformer that enables the model to copy and reuse information from earlier in the context, can emerge in two-layer transformers (Elhage et al., 2021), and many theoretical studies have focused on analyzing two-layer architectures to understand how induction head develops (Edelman et al., 2024; Nichani et al., 2024; Chen et al., 2024b). Among them, Bietti et al. (2024) demonstrated, both theoretically and empirically, that training a transformer leads to weight matrices behaving as associative memories, composing the induction head mechanism.

However, the effectiveness of the induction head mechanism tends to diminish for the latter part of a sequence (Bietti et al., 2024), which does not align with the behavior observed in real-world LLM. In addition, while the learning dynamics of the induction head have been elucidated in the existing work, it remains unclear how **in-context knowledge** provided in the prompt and **global knowledge** acquired during pretraining influence the output of two-layer transformers. Understanding how a transformer predicts the next token is crucial, as it allows us to model phenomena such as context hijacking (Jiang et al., 2024), in which altering the context disrupts factual recall, causing the model to produce incorrect outputs influenced by misleading in-context information.

In this paper, we theoretically and empirically show, from the perspective of associative memory, how a two-layer transformer with relative positional encoding (RPE) can avoid

| Paper | Model | Objective | Theoretical analysis |
|-------|-------|-----------|----------------------|
| Meng et al. (2022) | GPT | knowledge editing | no |
| Cabannes et al. (2024a) | linear model | scaling laws | yes |
| Cabannes et al. (2024b) | linear model | learning dynamics | yes |
| Nichani et al. (2024) | two-layer transformer | causal structure | yes |
| Friedman et al. (2023) | modified transformer | transformer programs | no |
| Jiang et al. (2024) | one-layer transformer | concept association | yes |
| Chen et al. (2024a) | linear model | noisy classification | yes |
| Bietti et al. (2024) | two-layer transformer | learning dynamics | yes |
| OURS | two-layer transformer | length generalization & global v.s. in-context | yes |

Table 1: Comparisons among related work and ours in terms of associative memory. The concept of associative memory is widely adopted for theoretical analyses of various phenomena.

failure in detecting patterns that appear in the latter part of a sequence. We also investigate how the model prioritizes in-context knowledge versus global knowledge when predicting the next token, using training data generated by a bigram model with triggered transitions. To summarize, we make the following contributions:

- We theoretically show that the induction head learned by a transformer using RPE can avoid neglect of patterns within the framework of associative memory.
- We experimentally confirmed that the model with RPE can comprehensively use in-context knowledge for the bigram model. We also demonstrated its ability to effectively infer on sequences where the transformer using absolute positional encoding (APE) fails in next-token prediction.
- We investigated how the model prioritizes in-context knowledge versus global knowledge when generating outputs, using both theoretical analysis and experimental evaluation.

## 2 Related work

**Associative memory** Associative memory refers to the storage and retrieval of patterns, and its computational model was designed to retrieve complete memories from sufficient fragments of information in neuroscience (Amari, 1972; Hopfield, 1982; 1984). After the emergence of modern Hopfield networks (Krotov & Hopfield, 2016), which store patterns in an energy function and retrieve them using an updating rule. Moreover, many studies employ the concept of associative memory in various ways. For instance, Zhang et al. (2024a) propose a memory unit that estimates a conditional probability distribution with Gaussian kernel smoothing. Some work considers attention blocks as memory because they read and write information flowing through the residual stream (Nichani et al., 2024; Friedman et al., 2023). Meng et al. (2022) locate factual association patterns in the weight matrix of an MLP. Recently, it has become increasingly popular to regard a matrix as associative memory from a theoretical perspective, as shown in Tab. 1. Previous studies examine scaling laws in relation to model capacity (Cabannes et al., 2024a), learning dynamics of linear model in terms of particle system (Cabannes et al., 2024b), and the effect of low-rank approximation of weight matrix to classification involving a noisy token (Chen et al., 2024a) for a linear model. Additionally, Jiang et al. (2024) demonstrate that a one-layer transformer can recover latent concept from long enough context.

**In-context learning and induction head** Since Brown et al. (2020) introduced ICL, theoretical understanding of this phenomenon has been a major interest within the community. For example, Akyürek et al. (2023) showed that transformer can implement a learning

algorithm that solves linear regression. Von Oswald et al. (2023) demonstrated that linear self-attention layer can emulate gradient descent. Similarly, many studies (Ahn et al., 2023; Mahankali et al., 2024; Zhang et al., 2024b; Dai et al., 2023) examined the theoretical connections between ICL and gradient descent. Mechanistic interpretability, which aims to understand the model behavior by reverse-engineering its internal components (Nanda et al., 2023; Wang et al., 2023a; Conmy et al., 2023), is another perspective for analyzing ICL (Olsson et al., 2022; Elhage et al., 2021). In particular, Elhage et al. (2021) discovered that two-layer attention-only transformers understand the previous use of the token and look for similar situations in the context. In addition, Olsson et al. (2022) provided empirical evidence that induction heads are the mechanism behind ICL. Moreover, Bansal et al. (2023) identified that many induction heads are also among the attention heads deemed critical for ICL. The most relevant study to ours is by Bietti et al. (2024), who investigated the learning dynamics of the induction head mechanism in two-layer transformers from the perspective of associative memory. In contrast, our study focuses on the functionality of the induction head mechanism and the influence of knowledge acquired during training and that obtained in context on the final output.

## 3 Preliminaries

### 3.1 Transformer architecture

In this work, we focus on a two-layer transformer described below.

**Calculation** Given a sequence of tokens $z_{1:T}$ of length $T \in \mathbb{N}$ from the vocabulary set $\mathcal{V}$, each token $z_t$ is embedded into a $d$-dimensional vector and the calculation proceeds as follows:

$$x_t^{(0)} := w_E(z_t) \in \mathbb{R}^d,$$

$$x_t^{(1)} := \Phi_1 x_{1:t}^{(0,v)} \sigma((W_K^1 x_{1:t}^{(0,k)})^\top W_Q^1 x_t^{(0,q)}) + x_t^{(0,v)} \in \mathbb{R}^d, \tag{1}$$

$$x_t^{(2)} := W_O^2 W_V^2 x_{1:t}^{(1)} \sigma((W_K^2 x_{1:t}^{(1)})^\top W_Q^2 x_t^{(1)}) + x_t^{(1)} \in \mathbb{R}^d, \tag{2}$$

$$x_t := W_2(\text{ReLU}(W_1 x_t^{(2)})) + x_t^{(2)}, \tag{3}$$

$$\hat{z}_{t+1} = \arg\max_{v \in \mathcal{V}} \sigma(W_U x_t)_v, \tag{4}$$

where $W_E \in \mathbb{R}^{d \times |\mathcal{V}|}$ is an embedding matrix, $\sigma : \mathbb{R}^t \to \mathbb{R}^t$ is the softmax function, $x^{(0,k)}, x^{(0,q)}$, and $x^{(0,v)}$ are the token embeddings, which may or may not include positional information, $\Phi_1 \in \mathbb{R}^{d \times d}$ represents $W_O^1 W_V^1$, $W_U = (w_U(v_1), \ldots, w_U(v_{|\mathcal{V}|}))^\top \in \mathbb{R}^{|\mathcal{V}| \times d}$ is an unembedding matrix, and $v_i$ represents the $i$-th vocabulary in $\mathcal{V}$. The feed-forward layer consists of matrices $W_1 \in \mathbb{R}^{d \times V}$ and $W_2 \in \mathbb{R}^{V \times d}$.

Note that Bietti et al. (2024) incorporate absolute positional encoding:

$$x_t^{(0,k)} = x_t^{(0,q)} = x_t^{(0,v)} = w_E(z_t) + p_t,$$

where $p_t$ is the $t$-th absolute positional encoding. In contrast, we use a simplified relative positional encoding:

$$x_{1:t}^{(0,k)} = w_E(z_{1:t}) + R_{1-t:0}, \text{ and } x_t^{(0,q)} = x_t^{(0,v)} = w_E(z_t),$$

where positional information $R_{1-t:0} = (r_{1-t}, \ldots, r_{-1}, r_0) \in \mathbb{R}^{d \times t}$ and $r_{-i}$ represents the positional information of the $i$-th previous token $z_{t-i}$ from the current one $z_t$.

### 3.2 Associative memory

Associative memory plays an important role in analyzing the behavior of memory recall at induction head in the next section. Before defining associative memory, we first adopt a

technical assumption commonly used in theoretical studies on transformers, that is, that embeddings are high-dimensional random vectors, allowing them to be nearly orthogonal. Mathematically, we impose the following assumption.

**Assumption 1** (Near-orthogonality (Huang et al., 2024; Li et al., 2024; Bietti et al., 2024)). *The embeddings $(u_i)_i$ are d-dimensional vectors with Gaussian entries, each having a unit norm $\|u_i\| = 1$ and $u_i^\top u_j = 0$ for $i \neq j$. Also, $W_0 u_i$ forms a new vector that has a near-unit norm and near-orthogonal to $(u_i)_i$ if $W_0 \in \mathbb{R}^{d \times d}$ is a Gaussian random matrix with appropriate entries.*

With the assumption, we define associative memory.

**Definition 1** (Associative Memory). *Suppose $(u_i)_i$ and $(v_j)_j$ are the embeddings satisfying Assumption 1. Then, A matrix W is called associative memory if it can be expressed as a weighted sum of outer products of $u_i$ and $v_j$:*

$$W = \sum_{i,j} \alpha_{i,j} u_i v_j^\top, \tag{5}$$

*where $\alpha_{i,j} \in \mathbb{R}$ is the score for the pair $(u_i, v_j)$.*

Thanks to orthogonality, we can derive the score $\alpha_{i,j}$ for the pair $(u_i, v_j)$ using the operation $u_i^\top W v_j$. Henceforth, we consider the weight matrices in the attention layer and the feed-forward network of the transformer as matrices that implement associative memory.

### 3.3 Induction head

Induction heads are a specific type of attention mechanism observed in transformer models, responsible for detecting patterns in token sequences and utilizing previous occurrences of the same token to improve predictions. The behavior of induction heads is particularly relevant for tasks that require understanding and leveraging repeated patterns, such as in language modeling.

In Fig. 2, we summarize the process by which the induction head identifies and outputs repetitive patterns in the input. When the prompt $ABC \dots A$ is given, the first-layer attention, which is called **previous token head**, copies information from the previous token to the current token. Consequently, the token $B$ retains the information of $A$ in the form of $\Phi_1 A$. Then, in the second-layer attention, the associative memory $W_K^2$ matches the pair $\Phi_1 A$ with $A$, attending to the position of token $B$. As a result, the information of $B$ is stored in the last token $A$ as $W_V^2 B$. Finally, the output matrix $W_O^2$ transforms $W_V^2 B$ into $W_U(B)$, producing an output that follows the repetitive pattern.

### 3.4 Bigram model with triggered transitions

This section describes the bigram model with triggered tokens used in Bietti et al. (2024).

In this bigram language model over a vocabulary of size V, we sample trigger tokens $q_k$ from a distribution $\pi_q$ and output tokens $o_k$ from $\pi_o$ each time we generate a training sequence. Output tokens always follow their corresponding trigger tokens in the generated sequences.

$$\begin{cases} \pi_b(j \mid i) & \text{if } i \notin \{q_k\}, \\ 1\{j = o_k\} & \text{if } i = q_k. \end{cases}$$

**Motivation for the model** In written texts centered on a particular theme, it is common for identical patterns of tokens to recur multiple times. For instance, in texts about Harry Potter or machine learning, the token "Avada" is often followed by "Kedavra," and "neural" is typically followed by "network." The induction head is a computational mechanism capable of detecting repeated patterns in context and facilitating the output of the tokens that complete these patterns when a trigger token appears.

The next token following a given token $z$ is determined by conditional probabilities in a pure bigram model. As a result, bigram sequences often lack consistent repeated patterns, which prevents the induction head from emerging. In contrast, pairs of trigger and output tokens in a sequence enable two-layer transformers to learn a circuit that leverages repeated patterns, rather than relying solely on bigram conditionals.

## 4 Theoretical result

We analyze two key aspects of induction heads: the impact of positional encoding on the neglect of patterns in a sequence, and the contributions of in-context information and pretrained knowledge during inference.

### 4.1 Neglect of in-context patterns

In this section, we address the theoretical aspects of overlooking patterns in a sequence by the induction head mechanism. We argue that the learning of the previous token head fails for transformers with APE, and thus, the model cannot take patterns in the latter half of a sequence into account for the prediction. Mathematically, we observe the representation of the weight matrix $\tilde{W}_K^1$ when weight matrices are sequentially trained, in the order of $W_O^2$, $W_K^2$, and $W_K^1$, through one step of gradient descent. We refer to Theorem 3 from Bietti et al. (2024):

**Theorem 1** (Theorem 3, Bietti et al. (2024)). *For any $t > 1$, the learned matrix achieves the following associative memory:*

$$p_{t-1}^\top \tilde{W}_K^1 p_t \approx \frac{\eta\alpha\alpha'(T-1)}{T^2 t}\left\{\mathbb{P}(t_q = t - 1)\left(1 - \frac{1}{t}\right) + O\left(\frac{1}{V}\right)\right\}, \tag{6}$$

where $p_t$ is the absolute positional encoding for token position $t$, $\alpha, \alpha' \in \mathbb{R}$ are constants, and $\mathbb{P}(t_q = t - 1)$ denote the probability that the first trigger token appears at position $t - 1$,

From Eq. 6, we can see that the score for the pair of $p_{t-1}$ and $p_t$ is inversely proportional to $t$. Therefore, as $t$ increases – meaning as we move toward the latter part of the sequence – the function of the associative memory diminishes.

In contrast, we prove that transformers employing RPE consistently direct the previous token head to attend to the preceding token, regardless of the input sequence length. We calculate how the transformer's weight matrices are represented by following the same learning procedure as when using a transformer with APE. We again present the training outcome of the matrix $W_K^1$ as associative memory, which is the key component for the previous token head. We provide the complete statement and proof in Appendix E.

**Theorem 2** (informal). *Under the setup described in Sec. E.2, a two-layer attention-only transformer with relative positional encoding learns the associations by one step of gradient descent for any vocabulary $v \in \mathcal{V}$:*

$$r_{-1}^\top W_K^1 w_E(v) = \frac{\eta\alpha\alpha'}{T}\sum_{t=1}^{T}\frac{P(t_q = t - 1)}{t} \cdot O(1) + \frac{\eta\alpha\alpha'}{T}\sum_{t=1}^{T}\frac{P(t_q = t)}{t^2}\left\{O\left(\frac{1}{V}\right) + O\left(\frac{1}{T}\right)\right\},$$

*where $\eta \in \mathbb{R}$ is the learning rate, $\alpha, \alpha' \in \mathbb{R}$ are constants, $t_q$ and $T$ are the first and second occurrence positions of trigger token, and $V$ is the size of the vocabulary set $\mathcal{V}$.*

Theorem 2 suggests that the association between $r_{-1}$ and $w_E(v)$ is independent of both the vocabulary $v$ and the token position $t$. Therefore, regardless of the input sequence length, the previous token head will always attend to previous tokens with consistent strength.

**Remark 1.** *A transformer using RPE learns to attend to the previous token regardless of its position within the sequence. However, if the input sequence contains out-of-distribution tokens, the induction head mechanism does not activate. In such cases, only a transformer with APE can effectively attend to the previous token.*

We have seen that positional encodings have a significant effect on the induction head's ability to generalize to longer sequences, as shown in the comparison between APE and RPE.

In the context of length generalization in transformers, models without positional encodings (NoPE) are often discussed (Kazemnejad et al., 2024; Wang et al., 2024). To deepen our understanding of NoPE settings, we show that a three-layer transformer without positional encoding can implement the induction head mechanism.

**Proposition 1.** *A construction exists for a three-layer transformer without positional encoding that successfully achieves the induction head mechanism.*

## 4.2 Global knowledge v.s. in-context knowledge

Here, we discuss how a two-layer transformer uses statistical information obtained from the training data and the information provided in-context by calculating the logits.

**Associative memory construction achieving induction head** We first introduce the following lemma, which states that the induction head mechanism is achievable by setting the appropriate weights for the matrices in a two-layer transformer with APE.

**Lemma 1** (Bietti et al. (2024)). *Define $Q$ as the support of $\pi_q$. The induction head can be achieved by constructing matrices in the following manner.*

$$W_Q^1 = W_Q^2 = I, \qquad W_K^1 = \sum_{t=2}^{T} p_t p_{t-1}^\top,$$

$$W_K^2 = \sum_{k \in Q} w_E(k)(W_O^1 W_V^1 w_E(k))^\top, \qquad W_O^2 = \sum_{v=1}^{V} w_U(v)(W_V^2 w_E(v))^\top,$$

*where $I$ is the identity matrix, and the matrices $W_O^1$, $W_V^1$, and $W_O^2$ are a Gaussian random matrix.*

It is straightforward to prove that a two-layer transformer with RPE also has associative memory construction that achieves induction head by modifying $W_K^1 = \sum_{k \in Q} w_E(k) r_{-1}^\top$. Now, using the transformer with RPE architecture, we define an **associative memory transformer**: a model that incorporates an induction head and a feed-forward network storing knowledge learned through pretraining.

**Definition 2** (associative memory transformer). *A two-layer transformer with RPE is called **associative memory transformer** if its weight matrices are set as in Lemma 1 except for*

$$W_K^1 = \sum_{k \in Q} w_E(k) r_{-1}^\top, \qquad W_1 = \begin{pmatrix} w_E(v_1)^\top \\ w_E(v_2)^\top \\ \vdots \\ w_E(v_V)^\top \end{pmatrix},$$

$$W_2 = \left( \sum_{u=1}^{V} \log \pi_b(u \mid v_1) w_U(u) \quad \sum_{u=1}^{V} \log \pi_b(u \mid v_2) w_U(u) \quad \dots \quad \sum_{u=1}^{V} \log \pi_b(u \mid v_V) w_U(u) \right),$$

This construction reveals that the feed-forward layer in Eq. 3 of Sec. 3.1 functions explicitly as a key-value memory (Geva et al., 2021). Each row of $W_1$ (key) acts as a detector for an input pattern $w_E(v)$, and each column of $W_2$ (value) encodes global knowledge, i.e., the distribution $\pi_b$ over the output vocabulary. Appendix D discusses the difference from the architecture of Bietti et al. (2024). Note that when the probability $\pi_b(u \mid v)$ approaches $+0$, its logarithm $\log \pi_b(u \mid v)$ diverges to $-\infty$. If we admit $\log \pi_b(u \mid v) = -\infty$, the transformer never outputs $u$ regardless of context. This does not align with the actual behavior of the induction head. Also, due to the limitations of floating-point representation in computers, it is not possible to represent $\infty$ directly. Thus, we consider $\epsilon$ as threshold and set $\pi_b(u \mid v) = \epsilon$.

**Associative memory transformer on limited input sequences.** We present the following proposition, which focuses on two token patterns in the input sequence.

**Proposition 2.** *Suppose a two-layer transformer is an associative memory transformer. Given a length-$T$ sequence $z_{1:T}$ where the last token is $z_T = q$, let the $t_1$-th token be $z_{t_1} = v_1$ following $z_{t_1-1} = q$, and let the $t_2$-th token be $z_{t_2} = v_2$ following $z_{t_2-1} = q$, where $v_1 \neq v_2$ and $(3 \leq)t_1 < t_2$ without loss of generality. Also assume that there exists only one occurrence for each of $v_1$ and $v_2$ in the context $z_{1:T}$. Then, the logits $\xi_{v_1}$ and $\xi_{v_2}$ can be expressed as follows:*

$$\xi_{v_1} = \frac{e^{\frac{e}{t_1+e-1}}}{Z} + \log \pi_b(v_1 \mid q), \qquad \xi_{v_2} = \frac{e^{\frac{1+e}{t_2+e-1}}}{Z} + \log \pi_b(v_2 \mid q), \tag{7}$$

*where $Z$ is the normalization factor of the softmax function and $e$ is Euler's number.*

We can observe a general trend from Eqs. 7. First, if the model contains global knowledge of both tokens $v_1$ and $v_2$, the output token is influenced by the knowledge, in addition to the in-context knowledge. In particular, the logits of tokens that rarely appear in the training dataset become significantly low. In our associative memory transformer model, global knowledge never promotes the output of any specific token; rather, it solely suppresses inappropriate predictions, because we always have $\log \pi_b(\cdot \mid q) \leq 0$. Second, if token patterns, such as $q\,v_1$ and $q\,v_2$, do not appear in the training data, global knowledge affects the token logits to the same extent, due to the way $W_F$ is defined using the threshold $\epsilon$ in Def. 2.

It is worth noting that an associative memory transformer exhibits a significant difference in scale when handling global knowledge and in-context knowledge. For instance, if $\pi_b(v \mid q) = 0.1$, then $\log \pi_b(v \mid q) \approx -2.3$. Even with the functionality of an induction head, it is unlikely to predict the token patterns from the context. This is avoidable by reformulating $W_O^2$ as in Def. 3 and adjusting the strength of induction head. We consider this setting in the next paragraph, along with more general input sequences.

If neither token appears during training, the difference between the two logits before applying the softmax function $\xi'_{v_2} - \xi'_{v_1}$ is calculated as follows:

$$\xi'_{v_2} - \xi'_{v_1} = \frac{e(t_1 - t_2) + t_1 + e - 1}{(t_1 + e - 1)(t_2 + e - 1)}. \tag{8}$$

The result implies that the token $v_2$ is more likely to be generated if $t_1$ is large and the distance between the tokens $v_1$ and $v_2$, i.e., $t_2 - t_1$ is small. This trend arises from the first layer of the attention block, where the attention is not fully concentrated on the previous token, resulting in a diffusion of attention across other tokens as well. In other words, the more tokens that precede the current token, the lower the attention score assigned to the current token.

**Stronger associative memory on general input sequences.** Next, we introduce a more sophisticated analysis of in-context and global knowledge by relaxing assumptions in the sequence $z_{1:T}$. The transformer considered here has associative memory with larger scores, characterized by a weighted sum of the individual terms of the weight matrices in Sec. 3.3, where each term is scaled by an appropriate coefficient. We consider a special case in which each term has the same coefficient $\tau_1, \tau_2, \tau_3 \in \mathbb{R}$.

**Definition 3** (stronger associative memory transformer). *We redefine the following matrices $W_K^1, W_K^2$ and $W_O^2$ in Def. 2 for a stronger associative memory transformer*

$$\hat{W}_K^1 = \tau_1 W_K^1, \quad \hat{W}_K^2 = \tau_2 W_K^2, \quad \hat{W}_O^2 = \tau_3 W_O^2, \tag{9}$$

*with the parameters $\tau_1, \tau_2, \tau_3 \in \mathbb{R}_+$. The other weight matrices remain the same.*

**Proposition 3.** *Suppose a two-layer transformer is a stronger associative memory transformer. Given a length-$T$ sequence $z_{1:T}$ where the last token is $z_T = q$, let $f(v)$ be the number of token pattern "$q\,v$" appearing in the sequence for vocabulary $v \in \mathcal{V}$. For sufficiently large $\tau_1$ and $\tau_2$, the logits $\xi_v$ can be expressed as follows:*

$$\xi_v \underset{\substack{\tau_1 \\ \tau_2}}{\approx} \log \pi_b(v \mid q) + \tau_3 \cdot \frac{f(v) + \mathbf{1}\{v = q\}\mathbf{1}\{z_1 = q\}}{\left(\sum_{v'=1}^{V} f(v')\right) + \mathbf{1}\{z_1 = q\}}. \tag{10}$$

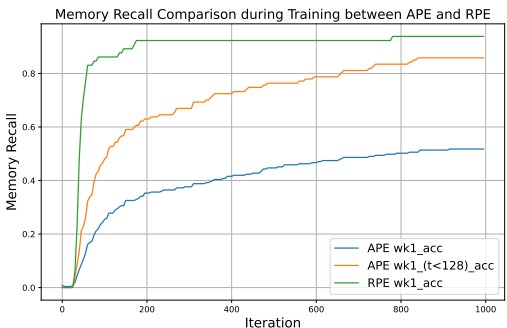

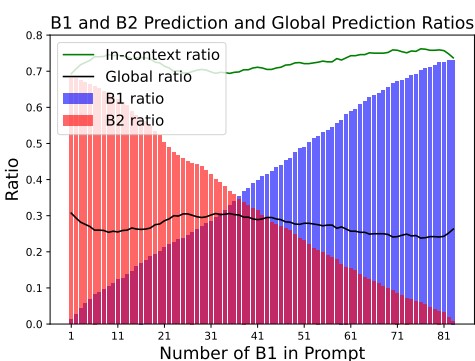

(a) Even when trained with sequences of length 256, the attention block in the first layer of TF$_{\text{APE}}$ fails to attend to the previous token at positions $t > 128$, while TF$_{\text{RPE}}$ attends to previous tokens regardless of the positions.

(b) Ratio of predicting $B_1$ and $B_2$ given the prompt "$A\,B_1\,,\ldots,\,A\,B_1\,,\,A\,B_2\,,\ldots,\,A\,B_2\,,\,A$". The transformer shows an increasing tendency to complete the context with $B_1$ as the frequency of token pattern $A\,B_1$ in the context grows.

Figure 1: (a) Comparison of a two-layer transformer with APE (Thm. 1) and RPE (Thm. 2) capturing previous token information. (b) Output behavior as the frequency of the token pattern $A\,B_1$ increases in the input sequence (Prop. 3).

It can be observed that the transformer equipped with a general associative memory is affected by the proportions of each token pattern present in the context as well as by the global knowledge. In this case, it can be concluded that the information regarding the positions where token patterns are observed does not influence the final logit. We can easily derive the following properties.

**Corollary 1.** *Given a length-T sequence $z_{1:T}$ where the last token is $z_T = q$ and the only token patterns $q\,v_1$ and $q\,v_2$ appear $f(v_1)$ and $f(v_2)$ times, respectively. The transformer outputs one of the following:* $\arg\max_{v\in\mathcal{V}\setminus\{v_1,v_2\}} \pi_b(v \mid q)$, $v_1$, *or* $v_2$.

For vocabulary $v \neq v_1, v_2$, the logit $\xi_v$ is determined only by $\log \pi_b(v \mid q)$ and this means that the candidate for the next token is $\arg\max_{v\in\mathcal{V}\setminus\{v_1,v_2\}} \pi_b(v \mid q)$. On the other hand, The logits for $v_1$ and $v_2$ have the other term in Prop. 3, which makes them other candidates for the next token. Overall, Corollary 1 states that the model output depends on the bigram conditionals $\pi_b(\cdot \mid q)$ and the frequency of token patterns $qv_1$ and $qv_2$ in the context.

## 5 Experiments

This section empirically examines how positional encoding affects the transformer's ability to capture patterns in input sequences, as discussed in Thm. 1 and 2, and how the transformer's output changes depending on the proportion of pattern occurrences within the context, as stated in Prop. 3. Henceforth, we denote by TF$_{\text{ape}}$ and TF$_{\text{rpe}}$ two-layer transformer with APE, RPE.

### 5.1 Neglect of in-context knowledge

**Training process of previous token head**  We investigated whether the training of transformers directs attention towards the previous token in the first attention layer. Specifically, TF$_{\text{rpe}}$ and TF$_{\text{ape}}$ were trained using sequences of length 256 generated according to the sequence generation rule. Details regarding the hyperparameters and sequence generation process are provided in Appendix G.

For evaluation of the learned associative memory $\sum_{(i,j)\in M} u_i v_j^\top$, we employed the memory recall metric (Dar et al., 2023; Geva et al., 2023; Bietti et al., 2024):

$$R(W) = \frac{1}{\|M\|} \sum_{(i,j)\in M} \mathbf{1}\{\arg\max_{i'} u_{i'}^\top W v_j = i\}.$$

For TF$_{\text{rpe}}$, we want each vocabulary $w_E(k)$ to be paired with the information of the previous position $r_{-1}$. Hence, $R(W_K^1)$ is computed as

$$\frac{1}{\|Q\|} \sum_{k\in Q} \mathbf{1}\{\arg\max_{i'} r_{i'}^\top W_K^1 w_E(k) = -1\}.$$

Similarly, we have for TF$_{\text{ape}}$

$$\frac{1}{T} \sum_{t=2}^{T} \mathbf{1}\{\arg\max_{t'} p_{t'}^\top W_K^1 p_t = t - 1\},$$

where $T$ is the maximum input length for the transformer.

In the RPE setting, the metric quantifies the amount of vocabulary tokens that are more strongly associated with the relative position embedding $r_{-1}$ than any other position embedding in the learned associative memory. We can find in Fig. 1a that the recall approaches 1.0 as training progresses. This indicates that nearly all vocabulary items are successfully paired with $r_{-1}$, confirming the effectiveness of RPE for associative memory formation. In contrast, for APE, we observe that memory recall is substantially higher for positions t < 128 compared to the full positions, even though they are trained with sequences of length 256. This supports our theoretical claim that it is harder to embed the relationship between $p_t$ and $p_{t-1}$ into associative memory when t is large, due to the diminishing alignment between position embeddings at distant positions.

**Discussion** One of the worst-case scenarios for TF$_{\text{ape}}$ occurs when a trigger token does not appear by chance in the range $t \le 128$ and only appears for the first time at $t > 128$. Although such a sequence is contrived, its probability of occurrence is non-zero. For such sequences, models with an incomplete previous token head fail to associate the trigger token's information with the output token, resulting in incorrect predictions without leveraging in-context information. In contrast, TF$_{\text{rpe}}$, which links words to their relative positions, ensures that the in-context information is not overlooked, regardless of where the trigger token first appears, thereby enabling correct predictions.

## 5.2 Global knowledge v.s. in-context knowledge

To validate the results derived from the theory, we conducted experiments using real data.

**Analogical reasoning task** We utilize the analogical reasoning task, one of the most practical tasks for bigram analysis. Specifically, we focus on the "capital-world" analogy type of questions from the Google Analogy Dataset (Mikolov et al., 2013), which contains a total of 19,544 questions. This subset includes 232 unique vocabulary items. Denoting a word $A$ (e.g., *Tokyo*) and its corresponding analogy counterpart $A^*$ (e.g., *Japan*), the prompts were constructed in the form $A\,A^*$, $B\,B^*$, $C\,C^*$, ..., with pairs separated by commas. Using these generated prompts, we trained TF$_{\text{rpe}}$. For details on the training and dataset setup, please refer to Appendix G.

**Collision of context information** As shown in Corollary 1, the model predicts either a token based on global knowledge or the token that appears in the context. To confirm this, we randomly sampled $A$ from the capitals in the Google Analogy Dataset and constructed prompts using word pairs $A\,B_1$ and $A\,B_2$ that were not learned as global knowledge. The resulting prompts were of the form $A\,B_1$, ..., $A\,B_1$, $A\,B_2$, ..., $A\,B_2$, $A$, which were then used for predicting the next token. We generated 1,000 prompts and calculated the proportion of cases where either $B_1$ or $B_2$ was predicted, while controlling the frequency of $A\,B_1$ and $A\,B_2$ in the context.

The result is shown in Fig. 1b. As the number of token pattern $A\,B_1$ increases, the model predicts $B_1$ as the next token more frequently. Furthermore, it can be observed that the prediction trend reverses when the frequency of $B_1$ and $B_2$ in the context is nearly equal. Additionally, we observe from global ratio that the model predicted some vocabulary $B$ from global knowledge. Especially, the figure illustrates that it is more likely to output global knowledge when the number of $B_1$ and $B_2$ are the same, or when the number of $B_1$ or $B_2$ is almost maximum. Please refer to Appendix H.2 for further discussion on these phenomena.

## 6  Conclusion

We analyzed the influence of interaction between global bigram and in-context knowledge to the two-layer transformer through the lens of associative memory. We theoretically and empirically verified that relative positional encoding empowers transformers with the ability to capture information in longer sequences, and that how the next token prediction is conducted within the transformer that was trained to store global knowledge and to have induction head.

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

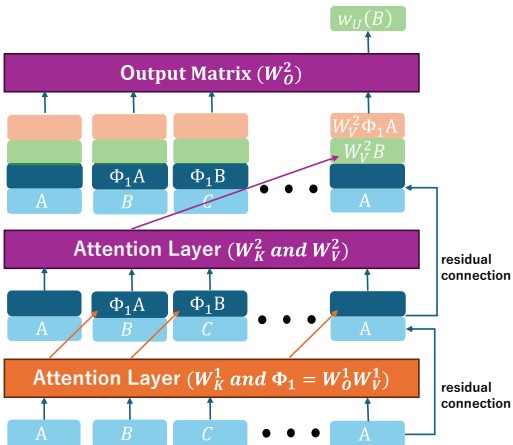

Figure 2: The visualization of induction head mechanism. The first attention layer copies the previous token information. Then, the current token matches the copied information to find the probable next token.

## A  Limitations

Our study has certain limitations that should be acknowledged. First, the analysis presented in this work is conducted using a two-layer transformer model. While this allows for a controlled and interpretable exploration of the underlying mechanisms, the findings may not directly generalize to LLMs, which typically involve significantly more layers and complex interactions.

Second, while our focus on induction heads provides valuable insights into ICL, it is important to note that ICL leverages additional computational circuits beyond induction heads. As such, our explanation captures only part of the broader mechanisms underlying ICL, leaving room for further investigation into other contributing factors.

## B  Ethics statement

This work is purely theoretical, supported by controlled experiments designed to validate the proposed analyses. No personal or sensitive data were involved at any stage of this research. All experiments were conducted using datasets that are publicly available, ensuring compliance with ethical standards for data usage.

## C  Additional related work

**In-context learning and induction head**    Bayesian inference is also a popular perspective for comprehending ICL. Xie et al. (2022) and Wies et al. (2024) investigated how a pretrained transformer implicitly discovers the underlying concept or latent task of a prompt by leveraging a mixture of distributions. An alternative line of work (Zhang et al., 2023) relates the Bayesian model averaging to the attention mechanism. Wang et al. (2023b) studies how information is aggregated to label words in ICL. In terms of two-layer transformer, Chen et al. (2024b) studied ICL on $n$-gram Markov chain, proving that gradient flow optimization under the cross-entropy loss leads to a model that exhibits a generalized induction head mechanism. Edelman et al. (2024) showed two-layer attention-only transformers learn induction heads that estimate the next token's conditional probability based on prior occurrences, enabling near Bayes-optimal performance. Similarly, Nichani et al. (2024) showed that two-layer transformers learn causal structure via gradient descent, suggesting that induction head is a special case of this structure. Ren et al. (2024) discovered a more generalized version of induction heads, which are referred to as semantic induction heads.

**Length generalization and positional encoding**   Although we point out the neglect of in-context information by a transformer with APE, this can be considered as the failure of length generalization. Since the introduction of the Transformer architecture (Vaswani, 2017), there has been a wealth of research on positional encoding (Zhao et al., 2024). Originally, Vaswani (2017) adopted sinusoidal positional encoding that transforms the tokens' absolute positions. Subsequently, Kiyono et al. (2021) and Likhomanenko et al. (2021) proposed shifting absolute sinusoidal positional encoding during training to achieve shift invariance. Wang et al. (2020) extends word embedding as continuous functions with respect to token position, leading to smooth word representations. Meanwhile, it is demonstrated that RPE is superior to APE for longer sequence generalization in various tasks (Neishi & Yoshinaga, 2019; Likhomanenko et al., 2021; Huang et al., 2020; Jelassi et al., 2023). Additionally, Sinha et al. (2022) questioned whether models can learn the relative distance between words when equipped with APE. The first RPE was formulated by Shaw et al. (2018), where a trainable encoding is added to the key before computing the dot product with the query. The transformer in our work also adopts this type of positional encoding, while we fix the positional encodings to randomly initialized vectors. A variety of different RPE methods were presented (Huang et al., 2020; Ke et al., 2021) until now, but DeBERTa (He et al., 2021), RoPE (Su et al., 2024), T5-Bias (Raffel et al., 2020), and ALiBi (Press et al., 2021) are the most popular choices. Ruoss et al. (2023) suggested that randomizing absolute and relative positional encoding enhance the model's ability to generalize to input sequences of lengths not encountered during training, but they used the encoder-ony Transformer, which is different from ours.

**Short summary of Bietti et al. (2024)**   The remarkable success of LLMs has increased the need for a deeper understanding of their internal mechanisms and for enhancing their reliability. Existing studies lack detailed insights into how reasoning abilities evolve during the learning process based on contextual information. Bietti et al. (2024) explored the dynamics of balancing in-context knowledge and global knowledge through a bigram model, analyzing the development of these abilities as part of the training dynamics. They conceptualized the weight matrices of transformers as associative memory and theoretically demonstrated that the induction head mechanism can be trained by proposing the storage of specific embedding pairs from training data. In their experiments, they trained a two-layer transformer on a bigram model estimated from the tiny Shakespeare dataset. The proximity of each weight matrix to the theoretically derived weights was measured using a memory recall probe. They analyzed how these weights changed over the course of training. Although this work is the most relevant to our research, we note that while they primarily focus on the learning dynamics of induction heads in a two-layer transformer, our primary emphasis lies on length generalization and the reasoning process.

## D   MLP as a key-value memory

Geva et al. (2021) discovered that the feed-forward layers in transformer-based language models function as key-value memory. In a transformer, the feed-forward layer is typically computed as

$$F(x) = W_2(\text{ReLU}(W_1 x)),$$

where $x \in \mathbb{R}^d$ is the input vector, and $W_1 \in \mathbb{R}^{d_h \times d}, W_2 \in \mathbb{R}^{d_o \times d_h}$. According to Geva et al. (2021), the rows of $W_1$ (keys) serve to detect specific patterns in the input sequence, while the columns of $W_2$ (values) have been shown to represent distributions over the output vocabulary.

Bietti et al. (2024) uses a transformer architecture that uses a linear projection $W_F$ instead of an MLP layer.

$$W_F = \sum_{v=1}^{V} \sum_{u=1}^{V} \log \pi_b(u \mid v) w_U(u) w_E(v)^\top.$$

Due to the near-orthogonality of the embedding vectors, this projection maps an input $x = w_E(v)$ to the output $\sum_{u=1}^{V} \log \pi_b(u \mid v) w_U(u)$. This enables the matrix $W_F$ to function

as an associative memory that stores the pairs of an embedding vector and corresponding global knowledge.

Now, suppose we construct the matrices $W_1$ and $W_2$ as follows:

$$W_1 = \begin{pmatrix} w_E(v_1)^\top \\ w_E(v_2)^\top \\ \vdots \\ w_E(v_V)^\top \end{pmatrix}, \tag{11}$$

$$W_2 = \left( \sum_{u=1}^{V} \log \pi_b(u \mid v_1) w_U(u) \quad \sum_{u=1}^{V} \log \pi_b(u \mid v_2) w_U(u) \cdots \sum_{u=1}^{V} \log \pi_b(u \mid v_V) w_U(u) \right), \tag{12}$$

Each row of $W_1$ acts as a detector for an input pattern $w_E(v)$, such that only the corresponding entry is activated when $w_E(v_i)$ is provided as input. Each row of $W_2$, in turn, encodes global knowledge, i.e., the distribution $\pi_b$ over the output vocabulary conditioned on the corresponding input token.

We can easily find that the resulting computation matches exactly that of the linear projection $W_F$ under the near-orthgonality condition.

Therefore, this construction confirms that the feed-forward layer operates as a key-value memory.

# E  Learning of ideal transformer

## E.1  Notation table

We prepare tab. 2, which provides a concise summary of the notations used throughout the appendices.

| Symbol | Description |
|--------|-------------|
| $d$ | Dimensionality of the positional encoding |
| $\mathcal{V}$ | Vocabulary set |
| $V$ | Vocabulary size |
| $w_E(z_t)$ | Embedding of token $z_t$ |
| $r_{s-t}$ | Relative positional encoding expressing the relation between $s$-th token and current one |
| $\sigma$ | softmax function |
| $\Phi_1$ | product of two randomly initialized matrices $W_O^1$ and $W_V^1$ |
| $t_q$ | Index of the first trigger token |
| $t_o$ | Index of the output token corresponding to the first trigger token, $t_o = t_q + 1$ |
| $T$ | Index of the second trigger token, or the input length |
| $[N]$ | a set of natural number up to $N$, $\{1, 2, \ldots, N\}$ |
| $l$ | cross entropy loss function |
| $\eta$ | learning rate |
| $\tau$ | constant, defined by $\mathbb{E}_{t_o}[\sum_{t=t_o}^{T} 1/t]$ |
| $\alpha$ | constant, $\alpha = \eta/VT$ |
| $\alpha'$ | constant, $\alpha = \eta\tau/VT$ |
| $q$ | a variable representing the trigger token |
| $\pi_u$ | a uniform distribution from which $z_1$ is sampled. |
| $\pi_q$ | a uniform distribution by which the trigger token is determined. |
| $\pi_o$ | a uniform distribution by which the output token is determined. |
| $\pi_b$ | a uniform distribution from which $z_{2:T}$ is sampled conditioned on the previous token. |

Table 2: Table of Notations

## E.2 Setup

Here we consider a simplified setting to analyze the influence of relative positional encoding in the training dynamics of our two-layer attention only transformer. Specifically,

1. **Input Sequence** We consider an input sequence $z_{1:T} \in \mathcal{V}^T$ which has one trigger token $q$ appearing twice, and ends with the trigger token. In other words, let $t_q$ be the first occurrence position. Then, we have $z_{t_q} = z_T = q$. From the bigram generation rule in Sec.3.4, $z_{t_q+1}$ and $z_{T+1}$ are the output token.

2. **Probability Distribution Assumptions**: The bigram is generated by uniform distributions over $[V]$ for any index $i$, i.e., $\pi_u$, $\pi_q$, $\pi_o$, and $\pi_b(\cdot \mid i)$ are uniformly distributed.

3. **Simplification of Loss Function**: Consider the loss only for sequences of length $T$ where the final input token $z_T$ is the second occurrence of the trigger token, and the label $y$ is $z_{T+1}$, the corresponding output token.

4. **Simplification for Learning Focus**: Our approach involves sequentially training $W_2^O$, $W_2^K$, and $W_1^K$ from top to bottom. We employ zero-initialization and carry out a single gradient descent step.

5. **Initialization and Freezing**: To achieve our goal of showing that $W_O^2$, $W_K^2$ and $W_K^1$ learn to be an associative memory, we zero-initialize the three matrices. For other matrices such as $W_V^2$, $W_V^1$ and $W_O^1$ are randomly initialized from Gaussian distribution. We set $W_Q^1$ and $W_Q^2$ to identity matrix. All these matrices except for $W_O^2$, $W_K^2$ and $W_K^1$ are frozen during the training process.

## E.3 Theoretical analysis of learning

**Theorem 3 (formal).** *Under the setup described in Sec. E.2, a two-layer attention only transformer with relative positional encoding learns associations of the associative memory transformer, i.e., the memories $W_O^2$, $W_K^2$ and $W_K^1$ can be written as the sum of the terms that constitute the associative memory transformer.*

$$W_K^1 = \sum_{k \in Q} \chi(k) w_E(k) r_{-1}^\top + (W_K^1)', \tag{13}$$

$$W_K^2 = \sum_{k \in Q} \psi(k) w_E(k) (W_O^1 W_V^1 w_E(k))^\top + (W_K^2)', \tag{14}$$

$$W_O^2 = \sum_{v=1}^{V} \omega(v) w_U(v) (W_V^2 w_E(v))^\top + (W_O^2)', \tag{15}$$

*where $\chi, \psi : \mathcal{V} \to \mathbb{R}$ and $\omega : \mathbb{N} \to \mathbb{R}$ denotes the learned score of each association, and $(W_K^1)', (W_K^2)'$ and $(W_O^2)'$ are the other associations.*

*Proof.* We will demonstrate that associative memory can be learned by performing one gradient descent step in a top-down manner, starting with $W_O^2$, followed by $W_K^2$, and then $W_K^1$.

**Step1: Training of $W_O^2$**

We begin with the dynamics of $W_O^2$. The input to the second layer attention, $x_t^{(1)}$, is composed of the sum of the values from the first layer attention, $x_t^{(1,0)}$, and the values from the residual connection, $x_t^{(1,1)}$. Since $W_K^1$ is initialized to zero, the attention of the first layer is evenly distributed across all the tokens $z_{1:t}$ and we have

$$x_t^{(1,0)} = \frac{1}{t} \sum_{s=1}^{t} \Phi_1 w_E(z_s),$$

where $\Phi_1 = W_O^1 W_V^1$. Additionally, the residual stream carries the token embedding:

$$x_t^{(1,1)} = w_E(z_t).$$

Since the matrix $W_K^2$ is also zero-initialized, the value of the second layer attention can be computed through

$$x_t^{(2)} = W_V^2 \left( \frac{1}{t} \sum_{s=1}^{t} x_s^{(1)} \right).$$

Now the logit prediction is given by $W_U(W_O^2 x_T^{(2)} + x_T^{(1)})$ with the residual connection. We note that when $d$ is large, thanks to the near-orthogonality, $W_U x_T^{(1)} = W_U \left( \sum_{s=1}^{T} \Phi_1 w_E(z_s)/T + w_E(z_T) \right)$ is negligible because $x_T^{(1)}$ does not contain $w_U(v)$ for any $v \in \mathcal{V}$. This enables us to use Lemma 2 and one gradient step with learning-rate $\eta$ yields

$$W_O^2 = \frac{\eta}{V} \sum_{k=1}^{V} w_U(k) \mathbb{E}[x_T^{(2)} \mid y = k]^\top - \frac{\eta}{V} \sum_{k=1}^{V} w_U(k) \mathbb{E} \left[ \frac{\hat{p}_W(k \mid x)}{p(y = k)} x_T^{(2)} \right]^\top$$

$$= \frac{\eta}{V} \sum_{k=1}^{V} w_U(k)(\mathbb{E}[x_T^{(2)} \mid y = k] - \mathbb{E}[x_T^{(2)}])^\top. \tag{16}$$

This transformation comes from the assumption that every token is sampled from a uniform distribution ($p(y = k) = 1/V$) and all the logits are 0 when $W_O^2 = O_{d \times d}$, which means that $\hat{p}_W(k \mid x) = 1/V$.

Given that $y = k$, it holds that the first output token $z_{t_o} = k$. For $z_i$ ($i \neq t_o$), the uniform distribution can possibly generate any token, independently of the condition $y = k$. Since we have

$$x_T^{(2)} = \frac{1}{T} \sum_{t=1}^{T} W_V^2 \left( w_E(z_t) + \frac{1}{t} \sum_{s=1}^{t} \Phi_1 w_E(z_s) \right),$$

the expression $\mathbb{E} \left[ x_T^{(2)} \mid y = k \right] - \mathbb{E} \left[ x_T^{(2)} \right]$ leaves only the terms related to $z_{t_o}$. This yields

$$\mathbb{E} \left[ x_T^{(2)} \mid y = k \right] - \mathbb{E} \left[ x_T^{(2)} \right]$$

$$= \frac{1}{T} W_V^2 (\mathbb{E}[w_E(z_{t_o}) \mid y = k] - \mathbb{E}[w_E(z_{t_o})])$$

$$+ \frac{1}{T} \cdot \mathbb{E} \left[ \sum_{t=t_o}^{T} W_V^2 \frac{1}{t} \Phi_1 w_E(z_{t_o}) \mid y = k \right]$$

$$- \frac{1}{T} \cdot \mathbb{E} \left[ \sum_{t=t_o}^{T} W_V^2 \frac{1}{t} \Phi_1 w_E(z_{t_o}) \right] \tag{17}$$

$$= \frac{1}{T} W_V^2 (w_E(k) - \mathbb{E}[w_E(z_{t_o})]) + \frac{\tau}{T} W_V^2 \Phi_1 (w_E(k) - \mathbb{E}[w_E(z_{t_o})]), \tag{18}$$

where $\tau := \mathbb{E} \left[ \sum_{t=t_o}^{T} \frac{1}{t} \right]$. Since each token is generated from a uniform distribution, the expected value of $w_E(z_{t_o})$, conditioned on nothing, is given by:

$$\mathbb{E} \left[ w_E(z_{t_o}) \right] = \frac{1}{V} \sum_{k=1}^{V} w_E(k).$$

Combining eq. 16 and 17, the near-orthogonality of Gaussian vectors gives us

$$w_U(k)^\top W_O^2 W_V^2 w_E(j) \approx \frac{\eta}{VT} \mathbf{1}(k = j) + O\left( \frac{\eta}{V^2 T} \right), \tag{19}$$

$$w_U(k)^\top W_O^2 W_V^2 \Phi_1 w_E(j) \approx \frac{\eta \tau}{VT} \mathbf{1}(k = j) + O\left( \frac{\eta \tau}{V^2 T} \right). \tag{20}$$

This is the same result as in the proof from Bietti et al. (2024). So far, the weight matrix $W_O^2$ is trained to perform as an associative memory so that

$$w_U(k)^\top W_O^2 W_V^2 w_E(z_t) \approx \alpha \mathbf{1}\{z_t = k\}, \tag{21}$$

with a constant $\alpha = \frac{\eta}{TV}$.

**Step2: Training of $W_K^2$**

Since the training of $W_O^2$ is finished, we have Eq. 19. When $W_K^1 = W_K^2 = O_{d \times d}$, Lemma 4 demonstrates that the next token prediction is distributed among all tokens, which means that $\hat{p}(v \mid x) = 1/V$ for any vocabulary $v$. According to Lemma 5, one step of gradient descent yields

$$W_K^2 = \frac{\eta}{TN} \sum_{k,t} \mathbb{E}\left[ w_U(k)^\top \Phi_2 x_t^{(1)} \cdot (x_t^{(1)} - \bar{x}^{(1)}) \times (x_T^{(1)})^\top \mid y = k \right]$$

$$- \frac{\eta}{TN} \sum_{k,t} \mathbb{E}\left[ w_U(k)^\top \Phi_2 x_t^{(1)} \cdot (x_t^{(1)} - \bar{x}^{(1)}) \times (x_T^{(1)})^\top \right], \tag{22}$$

with $\bar{x}^{(1)} = \frac{1}{T} \sum_{t=1}^T x_t^{(0)}$.

We simplify our transformer architecture by setting $x_t^{(1,1)}$ as the queries and values, and $x_t^{(1,0)}$ as the keys. Furthermore, we add the condition that the trigger token $z_{t_q} = j$ and rewrite eq. 22. This conditioning leverages the fact that each token in $z_{t_q}$ is sampled from a uniform distribution. Now we have

$$W_K^2 = \frac{\eta}{TN} \sum_{k,t} \frac{1}{N} \sum_{j=1}^N \mathbb{E}\left[ w_U(k)^\top \Phi_2 x_t^{(1,1)} \times (x_t^{(1,0)} - \bar{x}^{(1,0)})(x_T^{(1,1)})^\top \mid y = k, z_{t_q} = j \right]$$

$$- \frac{\eta}{TN} \sum_{k,t} \frac{1}{N} \sum_{j=1}^N \mathbb{E}\left[ w_U(k)^\top \Phi_2 x_t^{(1,1)} \times (x_t^{(1,0)} - \bar{x}^{(1,0)})(x_T^{(1,1)})^\top \mid z_{t_q} = j \right],$$

where $\bar{x}^{(1,0)} = \frac{1}{T} \sum_{t=1}^T x_t^{(1,0)}$. Using the formula 21, we obtain

$$W_K^2 \approx \frac{\alpha\eta}{TN^2} \sum_{j=1}^N \sum_{k=1}^N \mathbb{E}\left[ \sum_{t=1}^T \mathbb{1}\{z_t = k\} \times (x_t^{(1,0)} - \bar{x}^{(1,0)}) w_E(z_T)^\top \mid y = k, z_{t_q} = j \right]$$

$$- \frac{\alpha\eta}{TN^2} \sum_{j=1}^N \sum_{k=1}^N \mathbb{E}\left[ \sum_{t=1}^T \mathbb{1}\{z_t = k\} \times (x_t^{(1,0)} - \bar{x}^{(1,0)}) w_E(z_T)^\top \mid z_{t_q} = j \right]$$

$$= \frac{\alpha\eta}{TN^2} \sum_{j=1}^N \sum_{k=1}^N \Delta_{k,j} w_E(j)^\top, \tag{23}$$

where $\Delta_{k,j}$ is defined by

$$\Delta_{k,j} := \mathbb{E}\left[ \sum_{t=1}^T \mathbb{1}\{z_t = k\} \times (x_t^{(1,0)} - \bar{x}^{(1,0)}) \mid y = k, z_{t_q} = j \right]$$

$$- \mathbb{E}\left[ \sum_{t=1}^T \mathbb{1}\{z_t = k\} \times (x_t^{(1,0)} - \bar{x}^{(1,0)}) \mid z_{t_q} = j \right].$$

The sum in $\Delta_{k,j}$ can be partitioned into three distinct groups: (i) $\Delta_{k,j}^o$, where $t$ is the index of the output token; (ii) $\Delta_{k,j}^q$, where $t$ corresponds to the indices of the trigger tokens; and (iii) $\Delta_{k,j}^r$, which includes all other cases. Mathematically, let $t_o \geq 2$ be a random variable, and $t_q = t_o - 1$, we write

$$\Delta_{k,j}^o := \mathbb{E}\left[ \mathbb{1}\{z_{t_o} = k\}(x_{t_o}^{(1,0)} - \bar{x}^{(1,0)}) \mid y = k, z_{t_q} = j \right]$$

$$- \mathbb{E}\left[\mathbb{1}\{z_{t_o} = k\}(x_{t_o}^{(1,0)} - \bar{x}^{(1,0)}) \mid z_{t_q} = j\right],$$

$$\Delta_{k,j}^q := \mathbb{E}\left[\sum_{t \in \mathcal{T}_q} \mathbb{1}\{z_t = k\}(x_t^{(1,0)} - \bar{x}^{(1,0)}) \mid y = k, z_{t_q} = j\right]$$

$$- \mathbb{E}\left[\sum_{t \in \mathcal{T}_q} \mathbb{1}\{z_t = k\}(x_t^{(1,0)} - \bar{x}^{(1,0)}) \mid z_{t_q} = j\right],$$

$$\Delta_{k,j}^r := \mathbb{E}\left[\sum_{t \in \mathcal{T}_r} \mathbb{1}\{z_t = k\}(x_t^{(1,0)} - \bar{x}^{(1,0)}) \mid y = k, z_{t_q} = j\right]$$

$$- \mathbb{E}\left[\sum_{t \in \mathcal{T}_r} \mathbb{1}\{z_t = k\}(x_t^{(1,0)} - \bar{x}^{(1,0)}) \mid z_{t_q} = j\right],$$

where $\mathcal{T}_q = \{t_q, T\}$ and $\mathcal{T}_r = \{1 \leq i \leq T \mid i \in \mathbb{N}, i \neq t_o, i \neq t_q, i \neq T\}$.

First, we will take a look at $\Delta_{k,j}^o$. Note that $\mathbb{E}[\mathbb{1}\{z_{t_o} = k\} \mid y = k, z_{t_q} = j] = 1$ and $\mathbb{E}[\mathbb{1}\{z_{t_o} = k\} \mid z_{t_q} = j] = 1/N$, so we find that

$$\Delta_{k,j}^o = \left(1 - \frac{1}{N}\right) \mathbb{E}\left[x_t^{(1,0)} - \bar{x}^{(1,0)} \mid y = k\right]$$

$$= \frac{N-1}{N} \mathbb{E}\left[\frac{1}{t_o}\sum_{s=1}^{t_o} \Phi_1 w_E(z_s) - \frac{1}{T}\sum_{t=1}^{T}\frac{1}{t}\sum_{s=1}^{t} \Phi_1 w_E(z_s)\right]$$

$$= \frac{N-1}{N} \sum_{i=1}^{N} a_{k,j,i} \Phi_1 w_E(i), \tag{24}$$

where $a_{k,j,i}$ is the coefficient of $\Phi_1 w_E(i)$ by calculating

$$\mathbb{E}\left[\frac{1}{t_o}\sum_{s=1}^{t_o} \Phi_1 w_E(z_s) - \frac{1}{T}\sum_{t=1}^{T}\frac{1}{t}\sum_{s=1}^{t} \Phi_1 w_E(z_s)\right].$$

The analysis of $a_{k,j,i}$ is done by the work (Bietti et al., 2024):

$$\frac{1}{N}\sum_{k=1}^{N} a_{k,j,i} \approx \begin{cases} O\left(\frac{1}{N}\right), & \text{if } i \neq j, \\ \Omega\left(\frac{1}{T}\right), & \text{otherwise.} \end{cases}$$

Thus, we have the following approximation:

$$(\Phi_1 w_E(i))^\top \left(\frac{1}{N}\sum_{k=1}^{N} \Delta_{k,j}^o\right) \approx \begin{cases} O\left(\frac{1}{N}\right), & \text{if } i \neq j, \\ \Omega\left(\frac{1}{T}\right), & \text{if } i = j. \end{cases} \tag{25}$$

Similarly, we have $\Delta_{k,j}^q = O\left(\frac{1}{N}\right)$ and $\Delta_{k,j}^r = O\left(\frac{T}{N}\right)$ based on the discussion in Bietti et al. (2024). By combining these analysis with Eq. 25, we multiply $(\Phi_1 w_E(i))^\top$ from left and $w_E(j)$ from right to Eq. 23 and we establish that

$$(\Phi_1 w_E(i))^\top W_K^2 w_E(j) \approx \frac{\alpha\eta}{TN}\left\{\Omega\left(\frac{1}{T}\right)\mathbb{1}\{i = j\} + O\left(\frac{T}{N}\right)\right\} \tag{26}$$

**Step3: Training of $W_K^1$**
So far, the gradient descent step as to $W_O^2$ and $W_K^2$ gives them the ability to behave as an associative memory:

$$w_U(k)^\top W_O^2 W_V^2 w_E(z_t) = \alpha\mathbf{1}\{z_t = k\}, \tag{27}$$

$$(\Phi_1 w_E(z_t))^\top W_K^2 w_E(z_T) = \alpha' \mathbf{1}\{z_t = z_T\}. \tag{28}$$

From Lemma 3, we admit that the model predicts the next token almost randomly, i.e., $\hat{p}(v \mid x) = 1/V$ for any vocabulary $v$, when $W_K^0 = O_{d \times d}$. This enables us to employ Lemma 5 and $W_K^1$ after one gradient descent step gives the following:

$$W_K^1$$
$$= \frac{\eta}{V} \sum_{k=1}^N \mathbb{E}\left[ \frac{1}{T} \sum_{t=1}^T w_U(k)^\top \Phi_2 w_E(z_t) \frac{1}{t} \sum_{s=1}^t (\Phi_1 w_E(z_s))^\top W_K^2 w_E(z_T) q_{s,t} w_E(z_t)^\top \mid y = k \right]$$
$$- \frac{\eta}{V} \sum_{i=k}^N \mathbb{E}\left[ \frac{1}{T} \sum_{t=1}^T w_U(k)^\top \Phi_2 w_E(z_t) \frac{1}{t} \sum_{s=1}^t (\Phi_1 w_E(z_s))^\top W_K^2 w_E(z_T) q_{s,t} w_E(z_t)^\top \right]$$
$$- \frac{\eta}{V} \sum_{k=1}^N \mathbb{E}\left[ \left( w_U(k)^\top \Phi_2 \bar{x}_{1:T} \right) \frac{1}{T} \sum_{t=1}^T \frac{1}{t} \sum_{s=1}^t (\Phi_1 w_E(z_s))^\top W_K^2 w_E(z_T) q_{s,t} w_E(z_t)^\top \mid y = k \right]$$
$$+ \frac{\eta}{V} \sum_{k=1}^N \mathbb{E}\left[ \left( w_U(k)^\top \Phi_2 \bar{x}_{1:T} \right) \frac{1}{T} \sum_{t=1}^T \frac{1}{t} \sum_{s=1}^t (\Phi_1 w_E(z_s))^\top W_K^2 w_E(z_T) q_{s,t} w_E(z_t)^\top \right],$$

with $q_{s,t} = (w_E(z_s) + r_{s-t}) - (\bar{w}_E(z_{1:t}) + \bar{r}_{1-t:0})$ and $\bar{w}_E(z_{1:t}) = \frac{1}{t} \sum_{i=1}^t w_E(z_i)$. Now, fix an arbitrary vocabulary $v \in \mathcal{V}$. Given that

$$w_U(k)^\top \Phi_2 w_E(z_t) = \alpha \mathbf{1}\{z_t = k\}$$

and that

$$(\Phi_1 w_E(z_s))^\top W_K^2 w_E(z_T) = \alpha' \mathbf{1}\{z_s = z_T\},$$

which equals $\alpha'$ when $s = t_q$ or $s = T$, we manipulate the expression:

$$W_K^1$$
$$= \frac{\eta}{V} \sum_{k=1}^V \mathbb{E}\left[ \frac{1}{T} \sum_{t=1}^T \alpha \mathbf{1}\{z_t = k\} \frac{\alpha'}{t} (\mathbf{1}\{t_q \le t\} q_{t_q,t} + \mathbf{1}\{t = T\} q_{T,T}) w_E(z_t)^\top \mid y = k \right]$$
$$- \frac{\eta}{V} \sum_{k=1}^V \mathbb{E}\left[ \frac{1}{T} \sum_{t=1}^T \alpha \mathbf{1}\{z_t = k\} \frac{\alpha'}{t} (\mathbf{1}\{t_q \le t\} q_{t_q,t} + \mathbf{1}\{t = T\} q_{T,T}) w_E(z_t)^\top \right]$$
$$- \frac{\eta}{V} \sum_{k=1}^V \mathbb{E}\left[ \frac{1}{T} \sum_{t'=1}^T \mathbf{1}\{z_t' = k\} \frac{1}{T} \sum_{t=1}^T \frac{\alpha'}{t} (\mathbf{1}\{t_q \le t\} q_{t_q,t} + \mathbf{1}\{t = T\} q_{T,T}) w_E(z_t)^\top \mid y = k \right]$$
$$+ \frac{\eta}{V} \sum_{k=1}^V \mathbb{E}\left[ \frac{1}{T} \sum_{t'=1}^T \mathbf{1}\{z_t' = k\} \frac{1}{T} \sum_{t=1}^T \frac{\alpha'}{t} (\mathbf{1}\{t_q \le t\} q_{t_q,t} + \mathbf{1}\{t = T\} q_{T,T}) w_E(z_t)^\top \right].$$

With the near-orthogonality of the token embeddings, we write

$$W_K^1 w_E(v) = \frac{\eta \alpha \alpha'}{VT} \sum_{k=1}^V \sum_{t=1}^T \frac{1}{t} (A_{t,k} - B_{t,k} - C_{t,k} + D_{t,k})$$
$$+ \frac{\eta \alpha \alpha'}{VT^2} \sum_{k=1}^V (A'_{T,k} - B'_{T,k} - C'_{T,k} + D'_{T,k}). \tag{29}$$

where, we define each term as follows:

$$A_{t,k} = \mathbb{E}\left[ \mathbf{1}\{z_t = k\} \mathbf{1}\{t_q \le t\} q_{t_q,t} \mathbf{1}\{z_t = v\} \mid y = k \right],$$
$$B_{t,k} = \mathbb{E}\left[ \mathbf{1}\{z_t = k\} \mathbf{1}\{t_q \le t\} q_{t_q,t} \mathbf{1}\{z_t = v\} \right],$$
$$C_{t,k} = \mathbb{E}\left[ \frac{1}{T} \sum_{t'=1}^T \mathbf{1}\{z_t' = k\} \mathbf{1}\{t_q \le t\} q_{t_q,t} \mathbf{1}\{z_t = v\} \mid y = k \right],$$

$$D_{t,k} = \mathbb{E}\left[\frac{1}{T}\sum_{t'=1}^{T}\mathbf{1}\{z_t' = k\}\mathbf{1}\{t_q \le t\}q_{t_q,t}\mathbf{1}\{z_t = v\}\right],$$

$$A_{T,k}' = \mathbb{E}\left[\mathbf{1}\{z_T = k\}q_{T,T}\mathbf{1}\{z_T = v\} \mid y = k\right],$$

$$B_{T,k}' = \mathbb{E}\left[\mathbf{1}\{z_T = k\}q_{T,T}\mathbf{1}\{z_T = v\}\right],$$

$$C_{T,k}' = \mathbb{E}\left[\frac{1}{T}\sum_{t'=1}^{T}\mathbf{1}\{z_t' = k\}q_{T,T}\mathbf{1}\{z_T = v\} \mid y = k\right],$$

$$D_{T,k}' = \mathbb{E}\left[\frac{1}{T}\sum_{t'=1}^{T}\mathbf{1}\{z_t' = k\}q_{T,T}\mathbf{1}\{z_T = v\}\right].$$

We will further split these four terms based on relative positional encoding and token embedding that are contained in $q_{t_q,t}$. For example, we decompose $A_{t,k}$ into $A_{t,k}^R$ and $A_{t,k}^z$ by defining

$$A_{t,k}^R = \mathbb{E}\left[\mathbf{1}\{z_t = k\}\mathbf{1}\{t_q \le t\}\left(r_{t_q-t} - \bar{r}_{1-t:0}\right)\mathbf{1}\{z_t = v\} \mid y = k\right],$$

$$A_{t,k}^z = \mathbb{E}\left[\mathbf{1}\{z_t = k\}\mathbf{1}\{t_q \le t\}\left(w_E(z_{t_q}) - \bar{w}_E(z_{1:t})\right)\mathbf{1}\{z_t = v\} \mid y = k\right].$$

**(i) when $v = k$**
We begin with $A_{t,k}^R$:

$$A_{t,k}^R = \mathbb{E}\left[\mathbf{1}\{z_t = k\}\mathbf{1}\{t_q \le t\}\left(r_{t_q-t} - \bar{r}_{1-t:0}\right)\mathbf{1}\{z_t = v\} \mid y = k\right]$$

$$= \sum_{s=1}^{t} P(t_q = s \mid y = k)\mathbb{E}\left[\mathbf{1}\{z_t = k\} \mid y = k, t_q = s\right](r_{s-t} - \bar{r}_{1-t:0})$$

$$= P(t_q = t - 1)(r_{-1} - \bar{r}_{1-t:0}) + \mathbf{1}\{t = T\}P(t_q = t)(r_0 - \bar{r}_{1-t:0})$$

$$\quad + \frac{\mathbf{1}\{t = T\}}{V}\sum_{s \in [t-2]\cup\{t\}} P(t_q = s)(r_{s-t} - \bar{r}_{1-t:0})$$

$$= P(t_q = t - 1)(r_{-1} - \bar{r}_{1-t:0}) + \mathbf{1}\{t = T\}P(t_q = t)(r_0 - \bar{r}_{1-t:0}) + O\left(\frac{1}{V}\right).$$

Here, we used Lemma 6 and 7 for the calculation of the expectation $\mathbb{E}[\mathbf{1}\{z_t = k\} \mid \cdot]$. Similarly, we will examine $B_{t,k}^R$, $C_{t,k}^R$ and $D_{t,k}^R$ as well.

$$B_{t,k}^R = \mathbb{E}\left[\mathbf{1}\{z_t = k\}\mathbf{1}\{t_q \le t\}\left(r_{t_q-t} - \bar{r}_{1-t:0}\right)\mathbf{1}\{z_t = v\}\right]$$

$$= \sum_{s=1}^{t} P(t_q = s)\mathbb{E}[\mathbf{1}\{z_t = k\} \mid t_q = s](r_{s-t} - \bar{r}_{1-t:0})$$

$$= \sum_{s=1}^{t} \frac{P(t_q = s)}{V}(r_{s-t} - \bar{r}_{1-t:0})$$

$$= O\left(\frac{1}{V}\right).$$

For the term $C_{t,k}^R$, we make use of Lemma 8 and 9.

$$C_{t,k}^R$$

$$= \mathbb{E}\left[\frac{1}{T}\sum_{t'=1}^{T}\mathbf{1}\{z_{t'} = k\}\mathbf{1}\{t_q \le t\}\left(r_{t_q-t} - \bar{r}_{1-t:0}\right)\mathbf{1}\{z_t = v\} \mid y = k\right]$$

$$= \frac{1}{T}\sum_{s=1}^{t} P(t_q = s)(r_{s-t} - \bar{r}_{1-t:0})\mathbb{E}\left[\left(\sum_{t'=1}^{T}\mathbf{1}\{z_{t'} = k\}\right)\mathbf{1}\{z_t = v\} \mid y = k, t_q = s\right]$$

$$= \frac{P(t_q = t - 1)}{T}(r_{-1} - \bar{r}_{1-t:0})\left\{(1 + \mathbf{1}\{t = T\}) + \frac{(T - 3) \cdot \mathbf{1}\{t \neq T\}}{V - 1}\right\}$$

$$+ \frac{2P(t_q = t)}{T}(r_0 - \bar{r}_{1-t:0}) + \sum_{s=1}^{t-2} \frac{P(t_q = s)}{T} O\left(\frac{1}{V}\right)$$

$$= \frac{1}{T}\left\{P(t_q = t - 1)(1 + \mathbf{1}\{t = T\})(r_{-1} - \bar{r}_{1-t:0}) + 2P(t_q = t)(r_0 - \bar{r}_{1-t:0}) + O\left(\frac{1}{V}\right)\right\}.$$

As for $D_{t,k}^R$, the condition $y = k$ does not exist, and $z_t$ follows a uniform distribution. Therefore, we obtain

$$D_{t,k}^R = \mathbb{E}\left[\frac{1}{T}\sum_{t'=1}^{T}\mathbf{1}\{z_{t'} = k\}\mathbf{1}\{t_q \leq t\}\left(r_{t_q - t} - \bar{r}_{1-t:0}\right)\mathbf{1}\{z_t = v\}\right]$$

$$= \sum_{s=1}^{t}\frac{P(t_q = s)}{T}(r_{s-t} - \bar{r}_{1-t:0})\mathbb{E}\left[\sum_{t'=1}^{T}\mathbf{1}\{z_{t'} = k\}\mathbf{1}\{z_t = v\} \mid y = k, t_q = s\right]$$

$$= O\left(\frac{1}{V}\right).$$

Next, we will examine $A_{t,k}^z, B_{t,k}^z, C_{t,k}^z$ and $D_{t,k}^z$ in this order. Using lemma 10 and 11, we obtain.

$$A_{t,k}^z = \mathbb{E}\left[\mathbf{1}\{z_t = k\}\mathbf{1}\{t_q \leq t\}(x_{t_q} - \bar{x}_{1:t})\mathbf{1}\{z_t = v\} \mid y = k\right]$$

$$= \mathbb{E}\left[\mathbf{1}\{z_t = k\}\mathbf{1}\{t_q \leq t\}x_{t_q} \mid y = k\right] - \mathbb{E}\left[\mathbf{1}\{z_t = k\}\mathbf{1}\{t_q \leq t\}\bar{x}_{1:t} \mid y = k\right]$$

$$= P(t_q = t - 1)\mathbf{1}\{t = T\}w_E(k) + P(t_q = t)\mathbf{1}\{t = T - 1\})w_E(k)$$

$$- \frac{P(t_q = t - 1)}{t}(1 + \mathbf{1}\{t = T\})w_E(k) - \frac{\mathbf{1}\{t = T - 1\})P(t_q = t)}{t}w_E(k) + O\left(\frac{1}{V}\right).$$

We also get

$$B_{t,k}^z = \mathbb{E}\left[\mathbf{1}\{z_t = k\}\mathbf{1}\{t_q \leq t\}(x_{t_q} - \bar{x}_{1:t})\mathbf{1}\{z_t = v\}\right]$$

$$= \mathbb{E}\left[\mathbf{1}\{z_t = k\}\mathbf{1}\{t_q \leq t\}x_{t_q}\right] - \mathbb{E}\left[\mathbf{1}\{z_t = k\}\mathbf{1}\{t_q \leq t\}\bar{x}_{1:t}\right]$$

$$= P(z_t = k)\mathbb{E}[\mathbf{1}\{t_q \leq t\}x_{t_q} \mid z_t = k] - P(z_t = k)\mathbb{E}\left[\mathbf{1}\{t_q \leq t\}\bar{x}_{1:t} \mid z_t = k\right]$$

$$= O\left(\frac{1}{V}\right).$$

Now we move to $C_{t,k}^z$. From lemma 12 and 13, it follows that

$$C_{t,k}^z = \mathbb{E}\left[\frac{1}{T}\left(\sum_{t'=1}^{T}\mathbf{1}\{z_{t'} = k\}\right)\mathbf{1}\{t_q \leq t\}(x_{t_q} - \bar{x}_{1:t})\mathbf{1}\{z_t = v\} \mid y = k\right]$$

$$= \mathbb{E}\left[\frac{1}{T}\left(\sum_{t'=1}^{T}\mathbf{1}\{z_{t'} = k\}\right)\mathbf{1}\{t_q \leq t\}x_{t_q}\mathbf{1}\{z_t = k\} \mid y = k\right]$$

$$- \mathbb{E}\left[\frac{1}{T}\left(\sum_{t'=1}^{T}\mathbf{1}\{z_{t'} = k\}\right)\mathbf{1}\{t_q \leq t\}\bar{x}_{1:t}\mathbf{1}\{z_t = k\} \mid y = k\right]$$

$$= w_E(k)\left\{\frac{2P(t_q = t - 1)}{T}\mathbf{1}\{t = T\} + \frac{2P(t_q = t)}{T}\mathbf{1}\{t = T - 1\}\right.$$

$$\left. - \frac{P(t_q = t - 1)}{T}\frac{4 \cdot \mathbf{1}\{t = T\} + \mathbf{1}\{t \neq T\}}{t} - \frac{P(t_q = t)}{T}\frac{2 \cdot \mathbf{1}\{t = T - 1\}}{t}\right\}$$

$$+ \frac{1}{T} \cdot O\left(\frac{1}{V}\right).$$

For $D_{t,k}^z$, we have

$$
\begin{aligned}
D_{t,k}^z &= \mathbb{E}\left[\frac{1}{T}\left(\sum_{t'=1}^{T}\mathbf{1}\{z_{t'} = k\}\right)\mathbf{1}\{t_q \le t\}(x_{t_q} - \bar{x}_{1:t})\mathbf{1}\{z_t = v\}\right] \\
&= P(z_t = v)\mathbb{E}\left[\frac{1}{T}\left(\sum_{t'=1}^{T}\mathbf{1}\{z_{t'} = k\}\right)\mathbf{1}\{t_q \le t\}(x_{t_q} - \bar{x}_{1:t}) \mid z_t = v\right] \\
&= O\left(\frac{1}{V}\right),
\end{aligned}
$$

because of the term $\mathbf{1}\{z_t = v\}$ under no condition. So far, we have calculated $A_{t,k}, B_{t,k}, C_{t,k}$ and $D_{t,k}$ for $v = k$. Now, we turn our attention to $v \neq k$.

**(ii) when $v \neq k$**

In this case, it is guaranteed that $\mathbf{1}\{z_t = k\} \neq \mathbf{1}\{z_t = v\}$, and thus, we have $A_{t,k} = B_{t,k} = 0$. We only need to examine $C_{t,k}$ and $D_{t,k}$. We use the fact that the token $z_t$ has to be sampled uniformly from a potentially reduced vocabulary space with some words excluded depending on the position of $t$, we obtain

$$
\begin{aligned}
C_{t,k} &= \mathbb{E}\left[\frac{1}{T}\sum_{t'=1}^{T}\mathbf{1}\{z_t' = k\}\mathbf{1}\{t_q \le t\}q_{t_q,t}\mathbf{1}\{z_t = v\} \mid y = k\right] \\
&= P(z_t = v \mid y = k)\mathbb{E}\left[\frac{1}{T}\sum_{t'=1}^{T}\mathbf{1}\{z_t' = k\}\mathbf{1}\{t_q \le t\}q_{t_q,t} \mid y = k, z_t = v\right] \\
&= O\left(\frac{1}{V}\right),
\end{aligned}
$$

and

$$
\begin{aligned}
D_{t,k} &= \mathbb{E}\left[\frac{1}{T}\left(\sum_{t'=1}^{T}\mathbf{1}\{z_{t'} = k\}\right)\mathbf{1}\{t_q \le t\}(x_{t_q} - \bar{x}_{1:t})\mathbf{1}\{z_t = v\}\right] \\
&= P(z_t = v \mid y = k)\mathbb{E}\left[\frac{1}{T}\left(\sum_{t'=1}^{T}\mathbf{1}\{z_{t'} = k\}\right)\mathbf{1}\{t_q \le t\}q_{t_q,t}\mathbf{1}\{z_t = v\}\right] \\
&= O\left(\frac{1}{V}\right).
\end{aligned}
$$

So far, we have seen the content of $A_{t,k}, B_{t,k}, C_{t,k}$ and $D_{t,k}$. Now, we will shift our focus to $A'_{T,k}, B'_{T,k}, C'_{T,k}$ and $D'_{T,k}$. We recall that

$$
q_{s,t} = (w_E(z_s) + r_{s-t}) - (\bar{w}_E(z_{1:t}) + \bar{r}_{1-t:0}).
$$

By substituting $s = T$ and $t = T$, we have

$$
q_{T,T} = (w_E(z_T) + r_0) - (\bar{w}_E(z_{1:T}) + \bar{r}_{1-T:0}).
$$

Also, $\mathbf{1}\{z_T = v\}$ implies that the trigger token is $v$. we use Lemma 14 and obtain the following:

$$
\begin{aligned}
A'_{T,k} &= \mathbb{E}\left[\mathbf{1}\{z_T = k\}q_{T,T}\mathbf{1}\{z_T = v\} \mid y = k\right] \\
&= \begin{cases} P(t_q = T-1)\left(r_0 - \bar{r}_{1-T:0}\right) + P(t_q = T-1)\frac{T-2}{T}w_E(k) + O\left(\frac{1}{V}\right) & \text{if } v = k, \\ 0 & \text{otherwise.} \end{cases}
\end{aligned}
$$

Similarly to the argument of $B_{t,k}$ and $D_{t,k}$, the trigger token is chosen uniformly, and thus, we have $P(z_T = v) = \frac{1}{V}$. This gives

$$
B'_{T,k} = \mathbb{E}\left[\mathbf{1}\{z_T = k\}q_{T,T}\mathbf{1}\{z_T = v\}\right]
$$

$$
= \begin{cases} O\left(\frac{1}{V}\right) & \text{if } v = k, \\ 0 & \text{otherwise,} \end{cases}
$$

and

$$
D'_{T,k} = \mathbb{E}\left[\frac{1}{T}\sum_{t'=1}^{T}\mathbf{1}\{z_{t'}=k\}q_{T,T}\mathbf{1}\{z_T=v\}\right]
$$

$$
= \begin{cases} O\left(\frac{1}{V}\right) & \text{if } v = k, \\ 0 & \text{otherwise,} \end{cases}
$$

Finally, according to Lemma 15, it is established that

$$
C'_{T,k} = \mathbb{E}\left[\frac{1}{T}\sum_{t'=1}^{T}\mathbf{1}\{z_{t'}=k\}q_{T,T}\mathbf{1}\{z_T=v\} \mid y = k\right]
$$

$$
= \begin{cases} \frac{2P(t_q=T-1)}{T}\left(r_0 - \bar{r}_{1-T:0}\right) + \frac{2P(t_q=T-1)}{T}\frac{T-2}{T}w_E(k) + \frac{1}{T}\cdot O\left(\frac{1}{V}\right) & \text{if } v = k, \\ \frac{1}{T}O\left(\frac{1}{V}\right) & \text{otherwise.} \end{cases}
$$

To sum up, we get the following associative memory behaviors by examining Eq. 29. We point out that $\bar{r}_{1-t:0}$ also contains $r_{-1}$ and $r_0$.

$$
r_{-1}W_K^1 w_E(v) = \frac{\eta\alpha\alpha'}{T}\sum_{t=1}^{T}\frac{P(t_q=t-1)}{t}\left(1 - \frac{1+\mathbf{1}\{t=T\}}{T}\right)\left(1 - \frac{1}{t}\right)
$$

$$
- \frac{\eta\alpha\alpha'}{T}\sum_{t=1}^{T}\frac{P(t_q=t)}{t^2}\left(\mathbf{1}\{t=T\} - \frac{2}{T}\right)
$$

$$
- \frac{\eta\alpha\alpha'}{VT^2}\sum_{k=1}^{V}\frac{P(t_q=T-1)}{t}\left(1 - \frac{2}{T}\right),
$$

and

$$
r_0 W_K^1 w_E(v) = \frac{\eta\alpha\alpha'}{T}\sum_{t=1}^{T}\frac{P(t_q=t)}{t}\left(\mathbf{1}\{t=T\} - \frac{2}{T}\right)\left(1 - \frac{1}{t}\right)
$$

$$
- \frac{\eta\alpha\alpha'}{T}\sum_{t=1}^{T}\frac{P(t_q=t-1)}{t^2}\left(1 - \frac{1+\mathbf{1}\{t=T\}}{T}\right)
$$

$$
+ \frac{\eta\alpha\alpha'}{VT^2}\sum_{k=1}^{V}P(t_q=T-1)\left(1 - \frac{2}{T}\right).
$$

Also, for any $j \in \mathcal{V}$, we have

$$
w_E(j)W_K^1 w_E(v) = O\left(\frac{1}{V}\right).
$$

The proof is finished. □

## F   Other lemmas and proofs

**Lemma 2** (Lemma 2, Bietti et al. (2024)). *Given a data distribution $p$ over pairs $(x, y) \in \mathbb{R}^d \times [V]$ and a fixed matrix $W_U \in \mathbb{R}^{d\times d}$, minimizing the loss $L(W) = \mathbb{E}_{(x,y)\sim p}\left[\ell\left(y, W_U W x\right)\right]$ gives the following gradient:*

$$
\nabla_W L(W) = \sum_{v=1}^{V} p(y=v)w_U(v)(\hat{\mu}_v - \mu_v)^\top,
$$

*where $\mu_k = \mathbb{E}[x \mid y = v]$ and $\hat{\mu}_v = \mathbb{E}_x\left[\frac{\hat{p}_W(v|x)}{p(y=v)}x\right]$, and $\hat{p}_W(v \mid x)$ is the probability of generating $v$.*

**Lemma 3.** *Consider a two-layer attention only transformer that meets the initialization condition in Sec. E.2, and the input sequence of length $T$ ends with the trigger token $z_T = q$ at its second occurrence. Even after the gradient descent step of $W_O^2$ and $W_K^2$, the prediction $\hat{p}(v \mid z_{1:T}) = 1/V$ for any $v \in \mathcal{V}$.*

*Proof.* From the way the transformer is initialized, we have $W_K^1 = O_{d \times d}$. Therefore, the output of the first attention block for any $t$ is:

$$x_t^{(1)} = w_E(z_t) + \frac{1}{t} \sum_{s=1}^{t} \Phi_1 w_E(z_s).$$

In the second layer, the key matrix is trained such that $(\Phi_1 w_E(z_t))^\top W_K^2 w_E(z_T) = \alpha' \mathbf{1}\{z_t = z_T\}$. Thus, we have

$$\sigma\left((x_{1:T}^{(1)})^\top (W_K^2)^\top W_Q^2 x_T^{(1)}\right) = \sigma\left(\begin{pmatrix} (w_E(z_1) + \Phi_1 w_E(z_1))^\top \\ \left(w_E(z_2) + \sum_{i=1}^{2} \frac{\Phi_1 w_E(z_i)}{2}\right)^\top \\ \vdots \\ \left(w_E(z_{T-1}) + \sum_{i=1}^{T-1} \frac{\Phi_1 w_E(z_i)}{T-1}\right)^\top \\ \left(w_E(z_T) + \sum_{i=1}^{T} \frac{\Phi_1 w_E(z_i)}{T}\right)^\top \end{pmatrix}\right.$$

$$\times \left.\left(\sum_{k \in Q} (\Phi_1 w_E(k)) w_E(k)^\top\right) I x_T^{(1)}\right)$$

$$= \sigma\left(\begin{pmatrix} (w_E(z_1) + \Phi_1 w_E(z_1))^\top \\ \left(w_E(z_2) + \sum_{i=1}^{2} \frac{\Phi_1 w_E(z_i)}{2}\right)^\top \\ \vdots \\ \left(w_E(z_{t-1}) + \sum_{i=1}^{t-1} \frac{\Phi_1 w_E(z_i)}{t-1}\right)^\top \\ \left(w_E(z_t) + \sum_{i=1}^{t} \frac{\Phi_1 w_E(z_i)}{t}\right)^\top \end{pmatrix} \Phi_1 w_E(q)\right)$$

$$= \sigma\left(\frac{\eta\tau}{TV} \begin{pmatrix} 0 \\ \vdots \\ 0 \\ 1/(t_o - 1) \\ 1/t_o \\ \vdots \\ 1/(T-1) \\ 1/T \end{pmatrix}\right).$$

When $V$ is large enough, the attention is spread across the whole sequence evenly, and we obtain

$$x_T^{(2)} \approx \frac{1}{T} \sum_{t=1}^{T} W_O^2 W_V^2 x_t^{(1)} + x_T^{(1)} \tag{30}$$

$$= \frac{1}{T} \sum_{t=1}^{T} W_O^2 W_V^2 \left(w_E(z_t) + \frac{1}{t} \sum_{s=1}^{t} \Phi_1 w_E(z_s)\right) + w_E(z_T) + \frac{1}{T} \sum_{s=1}^{T} \Phi_1 w_E(z_s)$$

$$=_{w_U} \frac{\alpha}{T} \sum_{t=1}^{T} w_U(z_t),$$

where we focus on the terms represented by $w_U(v)$ for vocabulary $v$ by $=_{w_U}$. Taking the expectation over $z_{1:T}$ while keeping the trigger token $z_T = z_{t_o-1} = q$ fixed yields

$$\mathbb{E}[x_T^{(2)}] =_{w_U} \sum_{j=1}^{V} \mathbb{E}\left[x_T^{(2)} \mid q = j\right]$$

$$= \frac{\alpha}{T} \sum_{j=1}^{V} \left( \frac{T-2}{V-1} \sum_{\substack{1 \leq v \leq V \\ v \neq j}} w_U(v) + 2w_U(j) \right)$$

The coefficients of $w_U(v)$ are the same for any $v \in \mathcal{V}$ and this concludes the proof. $\quad\square$

**Lemma 4.** *Consider a two-layer attention only transformer that meets the initialization condition in Sec. E.2, and the input sequence of length $T$ ends with the trigger token $z_T = q$ at its second occurrence. After the gradient descent step of $W_O^2$, the prediction $\hat{p}(v \mid z_{1:T}) = 1/V$ for any $v \in \mathcal{V}$.*

*Proof.* Since $W_K^2 = O_{d \times d}$, it is guaranteed that we have Eq. 30 as an equation. Thus, the result follows directly from Lemma 3. $\quad\square$

**Lemma 5.** *Suppose the loss function is given by*

$$L(W) = \mathbb{E}_{(X,y)}[l(y, \xi(X))]$$
$$\xi(X) = W_U \Phi_2 X \bar{\sigma}(Z(W)^\top W_2 x_T),$$

*where $Z(W) = (z_1(W), \ldots, z_T(W))$ with $z_t(W) = \sum_{s=1}^{t} \Phi_1 x_s \sigma((x_{1:t} + R_{1-t:0})^\top W x_t)_s$, and $\bar{\sigma}(u_{1:T})_t = \frac{1}{T}(1 + u_t - \frac{1}{T}\sum_{s=1}^{T} u_s)$ is the linearization of the softmax function around 0. Then the gradient at $W = 0$ has the following form.*

$$\nabla_W L(W) = \sum_{i=1}^{N} \mathbb{E}\left[ \frac{1}{T} \sum_{t=1}^{T} w_U(i)^\top \Phi_2 x_t \frac{1}{t} \sum_{s=1}^{t} (\Phi_1 x_s)^\top W_K^2 x_T q_{s,t} x_t^\top \mid y = i \right]$$

$$- \sum_{i=1}^{N} \mathbb{E}\left[ \frac{1}{T} \sum_{t=1}^{T} w_U(i)^\top \Phi_2 x_t \frac{1}{t} \sum_{s=1}^{t} (\Phi_1 x_s)^\top W_K^2 x_T q_{s,t} x_t^\top \right]$$

$$- \sum_{i=1}^{N} \mathbb{E}\left[ w_U(i)^\top \Phi_2 \bar{x}_{1:T} \frac{1}{T} \sum_{t=1}^{T} \frac{1}{t} \sum_{s=1}^{t} (\Phi_1 x_s)^\top W_K^2 x_T q_{s,t} x_t^\top \mid y = i \right]$$

$$+ \sum_{i=1}^{N} \mathbb{E}\left[ w_U(i)^\top \Phi_2 \bar{x}_{1:T} \frac{1}{T} \sum_{t=1}^{T} \frac{1}{t} \sum_{s=1}^{t} (\Phi_1 x_s)^\top W_K^2 x_T q_{s,t} x_t^\top \right],$$

*where $q_{s,t} = (x_s + r_{s-t}) - (\bar{x}_{1:t} - \bar{r}_{1-t:0})$ and $\bar{x}_{1:t} = \frac{1}{t}\sum_{s=1}^{t} x_s$.*

*Proof.* The result is obtained by replacing $p_{1:t}^\top$ with $(x_{1:t} + R_{1-t:0})^\top$ and $p_t$ with $x_t$ in $z_t$ from the proof of Lemma 5 from Bietti et al. (2024). $\quad\square$

It is worth noting that $x_i = x_i^{(0)} = w_E(z_i)$ and that if we directly follows the transformer architecture the vector in the linearized version of softmax $\bar{\sigma}$ should be $Z'(W)^\top W_2 z_T'(W)$, where $Z'(W) = (z_1'(W), \ldots, z_T'(W))$ with $z_t'(W) = x_t + \sum_{s=1}^{t} \Phi_1 x_s \sigma((x_{1:t} + R_{1-t:0})^\top W x_t)_s$. Since $W_K^2$ is learned so that it only cares the ordered pair of $\Phi_1 w_E(z_t)$ and $w_E(z_t)$, the loss function is simplified and ignores the other terms. This also applies to $\xi(X)$, where

$$\xi'(X) = W_U \Phi_2 Z' \bar{\sigma}(Z'(W)^\top W_2 z_T')$$

is the authentic form of the output. Since we know $W_O^2$ (and therefore $\Phi_2$) is already trained, we can simplify the expression.

### F.1 Supporting lemmas

This section provides useful lemmas to show the learning of matrices, especially for $W_K^1$.

**Lemma 6.** *Given a sequence $z_{1:T}$ that satisfies Sec. E.2. For any $t \in [1, T]$, and any vocabulary $k \in \mathcal{V}$, we have*

$$\mathbb{E}[\mathbf{1}\{z_t = k\} \mid y = k, t_q = t - 1] = 1.$$

*Proof.* Since the trigger token is $z_{t_q} = z_{t-1}$, the next token $z_t$ is the output token. The condition $y = k$ gives us $z_t = k$. Therefore, $\mathbf{1}\{z_t = k\} = 1$ and the equation holds. $\square$

**Lemma 7.** *Given a sequence $z_{1:T}$ that satisfies Sec. E.2. Fix some $t \in [1, T] \subset \mathbb{N}$. For any $s \in [1, t - 2] \cup \{t\}$, and vocabulary $k \in \mathcal{V}$, we have*

$$\mathbb{E}[\mathbf{1}\{z_t = k\} \mid y = k, t_q = s] = \begin{cases} 1 & \text{if } s = t \text{ and } t = T - 1, \\ \frac{1}{V-1} & \text{if } s \le t - 2 \text{ and } t \ne T, \\ 0 & \text{otherwise}, \end{cases} \quad (31)$$

*Proof.* First, consider the case $s = t$. Under the condition $t_q = s$, we have $t_q = s = t$, and thus, $z_t$ is the trigger token. The only possible case allowing the same token for trigger and output is when $t_q = T - 1$. This yields $z_{T-1} = z_T = k$ because $z_T$ not only works as trigger, but also output. Therefore, when $s = t$, we have

$$\mathbb{E}[\mathbf{1}\{z_t = k\} \mid y = k, t_q = s] = \begin{cases} 1 & \text{if } t = T - 1, \\ 0 & \text{otherwise}. \end{cases} \quad (32)$$

Next, we focus on $s \le t - 2$. Suppose we are given $t = T$. The token $z_t$ is then the second trigger token, and $\mathbf{1}\{z_t = k\} = 1$ implies that $z_{t_q} = k$. However, since the first trigger position is $t_q = s \le t - 2$ and the output token is also $y = k$, $z_T(= k)$ cannot be the second trigger occurrence. Hence, it suffices to consider $t \ne T$. We now have $s \le t - 2 < T - 2$. Assuming $z_{t_q} = k$ will also lead to contradiction because the trigger token $k$ appears more than twice at positions $s, s + 1$ and $T$. With this in mind, we obtain

$$\begin{aligned}
\mathbb{E}[\mathbf{1}\{z_t = k\} \mid y = k, t_q = s] &= P(z_t = k \mid y = k, t_q = s) \\
&= \sum_{q \in \mathcal{V} \setminus \{k\}} P(z_t = k, z_{t_q} = q \mid \cdot) \\
&= \sum_{q \in \mathcal{V} \setminus \{k\}} P(z_t = k \mid \cdot, z_{t_q} = q) P(z_{t_q} = q \mid \cdot) \\
&= \sum_{q \in \mathcal{V} \setminus \{k\}} \frac{1}{V - 1} \frac{1}{V - 1} \\
&= \frac{1}{V - 1}. \quad (33)
\end{aligned}$$

Here we use $P(z_{t_q} = q \mid y = k, t_q = s) = 1/(V - 1)$ because the trigger token is uniformly sampled from $\mathcal{V} \setminus \{k\}$. Also, we have $P(z_t = k \mid y = k, t_q = s, z_{t_q} = q) = 1/(V - 1)$ because $z_t$ for $t \ne T$ is neither the trigger token nor the output token, and thus, sampled from $\mathcal{V} \setminus \{q\}$. These give us the following when $s \le t - 2$:

$$\mathbb{E}[\mathbf{1}\{z_t = k\} \mid y = k, t_q = s] = \begin{cases} \frac{1}{V-1} & \text{if } t \ne T, \\ 0 & \text{otherwise}. \end{cases} \quad (34)$$

Combining Eq. 32 and 34, we obtain the result. $\square$

**Lemma 8.** *Given a sequence $z_{1:T}$ that satisfies Sec. E.2. For any vocabulary $k \in \mathcal{V}$, we have*

$$\mathbb{E}\left[ \frac{1}{T} \left( \sum_{t'=1}^{T} \mathbf{1}\{z_{t'} = k\} \right) \mathbf{1}\{z_t = k\} \mid y = k, t_q = t - 1 \right] = \begin{cases} \frac{2}{T} & \text{if } t = T, \\ \frac{1}{T}\left(1 + \frac{T-3}{V-1}\right) & \text{otherwise}. \end{cases} \quad (35)$$

*Proof.* By the linearity of expectation, we can decompose the expectation as follows:

$$\mathbb{E}\left[\left(\sum_{t'=1}^{T}\mathbf{1}\{z_{t'}=k\}\right)\mathbf{1}\{z_t=k\}\mid\cdot\right] = \mathbb{E}[\mathbf{1}\{z_{t_q}=k\}\mathbf{1}\{z_t=k\}\mid\cdot]$$
$$+\mathbb{E}[\mathbf{1}\{z_{t_q+1}=k\}\mathbf{1}\{z_t=k\}\mid\cdot]$$
$$+\mathbf{1}\{t\neq T\}\mathbb{E}[\mathbf{1}\{z_T=k\}\mathbf{1}\{z_t=k\}\mid\cdot]$$
$$+\mathbb{E}\left[\left(\sum_{t'\neq t_q,t_q+1,T}\mathbf{1}\{z_{t'}=k\}\right)\mathbf{1}\{z_t=k\}\mid\cdot\right],$$

where we abbreviate the conditions of the expectations to enhance readability.
Note that we always have $\mathbf{1}\{z_t=k\}=1$ when $t_q=t-1$ and $y=k$. In the case of $t=T$, the trigger token appears at positions $T-1$ and $T$ given $t_q=t-1$. With the condition $y=k$, it follows that

$$\mathbf{1}\{z_{t_q}=k\}\mathbf{1}\{z_t=k\}=1,$$
$$\mathbf{1}\{z_{t_q+1}=k\}\mathbf{1}\{z_t=k\}=1.$$

In the above, we point out that we have $t_q+1=T$ in this situation. Additionally, we get

$$\mathbb{E}\left[\left(\sum_{t'\neq t_q,t_q+1,T}\mathbf{1}\{z_{t'}=k\}\right)\mathbf{1}\{z_t=k\}\mid\cdot\right]=0,$$

where the final equation follows from the fact that $z_{t'}$ is not a trigger token $k$.

For the case where $t\neq T$, the trigger token is not $k$ because otherwise there will be more than two triggers in the input sequence at positions $t_q, t_q+1$ and $T$. Thus, we have

$$\mathbb{E}[\mathbf{1}\{z_{t_q}=k\}\mathbf{1}\{z_t=k\}\mid\cdot]=0,$$
$$\mathbb{E}[\mathbf{1}\{z_{t_q+1}=k\}\mathbf{1}\{z_t=k\}\mid\cdot]=1,$$
$$\mathbb{E}[\mathbf{1}\{z_T=k\}\mathbf{1}\{z_t=k\}\mid\cdot]=0.$$

Following the same argument in Eq. 33, we also have

$$\mathbb{E}\left[\left(\sum_{t'\neq t_q,t_q+1,T}\mathbf{1}\{z_{t'}=k\}\right)\mathbf{1}\{z_t=k\}\mid\cdot\right]=\mathbb{E}\left[\sum_{t'\neq t_q,t_q+1,T}\mathbf{1}\{z_{t'}=k\}\mid\cdot\right]$$
$$=\frac{T-3}{V-1}.$$

We prove the statement by combining these results. □

**Lemma 9.** *Given a sequence $z_{1:T}$ that satisfies Sec. E.2. Fix some $t\in[1,T]\subset\mathbb{N}$. For any $s\in[1,t-2]\cup\{t\}$, and vocabulary $k\in\mathcal{V}$,*

$$\mathbb{E}\left[\left(\sum_{t'=1}^{T}\mathbf{1}\{z_{t'}=k\}\right)\mathbf{1}\{z_t=k\}\mid y=k,t_q=s\right] = \begin{cases} 2 & \text{if } s=t \text{ and } t=T-1, \\ \frac{2}{V-1}+\frac{T-4}{(V-1)^2} & \text{if } s\leq t-2 \text{ and } t\neq T, \\ 0 & \text{otherwise.} \end{cases}$$
(36)

*Proof.* First, we look at the scenario $s=t$. Since $t_q=s=t$, the input sequence $z_{1:T}$ will contain more than two trigger tokens unless $t=T-1$, which does not satisfy the condition of input sequence, we examine the case of $t=T-1$. This condition ensures that $\mathbf{1}\{z_{T-1}=k\}=\mathbf{1}\{z_T=k\}=1$, and therefore, we can use the instance $t=T$ from Lemma 8.

$$\mathbb{E}\left[\left(\sum_{t'=1}^{T}\mathbf{1}\{z_{t'}=k\}\right)\mathbf{1}\{z_t=k\}\mid\cdot\right]=2.$$

Next, we require $s \leq t - 2$. It is impossible to have $t = T$ because this violates the input sequence condition, where the trigger token only appears twice. Now, by the linearity of expectation, we can decompose the expectation as follows:

$$\mathbb{E}\left[\left(\sum_{t'=1}^{T} \mathbf{1}\{z_{t'} = k\}\right) \mathbf{1}\{z_t = k\} \mid \cdot\right] = \mathbb{E}[\mathbf{1}\{z_{t_q} = k\}\mathbf{1}\{z_t = k\} \mid \cdot]$$

$$+ \mathbb{E}[\mathbf{1}\{z_{t_q+1} = k\}\mathbf{1}\{z_t = k\} \mid \cdot]$$

$$+ \mathbb{E}[\mathbf{1}\{z_t = k\}\mathbf{1}\{z_t = k\} \mid \cdot]$$

$$+ \mathbb{E}[\mathbf{1}\{z_T = k\}\mathbf{1}\{z_t = k\} \mid \cdot]$$

$$+ \mathbb{E}\left[\left(\sum_{t' \neq t_q, t_q+1, t, T} \mathbf{1}\{z_{t'} = k\}\right) \mathbf{1}\{z_t = k\} \mid \cdot\right].$$

Since $t_q = s \leq t - 2$ and $t \neq T$, the token $z_t$ is neither trigger nor output. This yields the following:

$$\mathbb{E}[\mathbf{1}\{z_{t_q} = k\}\mathbf{1}\{z_t = k\} \mid \cdot] = 0,$$
$$\mathbb{E}[\mathbf{1}\{z_T = k\}\mathbf{1}\{z_t = k\} \mid \cdot] = 0.$$

Given $y$ equals $k$, we have $\mathbf{1}\{z_{t_q+1} = k\} = 1$. Therefore, from Lemma 7, we get

$$\mathbb{E}[\mathbf{1}\{z_{t_q+1} = k\}\mathbf{1}\{z_t = k\} \mid \cdot] = \mathbb{E}[\mathbf{1}\{z_t = k\} \mid \cdot]$$

$$= \frac{1}{V-1},$$

$$\mathbb{E}[\mathbf{1}\{z_t = k\}\mathbf{1}\{z_t = k\} \mid \cdot] = \mathbb{E}[\mathbf{1}\{z_t = k\} \mid \cdot]$$

$$= \frac{1}{V-1}.$$

Finally, for $t' \neq t_q, t_q + 1, t$ and $T$, the tokens $z_{t'}$ and $z_t$ are independent. With Lemma 7, we obtain

$$\mathbb{E}\left[\left(\sum_{t' \neq t_q, t_q+1, t, T} \mathbf{1}\{z_{t'} = k\}\right) \mathbf{1}\{z_t = k\} \mid \cdot\right] = \frac{T-4}{(V-1)^2}.$$

Integrating the equations above allows us to arrive at the conclusion. $\qquad\square$

**Lemma 10.** *Given a sequence $z_{1:T}$ that satisfies Sec. E.2. For any vocabulary $k \in \mathcal{V}$, and $t \in [1, T] \subset \mathbb{N}$, we have*

$$\mathbb{E}\left[\mathbf{1}\{z_t = k\}\mathbf{1}\{t_q \leq t\}x_{t_q} \mid y = k\right] = P(t_q = t - 1)\mathbf{1}\{t = T\}w_E(k)$$

$$+ P(t_q = t)\mathbf{1}\{t = T - 1\}w_E(k) + O\left(\frac{1}{V}\right).$$

*Proof.* We split $\mathbf{1}\{t_q \leq t\}$ into $\mathbf{1}\{t_q = t - 1\} + \mathbf{1}\{t_q = t\} + \mathbf{1}\{t_q \leq t - 2\}$. Then, the expectation can be decomposed as:

$$\mathbb{E}\left[\mathbf{1}\{z_t = k\}\mathbf{1}\{t_q \leq t\}x_{t_q} \mid y = k\right] = \mathbb{E}\left[\mathbf{1}\{z_t = k\}\mathbf{1}\{t_q = t - 1\}x_{t_q} \mid y = k\right]$$

$$+ \mathbb{E}\left[\mathbf{1}\{z_t = k\}\mathbf{1}\{t_q = t\}x_{t_q} \mid y = k\right]$$

$$+ \mathbb{E}\left[\mathbf{1}\{z_t = k\}\mathbf{1}\{t_q \leq t - 2\}x_{t_q} \mid y = k\right].$$

Note that when $t_q = t - 1$, $z_t$ is the output token. In addition, if we have $t = T$, $z_t$ works as the second output token as well, which means that $z_t = k$ and $x_{t_q} = w_E(k)$. If we find $t \neq T$, the trigger token is not $k$. Thus, we derive

$$\mathbb{E}\left[\mathbf{1}\{z_t = k\}\mathbf{1}\{t_q = t - 1\}x_{t_q} \mid y = k\right] = P(t_q = t - 1)\mathbb{E}\left[\mathbf{1}\{z_t = k\}x_{t_q} \mid y = k, t_q = t - 1\right]$$

$$= P(t_q = t - 1)\mathbf{1}\{t = T\}w_E(k)$$
$$+ P(t_q = t - 1)\mathbf{1}\{t \neq T\} \sum_{j \in \mathcal{V} \setminus \{k\}} \frac{w_E(j)}{V - 1}.$$

Next, we discuss $t_q = t$. Since $t_q$ is the first trigger position, we always observe that $t_q \leq T - 1$. If two trigger tokens are adjacent to each other, i.e., $t_q = T - 1$, we can state that $z_{T-1} = z_T = k$ because of the equation $y = k$. Otherwise, the trigger token must belong to $\mathcal{V} \setminus \{k\}$, and $\mathbf{1}\{z_t = k\} = 0$. This gives

$$\mathbb{E}\left[\mathbf{1}\{z_t = k\}\mathbf{1}\{t_q = t\}x_{t_q} \mid y = k\right] = P(t_q = t - 1)\mathbf{1}\{t = T\}w_E(k).$$

For $t_q \leq t - 2$, we again analyze the cases of $t = T$ and $t \neq T$. Provided that $t = T$, $\mathbf{1}\{z_t = k\} = 1$ implies that the trigger token is decided to be $k$, while the output token is also $k$. This leads to more than two occurrences of trigger token, which violates the input sequence condition. Now, we assume $t \leq T - 1$. This time, the trigger token $z_{t_q}$ is not $k$, and $z_t$ is not a trigger token, which yields

$$\mathbb{E}\left[\mathbf{1}\{z_t = k\}\mathbf{1}\{t_q \leq t - 2\}x_{t_q} \mid y = k\right]$$
$$= P(t_q \leq t - 2)\mathbb{E}[w_E(z_{t_q}) \mid y = k, t_q \leq t - 2]\mathbb{E}[\mathbf{1}\{z_t = k\} \mid z_{t_q}, y = k, t_q \leq t - 2]$$
$$= P(t_q \leq t - 2) \sum_{q \in \mathcal{V} \setminus \{k\}} \frac{w_E(q)}{V - 1}\mathbb{E}[\mathbf{1}\{z_t = k\} \mid z_{t_q} = q, y = k, t_q \leq t - 2]$$
$$= P(t_q \leq t - 2) \sum_{q \in \mathcal{V} \setminus \{k\}} \frac{w_E(q)}{(V - 1)^2}.$$

This concludes the proof. $\qquad\square$

**Lemma 11.** *Given a sequence $z_{1:T}$ that satisfies Sec. E.2. For any vocabulary $k \in \mathcal{V}$, and $t \in [1, T] \subset \mathbb{N}$, we have*

$$\mathbb{E}\left[\mathbf{1}\{z_t = k\}\mathbf{1}\{t_q \leq t\}\bar{x}_{1:t} \mid y = k\right] = \frac{P(t_q = t - 1)}{t}(1 + \mathbf{1}\{t = T\})w_E(k)$$
$$+ \frac{\mathbf{1}\{t = T - 1\}P(t_q = t)}{t}w_E(k) + O\left(\frac{1}{V}\right). \tag{37}$$

*Proof.* As in the proof of Lemma 10, we analyze each scenario depending on the value of $t_q$. Firstly, let us consider $t_q = t - 1$. This leads to the fact that $z_t$ is the output token. Given $y = k$, we have $z_t = k$.

$$\mathbb{E}\left[\mathbf{1}\{z_t = k\}\mathbf{1}\{t_q = t - 1\}\bar{x}_{1:t} \mid y = k\right] = P(t_q = t - 1)\mathbb{E}\left[\bar{x}_{1:t} \mid y = k, t_q = t - 1\right].$$

In the event that $t = T$ is valid, $x_{t_q} = x_T = w_E(k)$ holds and the other $T - 2$ tokens are not $k$. Thus, it gives

$$\mathbb{E}\left[\bar{x}_{1:T} \mid y = k\right] = \frac{1}{T}\left\{2w_E(k) + (T - 2) \sum_{j \in \mathcal{V} \setminus \{k\}} \frac{w_E(j)}{V - 1}\right\}.$$

When $t \neq T$ holds, we can say that $x_{t_q} \neq w_E(k)$, while $x_{t_q+1} = w_E(k)$ because of $y = k$. This gives us

$$\mathbb{E}[x_{t_q} \mid y = k, t_q = t - 1] = \sum_{q \in \mathcal{V} \setminus \{k\}} \frac{w_E(j)}{V - 1},$$
$$\mathbb{E}[x_{t_q+1} \mid y = k, t_q = t - 1] = w_E(k).$$

Also, we can derive the following equations for $i \in [1, t] \setminus \{t_q, t_q + 1\}$:

$$\mathbb{E}[x_i \mid y = k, t_q = t - 1] = \sum_{j \in \mathcal{V}} P(z_i = j \mid \cdot)w_E(j)$$

$$= \sum_{q=1}^{V} \sum_{j=1}^{V} P(z_i = j, z_{t_q} = q \mid \cdot) w_E(j)$$

$$= \sum_{q \in \mathcal{V}\backslash\{k\}} \sum_{j \in \mathcal{V}\backslash\{q\}} P(z_i = j, z_{t_q} = q \mid \cdot) w_E(j)$$

$$= \sum_{\substack{q \in \mathcal{V}\backslash\{k\} \\ j \in \mathcal{V}\backslash\{q\}}} P(z_i = j, \mid z_{t_q} = q, \cdot) P(z_{t_q} = q \mid \cdot) w_E(j)$$

$$= \sum_{q \in \mathcal{V}\backslash\{k\}} \sum_{j \in \mathcal{V}\backslash\{q\}} \frac{w_E(j)}{(V-1)^2}.$$

In the above, we use the fact that $P(x_i = j, x_{t_q} = q) = 0$ if $j = q$ or $q = k$ holds. To summarize, we have

$$
\mathbb{E}\left[\bar{x}_{1:t} \mid y = k, t_q = t-1\right] = \begin{cases} \frac{2w_E(k)}{t} + \frac{t-2}{t} \sum_{j\in\mathcal{V}\backslash\{k\}} \frac{w_E(j)}{V-1} & \text{if } t = T, \\ \frac{1}{t}\left\{ w_E(k) + \sum_{q\in\mathcal{V}\backslash\{k\}} \frac{w_E(q)}{V-1} \right\} \\ \quad + \frac{t-2}{t}\left\{ \sum_{\substack{q\in\mathcal{V}\backslash\{k\} \\ j\in\mathcal{V}\backslash\{q\}}} \frac{w_E(j)}{(V-1)^2} \right\} & \text{otherwise.} \end{cases}
$$

(38)

$$
= \begin{cases} \frac{2}{T} w_E(k) + O\left(\frac{1}{V}\right) & \text{if } t = T, \\ \frac{1}{t} w_E(k) + O\left(\frac{1}{V}\right) & \text{otherwise.} \end{cases}
$$

Next, assume $t_q = t$. In this case, $t = T$ implies $t_q = T$ which is a contradiction. Furthermore, $t_q = t < T - 1$ implies $t_q < t_{q+1} < T$. Under this condition, we cannot have $z_t = k$, because it produces three appearances of the trigger token. Hence, it suffices to consider $t = T - 1$. If $t_q = t = T - 1$ is true, we can state that $x_t = k$, leading to the formulation below:

$$\mathbb{E}[\mathbf{1}\{z_t = k\}\bar{x}_{1:t} \mid y = k, t_q = t] = \mathbb{E}[\bar{x}_{1:t} \mid y = k, t_q = t]$$

$$= \frac{1}{t}\left\{ w_E(k) + (T-2) \sum_{\substack{1 \le j \le V \\ j \ne k}} \frac{w_E(j)}{V-1} \right\}$$

$$= \frac{1}{T-1} w_E(k) + O\left(\frac{1}{V}\right),$$

where the second equation follows from the same argument in the instance $t = T$ in Eq. 38, but this time $x_T$ is not contained in $x_{1:t}$.

Lastly, we focus on $t_q = s \le t - 2$. We argue that the trigger token $z_{t_q} = z_T$ is not $k$, due to the condition of the number of trigger token in the input sequence. This means that $t \ne T$ and $z_t$ does not serve as a trigger token. Now, we split $\bar{x}_{1:t}$ into $\frac{1}{t}(x_{t_q} + x_{t_q+1} + x_t + \sum_{t'} x_{t'})$ and obtain

$$\mathbb{E}[\mathbf{1}\{z_t = k\}x_{t_q} \mid y = k, t_q = s] = \sum_{q\in\mathcal{V}\backslash\{k\}} P(z_t = k, z_{t_q} = q \mid \cdot) w_E(q)$$

$$= \sum_{q\in\mathcal{V}\backslash\{k\}} \frac{w_E(q)}{(V-1)^2},$$

$$\mathbb{E}[\mathbf{1}\{z_t = k\}x_{t_q+1} \mid y = k, t_q = s] = P(z_t = k \mid \cdot) w_E(k)$$

$$= \sum_{q\in\mathcal{V}\backslash\{k\}} P(z_t = k, z_{t_q} = q \mid \cdot) w_E(k)$$

$$= \sum_{q\in\mathcal{V}\backslash\{k\}} \frac{w_E(k)}{(V-1)^2},$$

$$= \frac{w_E(k)}{V-1},$$

$$\mathbb{E}[\mathbf{1}\{z_t = k\}x_t \mid y = k, t_q = s] = P(z_t = k \mid y = k, t_q = s)w_E(k)$$

$$= \frac{w_E(k)}{V-1},$$

$$\mathbb{E}[\mathbf{1}\{z_t = k\}x_{t'} \mid y = k, t_q = s] = \sum_{\substack{q \in \mathcal{V} \setminus \{k\} \\ j \in \mathcal{V} \setminus \{q\}}} P(z_t = k, z_{t'} = j, z_{t_q} = q \mid \cdot)w_E(j)$$

$$= \sum_{q \in \mathcal{V} \setminus \{k\}} P(z_{t_q} = q \mid \cdot)$$

$$\times \sum_{j \in \mathcal{V} \setminus \{q\}} P(z_t = k, z_{t'} = j \mid z_{t_q} = q, \cdot)w_E(j)$$

$$= \sum_{q \in \mathcal{V} \setminus \{k\}} \frac{1}{V-1} \sum_{j \in \mathcal{V} \setminus \{q\}} \frac{w_E(j)}{(V-1)^2}.$$

In short, we have

$$\mathbb{E}[\mathbf{1}\{z_t = k\}\bar{x}_{1:t} \mid y = k, t_q = s] = O\left(\frac{1}{V}\right).$$

Summing up all the related terms leads to Eq. 37, which finishes the proof. □

**Lemma 12.** *Given a sequence $z_{1:T}$ that satisfies Sec. E.2. For any vocabulary $k \in \mathcal{V}$, and $t \in [1, T] \subset \mathbb{N}$, we have*

$$\mathbb{E}\left[\frac{1}{T}\sum_{t'=1}^{T} \mathbf{1}\{z_{t'} = k\}\mathbf{1}\{t_q \leq t\}x_{t_q}\mathbf{1}\{z_t = k\} \mid y = k\right] = \frac{2w_E(k)}{T}\mathbf{1}\{t = T\}P(t_q = t-1)$$

$$+ \frac{2w_E(k)}{T}\mathbf{1}\{t = T-1\}P(t_q = t)$$

$$+ \frac{1}{T} \cdot O\left(\frac{1}{V}\right). \tag{39}$$

*Proof.* We consider the following cases:

**Case 1:** $t_q = t - 1$.
If $t = T$ holds, this gives $z_{t_q} = z_T = y = k$ and $\mathbf{1}\{z_{t-1} = k\} = \mathbf{1}\{z_t = k\} = 1$. Thus, we have

$$\mathbb{E}\left[\mathbf{1}\{z_{t-1} = k\}x_{t_q}\mathbf{1}\{z_t = k\} \mid y = k, t_q = t-1\right] = \mathbb{E}\left[x_{t_q} \mid y = k, t_q = t-1\right]$$

$$= w_E(k),$$

$$\mathbb{E}\left[\mathbf{1}\{z_t = k\}x_{t_q}\mathbf{1}\{z_t = k\} \mid y = k, t_q = t-1\right] = \mathbb{E}\left[x_{t_q} \mid y = k, t_q = t-1\right]$$

$$= w_E(k).$$

Since $z_{t'}$ for $t' \neq t_q, t_q + 1$ is not a trigger token, it follows that $\mathbf{1}\{z_{t'} = k\} = 0$, and we have

$$\mathbb{E}\left[\sum_{t' \neq t-1, t} \mathbf{1}\{z_{t'} = k\}x_{t_q}\mathbf{1}\{z_t = k\} \mid \cdot\right] = 0.$$

Otherwise, when $t \neq T$, $z_{t_q} = z_T$ is not $k$, yielding

$$\mathbb{E}\left[\mathbf{1}\{z_{t-1} = k\}x_{t_q}\mathbf{1}\{z_t = k\} \mid \cdot\right] = 0.$$

$$\mathbb{E}\left[\mathbf{1}\{z_T = k\}x_{t_q}\mathbf{1}\{z_t = k\} \mid \cdot\right] = 0.$$

Also, $z_{t_q}$ is sampled from $\mathcal{V} \setminus \{k\}$, we get

$$\mathbb{E}\left[\mathbf{1}\{z_t = k\}x_{t_q}\mathbf{1}\{z_t = k\} \mid y = k, t_q = t-1\right] = \mathbb{E}\left[x_{t_q} \mid y = k, t_q = t-1\right]$$

$$= \sum_{j \in \mathcal{V} \setminus \{k\}} \frac{w_E(j)}{V - 1}.$$

For the rest of $t'$, we note that $z_{t_q} \neq k$ and obtain

$$\mathbb{E} \left[ \sum_{t' \neq t, t-1, T} \mathbf{1}\{z_{t'} = k\} x_{t_q} \mathbf{1}\{z_t = k\} \mid \cdot \right] = \sum_{\substack{t' \\ q \in \mathcal{V} \setminus \{k\}}} P(z_{t'} = k, z_{t_q} = q \mid \cdot) w_E(q)$$

$$= (T - 3) \sum_{q \in \mathcal{V} \setminus \{k\}} \frac{w_E(q)}{(V - 1)^2}. \tag{40}$$

In summary, we find that

$$\mathbb{E} \left[ \sum_{t'=1}^{T} \mathbf{1}\{z_{t'} = k\} \mathbf{1}\{t_q = t - 1\} x_{t_q} \mathbf{1}\{z_t = k\} \mid y = k \right]$$

$$= P(t_q = t - 1)\mathbf{1}\{t = T\}2w_E(k) + P(t_q = t - 1)\mathbf{1}\{t \neq T\}O\left(\frac{1}{V}\right). \tag{41}$$

**Case 2:** $t_q = t$.
For this scenario, $\mathbf{1}\{z_t = k\} = 1$ holds only when $t_q = t = T - 1$. Thus, we can state that

$$\mathbb{E}[\mathbf{1}\{z_{t_q} = k\} x_{t_q} \mathbf{1}\{z_t = k\} \mid y = k, t_q = t] = \mathbb{E}[x_{t_q} \mid y = k, t_q = t]$$
$$= w_E(k),$$
$$\mathbb{E}[\mathbf{1}\{z_T = k\} x_{t_q} \mathbf{1}\{z_t = k\} \mid y = k, t_q = t] = \mathbb{E}[x_{t_q} \mid y = k, t_q = t]$$
$$= w_E(k).$$

For the other $t'$, we have $\mathbf{1}\{z_{t'} = k\} = 0$ because these tokens are not a trigger token. This means that

$$\mathbb{E} \left[ \sum_{t' \neq t, T} \mathbf{1}\{z_{t'} = k\} x_{t_q} \mathbf{1}\{z_t = k\} \mid y = k, t_q = t \right] = 0.$$

In short, we have

$$\mathbb{E} \left[ \frac{1}{T} \sum_{t'=1}^{T} \mathbf{1}\{z_{t'} = k\} \mathbf{1}\{t_q = t\} x_{t_q} \mathbf{1}\{z_t = k\} \mid y = k \right] = \frac{\mathbf{1}\{t = T - 1\} P(t_q = t)}{T} 2w_E(k). \tag{42}$$

**Case 3:** $t_q \leq t - 2$.
We first let $t = T$. If $\mathbf{1}\{z_t = k\} = 1$ holds, then the trigger token becomes $k$, but we find that $z_{t_q} = z_{t_q+1} = z_T = k$, meaning the trigger token appears more than two times. Therefore, we only consider $t \neq T$. Given $z_{t_q}(= z_T) = k$, it also fails to satisfy the required conditions for the sequence, hence it follows that

$$\mathbb{E}[\mathbf{1}\{z_s = k\} x_{t_q} \mathbf{1}\{z_t = k\} \mid y = k, t_q = s] = 0,$$
$$\mathbb{E}[\mathbf{1}\{z_T = k\} x_{t_q} \mathbf{1}\{z_t = k\} \mid y = k, t_q = s] = 0,$$

where $1 \leq s \leq t - 2$.
Similarly to Eq. 40, we can derive that

$$\mathbb{E}[\mathbf{1}\{z_t = k\} x_{t_q} \mathbf{1}\{z_t = k\} \mid y = k, t_q = s] = \mathbb{E}[x_{t_q} \mathbf{1}\{z_t = k\} \mid y = k, t_q = s]$$

$$= \sum_{q \in \mathcal{V} \setminus \{k\}} \frac{w_E(q)}{(V - 1)^2},$$

$$\mathbb{E}[\mathbf{1}\{z_{s+1} = k\} x_{t_q} \mathbf{1}\{z_t = k\} \mid y = k, t_q = s] = \mathbb{E}[x_{t_q} \mathbf{1}\{z_t = k\} \mid y = k, t_q = s]$$

$$= \sum_{q \in \mathcal{V} \setminus \{k\}} \frac{w_E(q)}{(V-1)^2}.$$

In addition, we have

$$\mathbb{E} \left[ \sum_{\substack{t' \neq s \\ t' \neq s+1 \\ t' \neq t \\ t' \neq T}} \mathbf{1}\{z_{t'} = k\} x_{t_q} \mathbf{1}\{z_t = k\} \mid y = k, t_q = s \right]$$

$$= \sum_{\substack{t' \\ q \in \mathcal{V} \setminus \{k\}}} P(z_{t'} = k, z_t = k, z_{t_q} = q \mid \cdot) w_E(q)$$

$$= \sum_{\substack{t' \\ q \in \mathcal{V} \setminus \{k\}}} P(z_{t'} = k, z_t = k, \mid z_{t_q} = q, \cdot) P(z_{t_q} = q \mid y = k, t_q = s) w_E(q)$$

$$= (T - 4) \sum_{q \in \mathcal{V} \setminus \{k\}} \frac{w_E(q)}{(V-1)^3}.$$

These equations give us that for $1 \leq s \leq t - 2$,

$$\mathbb{E} \left[ \sum_{t'=1}^{T} \mathbf{1}\{z_{t'} = k\} \mathbf{1}\{t_q = s\} x_{t_q} \mathbf{1}\{z_t = k\} \mid y = k \right] = O\left(\frac{1}{V^2}\right). \tag{43}$$

Gathering all the equations of $1 \leq t_q \leq t$ leads to Eq. 39 and we have proven the statement.
$\square$

**Lemma 13.** *Given a sequence $z_{1:T}$ that satisfies Sec. E.2. For any vocabulary $k \in \mathcal{V}$, and $t \in [1, T] \subset \mathbb{N}$, we have*

$$\mathbb{E} \left[ \frac{1}{T} \sum_{t'=1}^{T} \mathbf{1}\{z_{t'} = k\} \mathbf{1}\{t_q \leq t\} \bar{x}_{1:t} \mathbf{1}\{z_t = k\} \mid y = k \right]$$

$$= \frac{4 \cdot \mathbf{1}\{t = T\} + \mathbf{1}\{t \neq T\}}{Tt} P(t_q = t - 1) w_E(k) + \frac{2 \cdot \mathbf{1}\{t = T - 1\} P(t_q = t)}{Tt} w_E(k)$$

$$+ O\left(\frac{1}{VT}\right). \tag{44}$$

*Proof.* Considering the linearity of expectation, we focus on each term $x_i$ contained in $\bar{x}_{1:t}$. We consider the following cases:

**Case 1:** $t_q = t - 1$.
We already have the result as to $x_{t_q}$ in Eq. 41:

$$\mathbb{E} \left[ \frac{1}{T} \sum_{t'=1}^{T} \mathbf{1}\{z_{t'} = k\} \mathbf{1}\{t_q = t - 1\} x_{t_q} \mathbf{1}\{z_t = k\} \mid y = k \right]$$

$$= \frac{P(t_q = t - 1)}{T} \mathbf{1}\{t = T\} 2 w_E(k) + \frac{P(t_q = t - 1)}{T} \mathbf{1}\{t \neq T\} O\left(\frac{1}{V}\right).$$

We can also use Lemma 8 for $x_t = k$, i.e.,

$$\mathbb{E} \left[ \frac{1}{T} \sum_{t'=1}^{T} \mathbf{1}\{z_{t'} = k\} x_t \mathbf{1}\{z_t = k\} \mid y = k \right]$$

$$= \mathbb{E} \left[ \frac{1}{T} \sum_{t'=1}^{T} \mathbf{1}\{z_{t'} = k\} \mid y = k, t_q = t - 1 \right] P(t_q = t - 1) w_E(k)$$

$$= \begin{cases} \frac{P(t_q=t-1)}{T} 2 w_E(k) & \text{if } t = T, \\ \frac{P(t_q=t-1)}{T} \left(1 + \frac{T-3}{V-1}\right) w_E(k) & \text{otherwise.} \end{cases}$$

Now we deal with $x_i$ for $1 \le i \le t-2$. Note that $z_i \ne k$ for these $i$ under the conditions $t = T$ and $y = k$.

$$\mathbb{E}\left[\sum_{t'=1}^{T} \mathbf{1}\{z_{t'} = k\} x_i \mathbf{1}\{z_t = k\} \mid y = k, t_q = t-1\right]$$

$$= \mathbb{E}\left[\mathbf{1}\{z_t = k\} x_i \mathbf{1}\{z_t = k\} \mid \cdot\right]$$

$$+ \mathbb{E}\left[\mathbf{1}\{z_{t_q} = k\} x_i \mathbf{1}\{z_t = k\} \mid \cdot\right]$$

$$+ \mathbb{E}\left[\mathbf{1}\{z_i = k\} x_i \mathbf{1}\{z_t = k\} \mid \cdot\right]$$

$$+ \sum_{t'} \mathbb{E}\left[\mathbf{1}\{z_i = k\} x_i \mathbf{1}\{z_t = k\} \mid \cdot\right]$$

$$= \begin{cases} \sum_{j \in \mathcal{V} \setminus \{k\}} \frac{w_E(j)}{V-1} + \sum_{j \in \mathcal{V} \setminus \{k\}} \frac{w_E(j)}{V-1} & \text{if } t = T, \\ \sum_{j \in \mathcal{V} \setminus \{k\}} \frac{w_E(j)}{V-1} + \frac{w_E(k)}{V-1} + (T-3) \sum_{\substack{q \in \mathcal{V} \setminus \{k\} \\ j \in \mathcal{V} \setminus \{q\}}} \frac{w_E(j)}{(V-1)^3} & \text{otherwise} \end{cases}$$

$$= O\left(\frac{1}{V}\right)$$

**Case 2:** $t_q = t$.
For this scenario, we have seen that $t_q = t = T - 1$ is the only possible way for the expectation to take non-zero value. For $x_{t_q}$, we can observe Eq. 42 and obtain

$$\mathbb{E}\left[\frac{1}{T} \sum_{t'=1}^{T} \mathbf{1}\{z_{t'} = k\} \mathbf{1}\{t_q = t\} x_{t_q} \mathbf{1}\{z_t = k\} \mid y = k\right] = \frac{\mathbf{1}\{t = T-1\} P(t_q = t)}{T} 2 w_E(k).$$

The other terms related to $x_i$ for $1 \le i \le t-1$ can be calculated

$$\mathbb{E}\left[\frac{1}{T} \sum_{t'=1}^{T} \mathbf{1}\{z_{t'} = k\} x_i \mathbf{1}\{z_t = k\} \mid y = k, t_q = t\right]$$

$$= \frac{1}{T} \mathbb{E}\left[\mathbf{1}\{z_t = k\} x_i \mathbf{1}\{z_t = k\} \mid \cdot\right]$$

$$+ \frac{1}{T} \mathbb{E}\left[\mathbf{1}\{z_T = k\} x_i \mathbf{1}\{z_t = k\} \mid \cdot\right]$$

$$+ \frac{1}{T} \sum_{t' \ne t} \mathbb{E}\left[\mathbf{1}\{z_{t'} = k\} x_i \mathbf{1}\{z_t = k\} \mid \cdot\right].$$

$$= \frac{1}{T} \left(\sum_{j \in \mathcal{V} \setminus \{k\}} \frac{w_E(j)}{V-1} + \sum_{j \in \mathcal{V} \setminus \{k\}} \frac{w_E(j)}{V-1} + 0\right)$$

$$= \frac{1}{T} \cdot O\left(\frac{1}{V}\right).$$

We note that $\mathbf{1}\{z_t = k\} = 1$ and $\mathbf{1}\{z_{t'} = k\} = 0$ for $t' < t_q = t$ because of the input sequence condition.

**Case 3:** $t_q = s \le t-2$.
Again, we have the result for $x_{t_q}$ in Eq. 43.

$$\mathbb{E}\left[\frac{1}{T} \sum_{t'=1}^{T} \mathbf{1}\{z_{t'} = k\} \mathbf{1}\{t_q = s\} x_{t_q} \mathbf{1}\{z_t = k\} \mid y = k\right] = O\left(\frac{1}{V^2}\right).$$

We have $x_{t_q+1} = k$ under $y = k$, and Lemma 9 tells us that

$$\mathbb{E}\left[\frac{1}{T}\sum_{t'=1}^{T}\mathbf{1}\{z_{t'} = k\}\mathbf{1}\{t_q = s\}x_{t_q+1}\mathbf{1}\{z_t = k\} \mid y = k\right]$$

$$= \mathbb{E}\left[\sum_{t'=1}^{T}\mathbf{1}\{z_{t'} = k\}\mathbf{1}\{z_t = k\} \mid y = k, t_q = s\right]\frac{P(t_q = s)}{T}w_E(k)$$

$$= \begin{cases} \frac{P(t_q=s)}{T}\left(\frac{2}{V-1} + \frac{T-4}{(V-1)^2}\right)w_E(k) & \text{if } t \neq T, \\ 0 & \text{otherwise.} \end{cases}$$

Next, we consider $x_t$. Note that $t = T$ forces $x_{t_q}$ to be $w_E(k)$ for the expectation to take non-zero value, which contradicts the input sequence condition. Hence we assume $t \neq T$ so that $z_{t_q} = z_T \neq k$.

$$\mathbb{E}\left[\frac{1}{T}\sum_{t'=1}^{T}\mathbf{1}\{z_{t'} = k\}x_t\mathbf{1}\{z_t = k\} \mid y = k, t_q = s\right] = \frac{1}{T}\left\{\mathbb{E}\left[\mathbf{1}\{z_{t_q} = k\}x_t\mathbf{1}\{z_t = k\} \mid \cdot\right]\right.$$

$$+ \mathbb{E}\left[\mathbf{1}\{z_T = k\}x_t\mathbf{1}\{z_t = k\} \mid \cdot\right]$$

$$+ \mathbb{E}\left[\mathbf{1}\{z_{t_q+1} = k\}x_t\mathbf{1}\{z_t = k\} \mid \cdot\right]$$

$$+ \mathbb{E}\left[\mathbf{1}\{z_t = k\}x_t\mathbf{1}\{z_t = k\} \mid \cdot\right]$$

$$\left.+ \sum_{t'}\mathbb{E}\left[\mathbf{1}\{z_i = k\}x_t\mathbf{1}\{z_t = k\} \mid \cdot\right]\right\}$$

$$= \frac{1}{T}\left(\frac{2w_E(k)}{V-1} + \sum_{t'}\frac{w_E(k)}{(V-1)^2}\right)$$

$$= \frac{1}{T} \cdot O\left(\frac{1}{V}\right).$$

Lastly, regarding $x_i$ for $i \neq t, t_q, t_q + 1$, we also consider $t \neq T$ and it follows that

$$\mathbb{E}\left[\frac{1}{T}\sum_{t'=1}^{T}\mathbf{1}\{z_{t'} = k\}x_i\mathbf{1}\{z_t = k\} \mid y = k, t_q = s\right]$$

$$= \frac{1}{T}\left\{\mathbb{E}\left[\mathbf{1}\{z_{t_q} = k\}x_i\mathbf{1}\{z_t = k\} \mid \cdot\right]\right.$$

$$+ \mathbb{E}\left[\mathbf{1}\{z_T = k\}x_i\mathbf{1}\{z_t = k\} \mid \cdot\right]$$

$$+ \mathbb{E}\left[\mathbf{1}\{z_{t_q+1} = k\}x_i\mathbf{1}\{z_t = k\} \mid \cdot\right]$$

$$+ \mathbb{E}\left[\mathbf{1}\{z_t = k\}x_i\mathbf{1}\{z_t = k\} \mid \cdot\right]$$

$$\left.+ \sum_{t'}\mathbb{E}\left[\mathbf{1}\{z_i = k\}x_i\mathbf{1}\{z_t = k\} \mid \cdot\right]\right\}$$

$$= \frac{1}{T}\left\{0 + 0 + 2\sum_{q\in\mathcal{V}\setminus\{k\}}\frac{1}{V-1}\sum_{j\in\mathcal{V}\setminus\{q\}}\frac{w_E(j)}{(V-1)^2} + \sum_{t'}\sum_{q\in\mathcal{V}\setminus\{k\}}\frac{1}{V-1}\sum_{j\in\mathcal{V}\setminus\{q\}}\frac{w_E(j)}{(V-1)^3}\right\}$$

$$= \frac{1}{T} \cdot O\left(\frac{1}{V}\right).$$

Thus, we obtain Eq. 44. $\qquad\qquad\qquad\qquad\qquad\qquad\qquad\qquad\qquad\qquad\qquad\qquad\qquad\square$

**Lemma 14.** *Given a sequence $z_{1:T}$ that satisfies Sec. E.2. Then, we have*

$$\mathbb{E}\left[\mathbf{1}\{z_T = k\}q_{T,T}\mathbf{1}\{z_T = v\} \mid y = k\right] = \begin{cases} P(t_q = T - 1)(r_0 - \bar{r}_{1-T:0}) \\ \quad + P(t_q = T - 1)\frac{T-2}{T}w_E(k) + O\left(\frac{1}{V}\right) & \text{if } v = k, \\ 0 & \text{otherwise,} \end{cases}$$

where $q_{T,T} = (w_E(z_T) + r_0) - (\bar{w}_E(z_{1:T}) + \bar{r}_{1-T:0})$.

*Proof.* The inequality $v \neq k$ produces $\mathbf{1}\{z_T = k\}\mathbf{1}\{z_T = v\} = 0$, and thus, we focus on $v = k$. Since $z_T$ is the second trigger token, we have $z_{t_q} = z_T = k$. This is achievable only when $t_q = T - 1$.

$$
\begin{aligned}
&\mathbb{E}\left[\mathbf{1}\{z_T = k\}q_{T,T}\mathbf{1}\{z_T = v\} \mid y = k\right] \\
&= \mathbb{E}\left[\mathbf{1}\{z_T = k\}q_{T,T} \mid y = k\right] \\
&= P(z_T = k \mid y = k)\mathbb{E}\left[q_{T,T} \mid y = k, z_T = k\right] \qquad (45) \\
&= \sum_{i=1}^{T} P(z_T = k, t_q = i \mid y = k)\mathbb{E}\left[q_{T,T} \mid y = k, z_T = k\right] \\
&= P(t_q = T - 1 \mid y = k)P(z_T = k \mid y = k, t_q = T - 1)\mathbb{E}\left[q_{T,T} \mid y = k, z_T = k\right] \\
&= P(t_q = T - 1)\left\{w_E(k) + r_0 - \bar{r}_{1-T:0} - \frac{1}{T}\left(2w_E(k) + \sum_{\substack{1 \leq j \leq V \\ j \neq k}}\frac{(T-2)}{V-1}w_E(j)\right)\right\} \\
&= P(t_q = T - 1)\left(r_0 + \frac{T-2}{T}w_E(k) - \bar{r}_{1-T:0}\right) \\
&\quad + O\left(\frac{1}{V}\right).
\end{aligned}
$$

$\square$

**Lemma 15.** *Given a sequence $z_{1:T}$ that satisfies Sec. E.2. Then, we have*

$$
\mathbb{E}\left[\frac{1}{T}\sum_{t'=1}^{T}\mathbf{1}\{z_{t'} = k\}q_{T,T}\mathbf{1}\{z_T = v\} \mid y = k\right] = \begin{cases} \frac{2P(t_q=T-1)}{T}\{r_0 - \bar{r}_{1-T:0}\} \\ \quad + \frac{2P(t_q=T-1)}{T}\frac{T-2}{T}w_E(k) \\ \quad + \frac{1}{T} \cdot O\left(\frac{1}{V}\right) & \text{if } v = k, \\ \frac{1}{T}O\left(\frac{1}{V}\right) & \text{otherwise.} \end{cases}
$$

*where $q_{T,T} = (w_E(z_T) + r_0) - (\bar{w}_E(z_{1:T}) + \bar{r}_{1-T:0})$.*

*Proof.* We first consider the case $v = k$. If the second trigger token $z_T = v = k$ and the output token is also $k$, it is impossible to have $t_q < T - 1$ because it produces more than two trigger tokens in the input sequence. This means that we have, for $t' \in [T - 2]$, that

$$
\mathbb{E}\left[\mathbf{1}\{z_t' = k\}\mathbf{1}\{z_T = k\} \mid y = k\right] = 0.
$$

We can use Lemma 14 for $t' = T$:

$$
\mathbb{E}\left[\mathbf{1}\{z_T = k\}q_{T,T}\mathbf{1}\{z_T = v\} \mid y = k\right] = P(t_q = T - 1)\left(r_0 + \frac{T-2}{T}w_E(k) - \bar{r}_{1-T:0}\right) \\
+ O\left(\frac{1}{V}\right).
$$

Finally, for $t' = T - 1$, we note that the two conditions $y = k$ and $z_T = k$ imply $z_{T-1} = k$.

$$
\begin{aligned}
&\mathbb{E}\left[\mathbf{1}\{z_{T-1} = k\}q_{T,T}\mathbf{1}\{z_T = k\} \mid y = k\right] \\
&= P(z_T = k \mid y = k)\mathbb{E}\left[\mathbf{1}\{z_{T-1} = k\}q_{T,T} \mid y = k, z_T = k\right] \\
&= P(z_T = k \mid y = k)\mathbb{E}\left[q_{T,T} \mid y = k, z_T = k\right] \\
&= \mathbb{E}\left[\mathbf{1}\{z_T = k\}q_{T,T}\mathbf{1}\{z_T = v\} \mid y = k\right] \\
&= P(t_q = T - 1)\left(r_0 + \frac{T-2}{T}w_E(k) - \bar{r}_{1-T:0}\right) + O\left(\frac{1}{V}\right),
\end{aligned}
$$

where the second-to-last expression follows from Eq. 45.
Now, we look at the case $v \neq k$. Different from $v = k$, we have

$$\mathbb{E}\left[\mathbf{1}\{z_{T-1} = k\}q_{T,T}\mathbf{1}\{z_T = v\} \mid y = k\right] = 0,$$
$$\mathbb{E}\left[\mathbf{1}\{z_T = k\}q_{T,T}\mathbf{1}\{z_T = v\} \mid y = k\right] = 0.$$

Furthermore, the trigger token is uniformly distributed, we have

$$\mathbb{E}[\mathbf{1}\{z_T = v\} \mid y = k] = P(z_T = v \mid y = k)$$
$$= \sum_{i=1}^{T} P(z_T = v, t_q = i \mid y = k)$$
$$= \sum_{i=1}^{T-2} P(z_T = v \mid y = k, t_q = i)P(t_q = i \mid y = k)$$
$$= \frac{P(t_q \leq T - 2)}{V - 1}$$
$$= O\left(\frac{1}{V}\right).$$

This leads to the following expression for $t' \in [T - 2]$

$$\mathbb{E}\left[\mathbf{1}\{z_{t'} = k\}q_{T,T}\mathbf{1}\{z_T = v\} \mid y = k\right] = \mathbb{E}\left[\mathbf{1}\{z_T = v\}\mathbb{E}\left[\mathbf{1}\{z_{t'} = k\}q_{T,T} \mid y = k, z_T\right] \mid y = k\right]$$
$$= P(z_T = v \mid y = k)\mathbb{E}\left[\mathbf{1}\{z_{t'} = k\}q_{T,T} \mid y = k, z_T = v\right]$$
$$= O\left(\frac{1}{V}\right).$$

This concludes the proof. □

### F.2  global v.s. in-context

**Proposition 4 (Restated).** *Suppose a two-layer transformer is an associative memory transformer. Given a length-$T$ sequence $z_{1:T}$ where the last token is $z_T = q$, let the $t_1$-th token be $z_{t_1} = v_1$ following $z_{t_1-1} = q$, and let the $t_2$-th token be $z_{t_2} = v_2$ following $z_{t_2-1} = q$, where $v_1 \neq v_2$ and $(3 \leq)t_1 < t_2$ without loss of generality. Also assume that there exist only one $v_1$ and $v_2$ in the context $z_{1:T}$. Then, the logit $\xi_{v_1}$ and $\xi_{v_2}$ can be expressed as follows:*

$$\xi_{v_1} = \sigma\left(\frac{e}{t_1 + e - 1}\right) + \log \pi_b(v_1 \mid q),$$
$$\xi_{v_2} = \sigma\left(\frac{1+e}{t_2 + e - 1}\right) + \log \pi_b(v_2 \mid q),$$

*where $\sigma(\cdot)$ represents the softmax function.*

*Proof.* We consider the output of the first layer of attention for $z_t$. Since we use the associative memory transformer, we substitute $W_Q^1 = I$ and $W_K^1 = \sum_{k \in Q} w_E(k)r_{-1}^\top$ into Eq. 1, we have

$$x_t^{(1)} = \Phi_1 w_E(z_{1:t})\sigma\left((w_E(z_{1:t}) + R_{1-t:0})^\top \left(\sum_{k \in Q} r_{-1}w_E(k)^\top\right) I w_E(z_t)\right) + w_E(z_t)$$

$$= \Phi_1 w_E(z_{1:t})\sigma\left(\begin{pmatrix}(w_E(z_1) + r_{1-t})^\top \\ (w_E(z_2) + r_{2-t})^\top \\ \vdots \\ (w_E(z_{t-1}) + r_{-1})^\top \\ (w_E(z_t) + r_0)^\top\end{pmatrix} r_{-1}w_E(z_t)^\top w_E(z_t)\right) + w_E(z_t)$$

$$= \Phi_1 w_E(z_{1:t}) \sigma \left( \begin{pmatrix} 0 \\ 0 \\ \vdots \\ 1 \\ 0 \end{pmatrix} \right) + w_E(z_t) \tag{46}$$

$$= w_E(z_t) + \frac{e(\Phi_1 w_E(z_{t-1}))}{t+e-1} + \sum_{\substack{1 \leq i \leq t \\ i \neq t-1}} \frac{\Phi_1 w_E(z_i)}{t+e-1}, \tag{47}$$

where $e$ is Euler's number. In the transformation above, Eq. 46 follows from the near-orthogonality among the relative positional encodings and the embedding vectors Next, we calculate the output of the second layer attention block. We first consider softmax values and then attention head $W_O^2 W_V^2$. Similarly to the first attention layer, we use Eq. 47 and ideal matrices $W_Q^2 = I$ and $W_K^2 = \sum_{k \in Q} w_E(k)(\Phi_1 w_E(k))^\top$ to calculate softmax. For any $t$, we can express $(x_{1:t}^{(1)})^\top (W_K^2)^\top W_Q^2 x_t^{(1)}$ as

$$\begin{pmatrix} (w_E(z_1) + \Phi_1 w_E(z_1))^\top \\ \left( w_E(z_2) + \frac{e(\Phi_1 w_E(z_1))}{e+1} + \sum \frac{\Phi_1 w_E(z_i)}{e+1} \right)^\top \\ \vdots \\ \left( w_E(z_{t-1}) + \frac{e(\Phi_1 w_E(z_{t-2}))}{t+e-2} + \sum \frac{\Phi_1 w_E(z_i)}{t+e-2} \right)^\top \\ \left( w_E(z_t) + \frac{e(\Phi_1 w_E(z_{t-1}))}{t+e-1} + \sum \frac{\Phi_1 w_E(z_i)}{t+e-1} \right)^\top \end{pmatrix} \left( \sum_{k \in Q} (\Phi_1 w_E(k)) w_E(k)^\top \right) I x_t^{(1)},$$

where the summation in the $t$-th row is given by $\sum_{\substack{1 \leq i \leq t \\ i \neq t-1}}$. Since $x_T^{(1)} = w_E(q) + \frac{e(\Phi_1 w_E(z_{T-1}))}{T+e-1} + \sum_{\substack{1 \leq i \leq T \\ i \neq T-1}} \frac{\Phi_1 w_E(z_i)}{T+e-1}$, and thanks to the near-orthogonality, we have

$$\sigma \left( (x_{1:T}^{(1)})^\top (W_K^2)^\top W_Q^2 x_T^{(1)} \right) = \sigma(p),$$

where each entry of $p \in \mathbb{R}^T$ is

$$p_t = \begin{cases} 0 & \text{if } t < t_1 - 1, \\ 1/(t_1 + e - 2) & \text{if } t = t_1 - 1, \\ e/(t_1 + e - 1) & \text{if } t = t_1, \\ 1/(i + e - 1) & \text{if } t_1 < t < t_2 - 1, \\ 2/(t_2 + e - 2) & \text{if } t = t_2 - 1, \\ (1 + e)/(t_2 + e - 1) & \text{if } t = t_2, \\ 2/(i + e - 1) & \text{if } t_2 < t < T, \\ 3/(T + e - 1) & \text{if } t = T. \end{cases}$$

Back to Eq. 2, we can now write $x_T^{(2)}$ with $W_O^2 = \sum_{v=1}^{V} w_U(v)(W_V^2 w_E(v))^\top$

$$x_T^{(2)} = \sum_{t=1}^{T} W_O^2 W_V^2 \sigma(p_t) x_t^{(1)} + x_T^{(1)}$$

$$= \sum_{t=1}^{T} \sigma(p_t) \sum_{v=1}^{V} w_U(v)(W_V^2 w_E(v))^\top$$

$$\times W_V^2 \left( w_E(z_t) + \frac{e(\Phi_1 w_E(z_{t-1}))}{t+e-1} + \sum_{\substack{1 \leq i \leq t \\ i \neq t-1}} \frac{\Phi_1 w_E(z_i)}{t+e-1} \right) + x_T^{(1)}$$

$$= \sum_{t=1}^{T} \sigma(p_t) w_U(z_t) + w_E(q) + \frac{e(\Phi_1 w_E(z_{T-1}))}{T + e - 1} + \sum_{\substack{1 \leq i \leq t \\ i \neq T-1}} \frac{\Phi_1 w_E(z_i)}{T + e - 1}$$

Finally, the feed-forward network in the second layer is applied to $x_T^{(2)}$, and we obtain the following:

$$x_T = \text{MLP}(x_T^{(2)}) + x_T^{(2)}$$

$$= W_2 \left[ \text{ReLU} \left( \begin{pmatrix} w_E(v_1)^\top \\ w_E(v_2)^\top \\ \vdots \\ w_E(v_V)^\top \end{pmatrix} \left( w_E(q) + \sum_{t=1}^{T} \sigma(p_t) w_U(z_t) + C \right) \right) \right] + x_T^{(2)}$$

$$= W_2 \left[ \text{ReLU} \begin{pmatrix} 0 \\ \vdots \\ 0 \\ 1 \\ 0 \\ \vdots \\ 0 \end{pmatrix} \right] + x_T^{(2)}$$

$$= \left( \sum_{u=1}^{V} \log \pi_b(u \mid v_1) w_U(u) \quad \cdots \quad \sum_{u=1}^{V} \log \pi_b(u \mid v_V) w_U(u) \right) \begin{pmatrix} 0 \\ \vdots \\ 0 \\ 1 \\ 0 \\ \vdots \\ 0 \end{pmatrix} + x_T^{(2)}$$

$$= \sum_{u=1}^{V} \log \pi_b(u \mid q) w_U(u) + x_T^{(2)}$$

$$= \sum_{u=1}^{V} \log \pi_b(u \mid q) w_U(u) + \sum_{t=1}^{T} \sigma(p_t) w_U(z_t) + w_E(q) + C,$$

where $C = \frac{e(\Phi_1 w_E(z_{T-1}))}{T + e - 1} + \sum_{\substack{1 \leq i \leq t \\ i \neq T-1}} \frac{\Phi_1 w_E(z_i)}{T + e - 1}$.

We can compute the logits $\xi_v$ for $v \in \mathcal{V}$ after the unembedding layer. Since the row vectors of $W_U$ are also near-orthogonal to each other, it is sufficient to look at the coefficients of $w_U(v_1)$ and $w_U(v_2)$. Since $v_1$ and $v_2$ appear only once at $t = t_1$ and $t = t_2$ respectively in the context $z_{1:T}$, the logits are calculated as follows:

$$\xi_{v_1} = \sigma \left( \frac{e}{t_1 + e - 1} \right) + \mathbf{1}\{v_1 \in U_q\} \log \pi_b(v_1 \mid q),$$

$$\xi_{v_2} = \sigma \left( \frac{1 + e}{t_2 + e - 1} \right) + \mathbf{1}\{v_2 \in U_q\} \log \pi_b(v_2 \mid q).$$

$\square$

**Proposition 5** (**Restated**). *Suppose a two-layer transformer is a stronger associative memory transformer as in Def. 3. Given a length-T sequence $z_{1:T}$ where the last token is $z_T = q$, let $f(v)$ be the number of token pattern "qv" appearing in the sequence for vocabulary $v \in \mathcal{V}$. For large enough $\tau_1$ and $\tau_2$, the logits $\xi_v$ can be expressed as follows:*

$$\xi_v \underset{\substack{\tau_1 \\ \tau_2}}{\approx} \log \pi_b(v \mid q) + \tau_3 \cdot \frac{f(v) + \mathbf{1}\{v = q\}\mathbf{1}\{z_1 = q\}}{\left( \sum_{v'=1}^{V} f(v') \right) + \mathbf{1}\{z_1 = q\}}. \tag{48}$$

*Proof.* The proof proceeds in the same manner as in Prop. 2. With the associative memory $\hat{W}_K^1$, the attention values for $z_t$ is expressed as

$$x_t^{(1)} = \Phi_1 w_E(z_{1:t}) \sigma((w_E(z_{1:t}) + R_{1-t:0})^\top \left( \tau_1 \sum_{k \in Q} r_{-1} w_E(k)^\top \right) I w_E(z_t)) + w_E(z_t)$$

$$= \Phi_1 w_E(z_{1:t}) \sigma \left( \tau_1 \begin{pmatrix} (w_E(z_1) + r_{1-t})^\top \\ (w_E(z_2) + r_{2-t})^\top \\ \vdots \\ (w_E(z_{t-1}) + r_{-1})^\top \\ (w_E(z_t) + r_0)^\top \end{pmatrix} r_{-1} w_E(z_t)^\top w_E(z_t) \right) + w_E(z_t)$$

$$= \Phi_1 w_E(z_{1:t}) \sigma \left( \begin{pmatrix} 0 \\ 0 \\ \vdots \\ \tau_1 \\ 0 \end{pmatrix} \right) + w_E(z_t)$$

$$\underset{\tau_1}{\approx} \Phi_1 w_E(z_{t-1}) + w_E(z_t).$$

Now the second layer attention has the form

$$\sigma \left( (x_{1:t}^{(1)})^\top (\hat{W}_K^2)^\top W_Q^2 x_t^{(1)} \right)$$

$$\underset{\tau_1}{\approx} \sigma \left( \begin{pmatrix} (w_E(z_1) + \Phi_1 w_E(z_1))^\top \\ (w_E(z_2) + \Phi_1 w_E(z_1))^\top \\ \vdots \\ (w_E(z_{t-1}) + \Phi_1 w_E(z_{t-2}))^\top \\ (w_E(z_t) + \Phi_1 w_E(z_{t-1}))^\top \end{pmatrix} \left( \tau_2 \sum_{k \in Q} (\Phi_1 w_E(k)) w_E(k)^\top \right) I x_t^{(1)} \right).$$

We note that $x_T^{(1)} = w_E(q) + \Phi_1 w_E(z_{T-1})$, and consider the case $t = T$ in the above equation. Then, we will have

$$\sigma \left( (x_{1:T}^{(1)})^\top (\hat{W}_K^2)^\top W_Q^2 x_T^{(1)} \right)$$

$$\underset{\tau_1}{\approx} \sigma \left( \begin{pmatrix} (w_E(z_1) + \Phi_1 w_E(z_1))^\top \\ (w_E(z_2) + \Phi_1 w_E(z_1))^\top \\ \vdots \\ (w_E(z_{T-1}) + \Phi_1 w_E(z_{T-2}))^\top \\ (w_E(z_T) + \Phi_1 w_E(z_{T-1}))^\top \end{pmatrix} \left( \tau_2 \sum_{k \in Q} (\Phi_1 w_E(k)) w_E(k)^\top \right) I x_T^{(1)} \right)$$

$$= \sigma \left( \tau_2 \begin{pmatrix} (w_E(z_1) + \Phi_1 w_E(z_1))^\top \\ (w_E(z_2) + \Phi_1 w_E(z_1))^\top \\ \vdots \\ (w_E(a) + \Phi_1 w_E(q))^\top \\ \vdots \\ (w_E(b) + \Phi_1 w_E(q))^\top \\ \vdots \\ (w_E(z_T) + \Phi_1 w_E(z_{T-1}))^\top \end{pmatrix} \Phi_1 w_E(q) \right)$$

$$= \sigma \left( \sum_{i \in I_q \cup I_1} \tau_2 \mathbf{e}_i \right),$$

where $I_q = \{t \mid x_t^{(1)} = w_E(v') + \Phi_1 w_E(q), \exists v' \in \mathcal{V}\}$ is the index set of $x_t^{(1)}$ containing $\Phi_1 w_E(q)$, and $I_1 = \{1\}$ if $z_1 = q$ and $I_1 = \emptyset$ otherwise. Also, $\mathbf{e}_i$ denote the unit vector where only the $i$-th component is 1, and all other components are 0. Note that $|I_q \cup I_1| = \left(\sum_{v=1}^{V} f(v)\right) + \mathbf{1}\{z_1 = q\}$.

Going back to the output of the whole attention block including $\hat{W}_O^2$ and $W_V^2$, we have from Eq. 2 that

$$x_T^{(2)} \underset{\tau_1}{\approx} \sum_{t=1}^{T} \hat{W}_O^2 W_V^2 \sigma\Big(\sum_{i \in I_q \cup I_1} \tau_2 \mathbf{e}_i\Big)_t x_t^{(1)} + x_T^{(1)}$$

$$\underset{\substack{\tau_1 \\ \tau_2}}{\approx} \frac{1}{|I_q \cup I_1|} \sum_{i \in I_q \cup I_1} \hat{W}_O^2 W_V^2 x_i^{(1)} + x_T^{(1)}$$

$$= \frac{1}{|I_q \cup I_1|} \sum_{v=1}^{V} \tau_3 w_U(v)(W_V^2 w_E(v))^\top \left\{ f(v) \left( W_V^2 (w_E(v) + \Phi_1 w_E(q)) \right) \right\}$$

$$+ \mathbf{1}\{z_1 = q\} \left( W_V^2(w_E(q) + \Phi_1 w_E(q)) \right) + x_T^{(1)}$$

$$= \sum_{v=1}^{V} \frac{f(v) + \mathbf{1}\{v = q\}\mathbf{1}\{z_1 = q\}}{\left(\sum_{v'=1}^{V} f(v')\right) + \mathbf{1}\{z_1 = q\}} \cdot \tau_3 w_U(v) + w_E(q) + \Phi_1 w_E(z_{T-1})$$

$$:= \sum_{v=1}^{V} g(v) \cdot w_U(v) + w_E(q) + \Phi_1 w_E(z_{T-1}).$$

Lastly, through the feed-forward network, we obtain the outcome.

$$x_T$$
$$= \text{MLP}(x_T^{(2)}) + x_T^{(2)}$$

$$\underset{\substack{\tau_1 \\ \tau_2}}{\approx} \left( \sum_{u=1}^{V} \log \pi_b(u \mid v_1) w_U(u) \quad \dots \quad \sum_{u=1}^{V} \log \pi_b(u \mid v_V) w_U(u) \right) \begin{pmatrix} 0 \\ \vdots \\ 0 \\ 1 \\ 0 \\ \vdots \\ 0 \end{pmatrix} + x_T^{(2)}$$

$$= \sum_{u=1}^{V} \log \pi_b(u \mid q) w_U(u) + x_T^{(2)}$$

$$= \sum_{u=1}^{V} \log \pi_b(u \mid q) w_U(u) + \left( \sum_{v=1}^{V} g(v) w_U(v) \right) + w_E(q) + \Phi_1 w_E(z_{T-1})$$

$$= \sum_{v=1}^{V} \left( \mathbf{1}\{v \in U_q\} \log \pi_b(v \mid q) + \tau_3 \cdot \frac{f(v) + \mathbf{1}\{v = q\}\mathbf{1}\{z_1 = q\}}{\left(\sum_{v'=1}^{V} f(v')\right) + \mathbf{1}\{z_1 = q\}} \right) \cdot w_U(v)$$

$$+ w_E(q) + \Phi_1 w_E(z_{T-1}).$$

The proof concludes by examining the coefficients of $w_U(v)$. $\qquad \square$

### F.3 Induction head without positional encoding

In this section, we provide the construction of a three-layer transformer with no positional encoding that achieves the induction head mechanism.

We first introduce the following theorem(Kazemnejad et al., 2024) stating that transformer can implement APE using one transformer block.

**Theorem 4** (Theorem 1, (Kazemnejad et al., 2024)). *Let $z_{1:T+1}$ be an input sequence of length $T + 1$, where $z_1 = \langle bos \rangle$. Then, there exist a transformer block consisting of an attention block and feed-froward network that recovers absolute positions $[1, 2, \ldots T + 1]$ and writes in the hidden state $H^{(1)}$.*

In the proof, they prepare extra 3 dimensions for word embeddings to effectively reconstruct the positional information:

$$W_E = \begin{pmatrix} 1 & 1 & 1 & \cdots & 1 \\ 1 & 0 & 0 & \cdots & 0 \\ 0 & 0 & 0 & \cdots & 0 \\ a^e_{1,1} & a^e_{1,2} & a^e_{1,3} & \cdots & a^e_{1,V} \\ \vdots & \vdots & \vdots & \ddots & \vdots \\ a^e_{d,1} & a^e_{d,2} & a^e_{d,3} & \cdots & a^e_{d,V} \end{pmatrix},$$

where $\langle bos \rangle$ is represented by the first column without loss of generality.

Theorem 4 states that the hidden state $H^{(1)}$ has the following form after the appropriate transformer block:

$$H^{(1)} = \begin{pmatrix} 1 & 1 & 1 & \cdots & 1 \\ 1 & 0 & 0 & \cdots & 0 \\ 1 & 2 & 3 & \cdots & T+1 \\ a_{1,1} & a_{1,2} & a_{1,3} & \cdots & a_{1,T+1} \\ \vdots & \vdots & \vdots & \ddots & \vdots \\ a_{d,1} & a_{d,2} & a_{d,3} & \cdots & a_{d,T+1} \end{pmatrix},$$

Now we show that a three-layer transformer with no positional encoding can implement induction head mechanism.

**Proposition 6** (Restated). *There exists construction of a three-layer transformer without positional encoding that achieves induction head mechanism.*

*Proof.* The first layer of the transformer block is given by the construction in Theorem 4, which gives us the hidden state:

$$H^{(1)} = \begin{pmatrix} 1 & 1 & 1 & \cdots & 1 \\ 1 & 0 & 0 & \cdots & 0 \\ 1 & 2 & 3 & \cdots & T+1 \\ a_{1,1} & a_{1,2} & a_{1,3} & \cdots & a_{1,T+1} \\ \vdots & \vdots & \vdots & \ddots & \vdots \\ a_{d,1} & a_{d,2} & a_{d,3} & \cdots & a_{d,T+1} \end{pmatrix},$$

where each entry denoted by $a_{*,*}$ is a Gaussian random value to guarantee near-orthogonality.

Then, for the second transformer block, we use the following matrix weights.

$$W^2_Q = \begin{pmatrix} 1 & 0 & 0 & \cdots & 0 \\ 0 & 0 & 0 & \cdots & 0 \\ \vdots & \vdots & \vdots & \ddots & \vdots \\ 0 & 0 & 0 & \cdots & 0 \end{pmatrix},$$

$$W^2_K = \begin{pmatrix} 0 & 0 & C & \cdots & 0 \\ 0 & 0 & 0 & \cdots & 0 \\ \vdots & \vdots & \vdots & \ddots & \vdots \\ 0 & 0 & 0 & \cdots & 0 \end{pmatrix},$$

$$W_V^2 = \begin{pmatrix} 0 & 0 & 0 & 0 & \cdots & 0 \\ 0 & 0 & 0 & 0 & \cdots & 0 \\ 0 & 0 & 0 & 0 & \cdots & 0 \\ 0 & 0 & 0 & w_{1,1}^v & \cdots & w_{1,d}^v \\ \vdots & \vdots & \vdots & \vdots & \ddots & \vdots \\ 0 & 0 & 0 & w_{d,1}^v & \cdots & w_{d,d}^v \end{pmatrix},$$

$$W_O^2 = \begin{pmatrix} 0 & 0 & 0 & 0 & \cdots & 0 \\ 0 & 0 & 0 & 0 & \cdots & 0 \\ 0 & 0 & 0 & 0 & \cdots & 0 \\ 0 & 0 & 0 & w_{1,1}^o & \cdots & w_{1,d}^o \\ \vdots & \vdots & \vdots & \vdots & \ddots & \vdots \\ 0 & 0 & 0 & w_{d,1}^o & \cdots & w_{d,d}^o \end{pmatrix},$$

where $C \in \mathbb{R}_+$ is a constant value, and $w_{*,*}^*$ denotes the Gaussian entry to guarantee the near-orthogonality. With these matrix, the query and key vectors are represented by

$$q_t = (1, 0, \ldots, 0)^\top,$$
$$k_i = (C \cdot i, 0, \ldots 0)^\top.$$

Thus, the block outputs

$$\sum_{i=1}^{t-1} \sigma(\langle k_i, q_t \rangle) W_O^2 W_V^2 h_i^{(1)} \approx W_O^2 W_V^2 h_{t-1}^{(1)}, \tag{49}$$

where the approximation holds for sufficiently large $C$ and $h_{t-1}^{(1)}$ is the $t-1$-th column of $H^{(1)}$. Here, we assume that the attention block uses strict causal attention, which attends only to positions in $\{1, 2, \ldots, t-1\}$ for the token at position $t$.

Eq. 49 implies that the attention block attends only to the previous token, thus forming a previous token head. With the residual connection, the hidden state $H^{(2)}$ is represented as:

$$H^{(2)} = \begin{pmatrix} 1 & 1 & 1 & \cdots & 1 \\ 1 & 0 & 0 & \cdots & 0 \\ 1 & 2 & 3 & \cdots & T+1 \\ a_{1,1} & a_{1,2} & a_{1,3} & \cdots & a_{1,T+1} \\ \vdots & \vdots & \vdots & \ddots & \vdots \\ a_{d,1} & a_{d,2} & a_{d,3} & \cdots & a_{d,T+1} \end{pmatrix} + \begin{pmatrix} 0 & 0 & 0 & \cdots & 0 \\ 0 & 0 & 0 & \cdots & 0 \\ 0 & 0 & 0 & \cdots & 0 \\ \mathbf{0} & \Phi_2 \mathbf{a}_1 & \Phi_2 \mathbf{a}_2 & \cdots & \Phi_2 \mathbf{a}_T \end{pmatrix},$$

where $\mathbf{a}_i = (a_{1,i}, \ldots, a_{d,i})^\top$ and $\Phi_2$ is the bottom right sub-matrix from $(W_O^2 W_V^2)_{4:d+3,4:d+3}$.

Finally, in the third transformer block, we set the following attention matrix weights:

$$W_Q^3 = \begin{pmatrix} 0 & 0 & 0 & 0 & 0 & \cdots & 0 \\ 0 & 0 & 0 & 0 & 0 & \cdots & 0 \\ 0 & 0 & 0 & 0 & 0 & \cdots & 0 \\ 0 & 0 & 0 & 1 & 0 & \cdots & 0 \\ 0 & 0 & 0 & 0 & 1 & \cdots & 0 \\ \vdots & \vdots & \vdots & \vdots & \vdots & \ddots & \vdots \\ 0 & 0 & 0 & 0 & 0 & \cdots & 1 \end{pmatrix},$$

$$W_K^3 = \begin{pmatrix} 0 & 0 & 0 & 0 & \cdots & 0 \\ 0 & 0 & 0 & 0 & \cdots & 0 \\ 0 & 0 & 0 & 0 & \cdots & 0 \\ 0 & 0 & 0 & w_{1,1}^{k'} & \cdots & w_{1,d}^{k'} \\ \vdots & \vdots & \vdots & \vdots & \ddots & \vdots \\ 0 & 0 & 0 & w_{d,1}^{k'} & \cdots & w_{d,d}^{k'} \end{pmatrix},$$

$$W_V^3 = \begin{pmatrix} 0 & 0 & 0 & 0 & \cdots & 0 \\ 0 & 0 & 0 & 0 & \cdots & 0 \\ 0 & 0 & 0 & 0 & \cdots & 0 \\ 0 & 0 & 0 & w_{1,1}^{v'} & \cdots & w_{1,d}^{v'} \\ \vdots & \vdots & \vdots & \vdots & \ddots & \vdots \\ 0 & 0 & 0 & w_{d,1}^{v'} & \cdots & w_{d,d}^{v'} \end{pmatrix},$$

$$W_O^3 = \begin{pmatrix} 0 & 0 & 0 & 0 & \cdots & 0 \\ 0 & 0 & 0 & 0 & \cdots & 0 \\ 0 & 0 & 0 & 0 & \cdots & 0 \\ 0 & 0 & 0 & w_{1,1}^{o'} & \cdots & w_{1,d}^{o'} \\ \vdots & \vdots & \vdots & \vdots & \ddots & \vdots \\ 0 & 0 & 0 & w_{d,1}^{o'} & \cdots & w_{d,d}^{o'} \end{pmatrix},$$

where $w_{*,*}^{v'}$ denotes the Gaussian entry to guarantee the near-orthogonality, and the sub-matrices in the bottom right of $W_K^3$ and $W_O^3$ are constructed the same way as $W_K^2$ and $W_O^2$ in Lemma 1. The result of this calculation is the same as in Lemma 1 except for the first three rows. Thus, the induction head mechanism is implemented in the three-layer transformer with no positional encoding. □

# G Experimental setup

## G.1 Example of Bigram Sequence Generation with Trigger-Output Pairs

To illustrate the sequence generation procedure in our modified bigram language model, we provide a concrete example with a small vocabulary V = A, B, C, D. The generation proceeds in discrete steps as follows:

**Step 0**: Sample trigger-output pair

A new trigger-output pair $(q_k, o_k)$ is sampled from $\pi_q$ and $\pi_o$. For example, suppose $(q_1, o_1) = (A, C)$.

**Step 1**: Sample the first token

The first token $z_1$ is sampled from a predefined distribution over $V$, e.g., uniformly. Suppose $z_1 = B$.

**Step 2**: Trigger check and transition

If $z_1 = q_1$, then set $z_2 := o_1$. Otherwise, sample $z_2 \sim \pi_b(\cdot \mid z_1)$. Since $z_1 \neq q_1$, we sample $z_2 = A$.

**Step 3**: Trigger condition applies

Since $z_2 = q_1 = A$, we set $z_3 := o_1 = C$.

**Step 4**: Normal bigram transition

Since $z_3 \neq q_1$, sample $z_4 \sim \pi_b(\cdot \mid z_3)$ and suppose $z_4 = D$. These steps result in the sequence $BACD$.

**Step 5**: Re-sample trigger-output pair

Return to Step 0 and sample a new $(q_k, o_k)$ from $\pi_q$ and $\pi_o$ to generate a new sequence.

## G.2 Neglect of in-context knowledge

We conducted our experiments using the following setup, carefully designed to evaluate the performance of a transformer model with APE and RPE.

**Sequence generation**   Following Bietti et al. (2024), we define $\pi_u$ and $\pi_b$ as unigram and bigram character-level distributions estimated from the Tiny Shakespeare dataset (Karpathy, 2015). We sample triggers from $\pi_q = \pi_u$, and corresponding outputs $o_k$ are also sampled uniformly for each sequence generation. The vocabulary size $|\mathcal{V}|$ is 65.

**Data Arguments**   The model was trained on sequences with length of **256 tokens** for the previous token head experiment and **128 tokens** for the generalization experiment. The number of trigger tokens was set to **5**, and we did not fix trigger tokens, i.e., all trigger tokens were sampled uniformly.

**Model Configuration**   The transformer model used for the experiments had the following configuration:

- **Model dimension**: **256**
- **Vocabulary size**: **65**
- Input sequences were capped at a maximum length of **256 tokens**.
- The embedding layer $W_E$ and unembedding layer $W_U$ were **frozen** during training.
- The model did not use $W_F$ in the second layer to focus on the role of induction head.
- The model has 0.5M parameters.

**Optimization Setup**   The optimization process was conducted using the following hyper-parameters:

- **Optimizer**: Stochastic Gradient Descent (SGD) with momentum
- **Batch size**: **512**
- **Learning rate**: **0.2**
- **Momentum**: **0.9**
- **Weight decay**: **0.0001**

**Training Strategy**

- The model was trained for a total of **1000 iterations**, which takes 0.5 hours of A100 GPU time.
- Loss computation was restricted to output tokens.

### G.3   Global vs in-context

We conducted our experiments using the following setup for the evaluation of performance of a transformer model with RPE trained on The Google analogy dataset.

**Data Arguments**   The model was trained on sequences with a maximum length of **256 tokens**.

Since The Google analogy dataset offers analogy pairs, such as (Tokyo, Japan), the bigram conditionals will mostly be in the form like $\pi_b(\text{Japan} \mid \text{Tokyo}) = 1$. We constructed new bigram conditionals by adding some randomenss.

We first sample **10 fake target words**, such as USA, China for each source word. Then, the conditionals will be calculated as follows:

1. count the number $c_A(B)$ of analogical pair $(A, B)$.
2. sample $p_A \in [0.01, 0.1]$ for each source word $A$.
3. consider the analogical pair $(A, B_i)$ appear $p_r c_A(B)$ times, where $B_i$ is the fake target words for source word $A$.

4. calculate the bigram conditionals based on the number of appearance of analogical pairs.

Everytime we generate an input sequence, we sample **5** trigger tokens from the set of all source words, and uniformly selects the corresponding output tokens from all vocabulary. we start from a source word that is sampled uniformly from the set of source words. Then, we transition with either the bigram conditionals or to the output token if the current token is the trigger token. Then we place comma, and sample uniformly a new word from the set of source words for the next token, and repeat this process until the maximum length is achieved.

**Model Configuration**    The transformer model used for the experiments had the following architecture and configuration:

- **Model dimension**: **512**
- **Vocabulary size**: **233**
- Input sequences were capped at a maximum length of **256 tokens**.
- The embedding and output layers were **frozen** during training.
- The model has 2M parameters.

**Optimization Setup**    The optimization process was conducted using the following hyperparameters:

- **Optimizer**: Stochastic Gradient Descent (SGD) with momentum
- **Batch size**: **512**
- **Learning rate**: **0.1**
- **Momentum**: **0.9**
- **Weight decay**: **0.0001**

**Training Strategy**

- The model was trained for a total of **100,000 iterations**, which required 9 hours of A100 GPU time.
- Loss computation was applied to all tokens except for commas.

### G.4   About datasets

The Tiny Shakespeare dataset is licensed under the Apache License, Version 2.0, and we can freely use the content as long as we comply with the terms of the license. Similarly, the Google Analogy dataset is licensed under CC BY-NC 4.0, allowing use of the content for non-commercial purposes with appropriate attribution. Our use of these datasets is consistent with their intended purposes. Furthermore, there is no explicit measure that should be taken to check for personal information or offensive content.

## H   Other experimental results and analyses

### H.1   Generalization to longer sequence

To evaluate the generalization ability of $TF_{ape}$ and $TF_{rpe}$, we trained these two types of transformers with sequences of length 128 generated from the bigram model. Details regarding the training procedure are provided in the Appendix G. We measured the accuracy of $TF_{APE}$ and $TF_{RPE}$ with respect to the second and subsequent occurrences of the output tokens. In addition, We present the memory recall to examine the generalization.

The results are shown in Tab. 3. In terms of accuracy, transformer with RPE raises its accuracy even when the input sequence length increased to 256, while transformer with

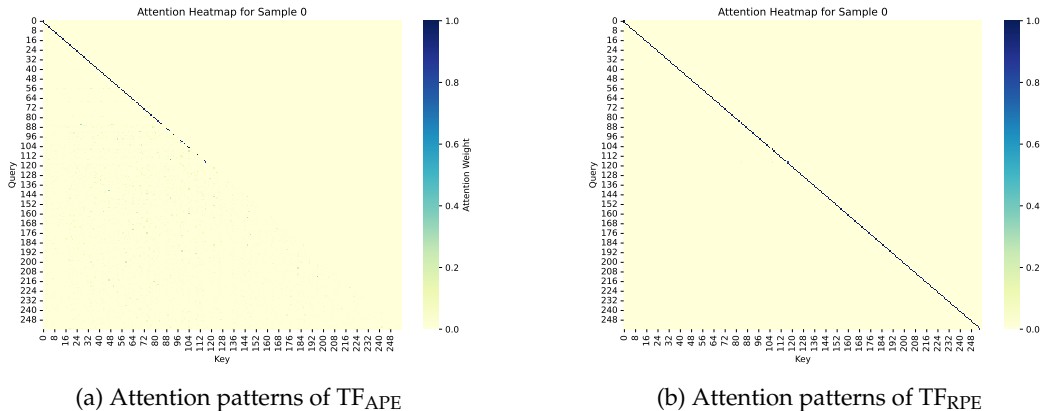

(a) Attention patterns of $\text{TF}_{\text{APE}}$             (b) Attention patterns of $\text{TF}_{\text{RPE}}$

Figure 3: Two-layer transformers with APE and RPE that are trained on sequences of length 128 show different first-layer attention patterns for length-256 sequences. The previous token head of $\text{TF}_{\text{APE}}$ do not function well for positions $t > 100$, while $\text{TF}_{\text{RPE}}$ attends to previous tokens regardless of the positions.

| | acc. ($t < 128$) | acc. ($t < 256$) | score ($t < 128$) | score ($t < 256$) |
|---|---|---|---|---|
| $\text{TF}_{\text{APE}}$ | $0.8278 \pm 0.0000$ | $0.7925 \pm 0.0000$ | $0.5460 \pm 0.0007$ | $0.1979 \pm 0.0001$ |
| $\text{TF}_{\text{RPE}}$ | $0.8727 \pm 0.0000$ | $0.8974 \pm 0.0000$ | $0.9513 \pm 0.0000$ | $0.9337 \pm 0.0001$ |

Table 3: The evaluation results under the change in sequence length from 128 during training to 256 during evaluation. Transformer with APE degrades its accuracy and attention score as the sequence length grows, while transformer with RPE shows a stable accuracy and score.

APE dropped its accuracy to 79.25%. Although the difference may seem trivial, it is worth noting that the accuracy for $t < 256$ is computed through the whole sequence including $t < 128$. This means that transformer with APE is more likely to fail to predict output tokens at positions $128 < t < 256$. Looking at the attention scores, we can confirm that $\text{TF}_{\text{APE}}$ is incapable of attending to previous tokens, even for positions $t < 128$. In contrast, $\text{TF}_{\text{RPE}}$ is trained to have a previous attention head that keeps its performance no matter how long the sequence is.

In fact, the attention pattern heatmaps in Fig. 3 in Appendix H tell us that the attention is no longer directed to the previous token after $t > 100$ for $\text{TF}_{\text{APE}}$, while $\text{TF}_{\text{RPE}}$ continues to attend to the previous token. These experimental results match our theoretical result, where RPE is better suited for length generalization. It can be seen from Fig. 3b that the first-layer attention head of $\text{TF}_{\text{RPE}}$ sometimes attends to the current token. This phenomenon is also explainable from the proof of Theorem 3, which shows that the attention matrix $W_K^1$ not only learns the associations between $w_E(v)$ and $r_{-1}$, but also the pair of $w_E(v)$ and $r_0$.

## H.2   Discussion of collision of context information

We discuss the result shown in Fig. 1b. The figure illustrates that it is more likely to output global knowledge when the number of $B_1$ and $B_2$ are the same, or when the number of $B_1$ or $B_2$ is almost maximum. Using Proposition 2, we can explain these phenomena by considering inequalities using the global knowledge $\log \pi_b(\cdot \mid A)$ and the strength of in-context knowledge $\tau_3$. Using Proposition 2, we can explain these phenomena as follows. If we have the relationships $\log \pi_b(B_* \mid A) + \tau_3 > \log \pi_b(B \mid A) > \log \pi_b(B_* \mid A) + \tau_3/2$, where $B_*$ represents $B_1$ and $B_2$, the model outputs global knowledge $B$ when $AB_1$ and $AB_2$ occur equally frequently in the prompt. On the other hand, if we have $\log \pi_b(B_2 \mid A) + \tau_3 > \log \pi_b(B \mid A) > \log \pi_b(B_1 \mid A) + \tau_3$ and $\log \pi_b(B \mid A) > \log \pi_b(B_2 \mid A)$, the prediction of the model changes to $B_2$ to $B$ as the number of $B_1$ increases.

