# OpenReview forum: "Rethinking Associative Memory Mechanism in Induction Head"
_colmweb.org/COLM/2025/Conference — COLM 2025_

### Official Review · Reviewer_Pgux · 2025-05-10

**Rating:** 8
**Confidence:** 2
**Ethics Flag:** 1

**Summary:**

This paper provides new theoretical results for induction heads on top of the associative memory viewpoint of (Bietti 2024). Specifically, it shows that for two-layer transformers, relative positional embedding (RPE) can help detect patterns in long sequences that previously have been shown to fail for absolute positional embedding (APE). Further, it shows how in-context knowledge and global knowledge (from pretraining) interact for next token prediction.

Overall, this is an interesting set of results that improve our understanding of Transformer models. The writing is clear and convincing.

**Questions To Authors:**

1. I don't fully appreciate the term "oversight of patterns". Oversight can mean either "unintentional failure" which is your meaning but also can mean "action of supervising something". Is there some other term available?
2. Definition 1, Line 120: Is there a typo? "A matrix is called..." Do you mean W?
3. Proposition 1: I don't understand the purpose of this for this paper?

**Reasons To Accept:**

1. New theoretical results for induction heads

**Reasons To Reject:**

n/a

---

> ### Author Response · Authors · 2025-06-01
> **Rebuttal by Authors**
>
> Thank you very much for your helpful feedback and support for the paper!
>
> > Q.1  Oversight can mean either "unintentional failure" which is your meaning but also can mean "action of supervising something". Is there some other term available for “oversight of patterns”?
>
> **A.1** Thank you for the helpful comment. As you correctly understand our intention to the term “oversight”, we used it to describe that the model fails to look at important information in a long context, but readers can be confused if they interpret the word as supervision of pattern. We will replace *oversight of pattern* to the following:
>
> **(before)** oversight of pattern
>
> **(after)** neglect of in-context pattern
>
> > Q.2 Definition 1, Line 120: Is there a typo? “A matrix is called…” Do you mean W?
>
> **A.2** Thank you for pointing this out. Yes, we define a matrix W as an associative memory if W satisfies the condition written in Definition 1. We will revise the sentence to make the subject explicit and avoid confusion:
>
> **(before)** *A matrix is called associative memory when W is expressed as the sum of the products of ui and vj*
>
> **(after)** *A matrix W is called an associative memory if it can be expressed as a weighted sum of outer products of u_i​ and v_j​*
>
>
> > Q.3 Proposition 1: I don't understand the purpose of this for this paper?
>
> **A.3** In the context of length generalization in transformers, models without positional encodings (NoPE) are often discussed [1, 2]. For this reason, we considered the construction of an induction head under a NoPE setting.
>
> Prior to Proposition 1, we observe that positional encodings have a significant effect on the induction head’s ability to generalize to longer sequences, as shown in the comparison between APE and RPE. However, it remained unclear whether an induction head mechanism could be implemented at all in a NoPE transformer. To address this, Proposition 1 provides a constructive proof that shows an induction head can, in principle, be realized even in a NoPE transformer.
>  However, we note that this is purely a constructive result; we did not analyze whether such a mechanism can emerge through training and whether it generalizes to a long sequence. Thus, the proposition may have been confusing.
>
> (references)
>
> [1] Kazemnejad, Amirhossein, et al. "The impact of positional encoding on length generalization in transformers." Advances in Neural Information Processing Systems 36 (2023): 24892-24928.
>
> [2] Wang, Jie, et al. "Length generalization of causal transformers without position encoding." arXiv preprint arXiv:2404.12224 (2024).

---

> > ### Comment · Reviewer_Pgux · 2025-06-05
> >
> > Thanks for the clarifications.
> >   - A.1. Yes, I think the new term is better.
> >   - A.2 ok
> >   - A.3 I see. It may be good to add some more explanation like what you have here to the paper.

---

### Official Review · Reviewer_8sjk · 2025-05-13

**Rating:** 6
**Confidence:** 3
**Ethics Flag:** 1

**Summary:**

This paper aims to understand the ability of transformers to coordinate long-context information to answer in-context learning (ICL) queries and global knowledge acquired during the pre-training stage. It focuses on two-layer transformers, which have been shown to develop induction heads for ICL, from the perspective of associative memories. The model is trained on a bigram model with some trigger tokens for next-token prediction (NTP). The paper shows, theoretically and empirically, that (a) transformers with RPE can use long-context information, whereas transformers with APE don’t work well in this setting as the attention scores to latter parts of the sequence become smaller with increasing length, and (b) how transformers prioritize in-context knowledge over global knowledge by analyzing the weight matrices and resulting logits for specific prompts.

**Questions To Authors:**

I like Table 1 which summaries and compares related work on associative memories. Can the authors also include Nichani et al. 2024 and Friedman et al. 2023 they discussed in Section 2 in the table?  The sentence in lines 59-60 is incomplete.

There’s a typo in the equation below line 97.

**Reasons To Accept:**

The paper studies interesting questions of (a) effect of RPE vs APE in length generalization for ICL in transformers and (b) how global and in-context knowledge influences the output of transformer.

The theoretical and empirical results presented in the paper are interesting and add to understanding ICL in transformers.

**Reasons To Reject:**

The paper only considers a simple data model and one real dataset; it can be strengthened by including some (more) experiments on real data with larger models to support the key findings, and investigate how general they are.

The paper writing and organization should be improved. Currently, the title and abstract don’t clearly reflect the main message of the paper. It seems to have two different focus points - length generalization and global vs in-context knowledge, but they seem to be treated as two separate things and are not connected well together. Due to this, the paper doesn’t seem to have a unified focused narrative.

---

> ### Author Response · Authors · 2025-06-01
> **Rebuttal by Authors**
>
> Thank you for the helpful feedback on our work. We write [FIX] for the revised version of our paper in the thread.
> > Q.1  Can the authors also include Nichani et al. 2024 and Friedman et al. 2023 they discussed in Section 2 in the table?
>
> **A.1** Thank you for pointing this out. We will revise the table to include additional rows for Nichani et al. (2024) and Friedman et al. (2023): [FIX1].
>
> > Q.2 The sentence in lines 59-60 is incomplete.
>
> **A.2** We will fix the grammatical errors and replace the sentence. Thank you!
>
> *Modern Hopfield networks (Krotov & Hopfield, 2016) are associative memory models that store patterns in an energy function and retrieve them via an updating rule.*
>
> > Q.3 There’s a typo in the equation below line 97.
>
> **A.2** Thank you for pointing that out! The second term in Equation (1) is incorrect. This term corresponds to the residual connection, and it should be x_t^{(0,v)}​. We will fix this in the revised version.
>
> > R.1 The paper only considers a simple data model and one real dataset; it can be strengthened by including some (more) experiments on real data with larger models to support the key findings, and investigate how general they are.
>
> **C.1** We appreciate the suggestion to expand our evaluation to larger models and additional real-world datasets. Our goal in this work is to strike a careful balance between theory, experiment, and interpretability. In practice, pursuing generality often comes at the cost of transparency and analytical tractability. We believe that overly complex settings can obscure the underlying mechanisms we aim to understand. Our current setup with a simplified data model and a two-layer transforme is deliberately chosen to isolate and study the induction head mechanism in depth. This follows a similar spirit to prior work that employs even simpler architectures, such as linear transformers or other minimal designs, to reveal core principles.
>
> We view our work as a first step toward understanding how positional encodings affect the formation and function of induction heads. We hope that the theoretical insights and targeted experiments provided here will guide future investigations into more complex settings.
>
> > R.2 Currently, the title and abstract don’t clearly reflect the main message of the paper. It seems to have two different focus points - length generalization and global vs in-context knowledge, but they seem to be treated as two separate things and are not connected well together.
>
> **C.2** We believe the current title does reflect the main message of the paper. Both length generalization and the balance between global and in-context knowledge are analyzed through the unified lens of associative memory and the induction head mechanism. The use of "Rethinking" in the title is meant to highlight our novel findings that go beyond earlier work—particularly regarding the impact of relative positional encoding (RPE), which was not previously examined, and the interplay between global and contextual knowledge in token prediction.
>
> That said, we acknowledge that the current abstract may not clearly convey this framing. In particular, the role of associative memory, a central conceptual component of our analysis, is not sufficiently emphasized. Additionally, terms used in the abstract such as  "in-context information over long context" and "how a transformer thoroughly captures in-context information" may obscure the relevance to length generalization, especially for readers who are scanning quickly.
> To address this, we propose the following revised version of the abstract, which makes the connection to associative memory and length generalization more explicit: [FIX2]

---

> > ### Author Response · Authors · 2025-06-01
> > **(NOT REQUIRED TO READ) FIXES**
> >
> > > [FIX1]
> >
> > | Paper | Model | Objective | Theoretical Analysis |
> > |-------|-------|-----------|----------------------|
> > | Nichani et al. (2024) | two-layer Transformer | causal structure | Yes |
> > | Friedman et al. (2023) | modified Transformer | transformer programs| No |
> >
> >
> > > [FIX2]
> >
> > **(before)**  *Induction head mechanism is a part of the computational circuits for in-context learning (ICL) that enable large language models (LLMs) to adapt to new tasks without fine-tuning. Most existing work explains the training dynamics behind acquiring such a powerful mechanism. However, the model’s ability to coordinate in-context information over long contexts and global knowledge acquired during pretraining remains poorly understood. This paper investigates how a two-layer transformer thoroughly captures in-context information and balances it with pretrained bigram knowledge in next token prediction, from the viewpoint of associative memory. We theoretically analyze the representation of weight matrices in attention layers and the resulting logits when a transformer is given prompts generated by a bigram model. In the experiments, we design specific prompts to evaluate whether the outputs of the trained transformer align with the theoretical results.*
> >
> > **(after)**  *Induction head mechanism is a part of the computational circuits for in-context learning (ICL) that enable large language models (LLMs) to adapt to new tasks without fine-tuning. Most existing work explains the training dynamics behind acquiring such a powerful mechanism. However, it is unclear how a transformer extract information from long contexts and then use it to coordinate with global knowledge acquired during pretraninig.  This paper considers weight matrices as associative memory to investigate how an induction head functions over long contexts  and balances in-context and global bigram knowledge in next token prediction. We theoretically analyze the representation of the learned associative memory in attention layers and the resulting logits when a transformer is given prompts generated by a bigram model. In the experiments, we design specific prompts to evaluate whether the outputs of the trained transformer align with the theoretical results.*

---

> ### Comment · Reviewer_8sjk · 2025-06-06
>
> Thank you for the response.
>
> Thanks for the clarifications and for addressing the concerns about the writing. I think the updated abstract reads better (except a typo in "how a transformer extract information from long contexts and then use it to", please fix) and clearly reflects the interesting contribution of the work. However, the response to the concern about limited empirical validation is not satisfactory, so I will maintain my score.

---

### Official Review · Reviewer_dwGn · 2025-05-13

**Rating:** 6
**Confidence:** 2
**Ethics Flag:** 1

**Summary:**

This paper presents a theoretical analysis of associative memory in two-layer transformers, extending some of what is known about how induction heads work to more than one word. A new result shows that such a transformer has attention mechanisms that effectively mirror the transition probabilities in a bigram model.  An experiment on an analgoies task provides some suggestive evidence that such a transformer can be learned from data  (though only word-pair data, since the analysis is focused on bigrams).

To be honest: I have not followed all of the induction head literature closely in the last year or so, so I am not particularly well-positioned to analyze the novel contribution of this paper.

**Questions To Authors:**

* Many papers cite only the arXiv version of works that have since been published (e.g. Friedman et al 2023, "Learning transformer programs").  I encourage the authors to check all their references for canonical published versions.

**Reasons To Accept:**

* Introduces an explicit formulation of an associative memory transformer, and links this mechanism with bigram knowledge.
* One new theoretical result on how associative memory implements bigram probabilities, together with one empirical study.

**Reasons To Reject:**

* The paper relies very heavily on Bietti et al 2024.   While it does seem independent and a new contribution, a lot of the paper is hard to follow if the reader is not intimately familiar with that earlier paper.
* Unclear whether / how the results generalize beyond bigrams, since both theoretical and empirical analyses depend on either text generated by a bigram model or word-pair tasks.  More discussion of how the simplifying assumptions in the theoretical results related to actual transformer implementations and natural language data would be helpful.

---

> ### Author Response · Authors · 2025-06-01
> **Rebuttal by Authors**
>
> We thank the reviewer for the thoughtful review. We write [FIX] for the revised version of our paper in the thread.
>
> > R.1 Unclear whether / how the results generalize beyond bigrams, since both theoretical and empirical analyses depend on either text generated by a bigram model or word-pair tasks. More discussion of how the simplifying assumptions in the theoretical results related to actual transformer implementations and natural language data would be helpful.
>
> **C.1** Even in this simplified setup, we are able to demonstrate that absolute positional encodings can negatively affect the functionality of the induction head. This finding may help explain why many real-world transformer architectures favor relative positional encodings. It is common in theoretical work to use simplified architectures or datasets, or to impose strong assumptions in order to enable formal proofs. We believe this is a necessary and accepted trade-off. Importantly, if a phenomenon cannot be understood in a small or simplified model, it is unlikely to be interpretable in larger or more complex systems. In this sense, our paper aims to offer a first step toward understanding more general transformer behaviors by carefully analyzing a tractable setting.
>
> We would also like to echo the reasoning articulated in Bietti et al. [1], who write:
>
> *“Our goal was to simplify the model as much as possible to ease the understanding of what is happening, while ensuring that the model is still rich enough to capture the relevant phenomena to solve the task. It would definitely be much more cumbersome to illustrate the memory viewpoint, and theoretically study training dynamics on a model where many more components are trained. We hope the simplicity of our architecture can help provide better intuition for what we believe to be a key internal mechanism in all transformer models.”*
>
> We share this philosophy. The simplicity of our setting is not a limitation, but a deliberate design choice to make induction heads and associative memory more interpretable, while still preserving the essential dynamics of transformer behavior.
>
> [1] https://openreview.net/forum?id=3X2EbBLNsk#:~:text=Our%20goal%20was,all%20transformer%20models.
>
> > R.1 While it does seem independent and a new contribution, a lot of the paper is hard to follow if the reader is not intimately familiar with that earlier paper.
>
> We admit that the earlier paper can enhance the understanding of our work, even though we do provide contexts and explanations of the notations and assumptions used in our paper. That said, we agree that including a brief summary of the earlier work—particularly Bietti et al.—can offer helpful context and background knowledge to readers, and make the motivation and contributions of our paper more transparent.
>
> To address this, we plan to include the following short summary paragraph about Bietti et al. in the appendix: [FIX2]
>
> > Q.1  I encourage the authors to check all their references for canonical published versions.
>
> **A.1** Thank you for the helpful suggestion. We will revise the reference section to ensure that cited papers include their canonical published versions, with the corresponding conference or journal names where applicable. The following papers are included: [FIX1].

---

> > ### Author Response · Authors · 2025-06-01
> > **(NOT REQUIRED TO READ) FIXES**
> >
> > > [FIX1]
> > - what learning algorithm is in-context learning investigations with linear models
> > - Language models are few-shot learners
> > - Scaling laws for associative memories
> > - How truncating weights improves reasoning in language models
> > - Unveiling induction heads: Provable training dynamics and feature learning in transformers
> > - Analyzing transformers in embedding space
> > - Pre-training of deep bidirectional transformers for language understanding
> > - The evolution of statistical induction heads: In-context learning markov chains
> > - Learning transformer programs
> > - Transformer feed-forward layers are key-value memories
> > - Dissecting recall of factual associations in auto-regressive language models
> > - Decoding enhanced bert with disentangled attention
> > - In-context convergence of transformers
> > - Improve transformer models with better relative position embeddings
> > - Do llms dream of elephants (when told not to)? latent concept association and associative memory in transformers
> > - Rethinking positional encoding in language pre-training
> > - Shape: Shifted absolute position embedding for transformers
> > - Dense associative memory for pattern recognition
> > - One step of gradient descent is provably the optimal in-context learner with one layer of linear self-attention
> > - Efficient estimation of word representations in vector space
> > - Progress measures for grokking via mechanistic interpretability
> > - How transformers learn causal structure with gradient descent
> > - Randomized positional encodings boost length generalization of transformers
> > - Self-attention with relative position representations
> > - The curious case of absolute position embeddings
> > - Encoding word order in complex embeddings
> > - Interpretability in the wild: a circuit for indirect object identification in gpt-2 small
> > - An explanation of in-context learning as implicit bayesian inference
> > - Trained transformers learn linear models in-context
> > - Length extrapolation of transformers: A survey from the perspective of position encoding
> >
> > > [FIX2]
> >
> > The remarkable success of LLMs has increased the need for a deeper understanding of their internal mechanisms and for enhancing their reliability. Existing studies lack detailed insights into how reasoning abilities evolve during the learning process based on contextual information. Bietti et al. (2024) [2] explored the dynamics of balancing in-context knowledge and global knowledge through a bigram model, analyzing the development of these abilities as part of the training dynamics. They conceptualized the weight matrices of transformers as associative memory and theoretically demonstrated that the induction head mechanism can be trained by proposing the storage of specific embedding pairs from training data. In their experiments, they trained a two-layer transformer on a bigram model estimated from the tiny Shakespeare dataset. The proximity of each weight matrix to the theoretically derived weights was measured using a memory recall probe. They analyzed how these weights changed over the course of training. This work is the most relevant to our research. We note, however, that while they primarily focus on the learning dynamics of induction heads in a two-layer transformer, our primary emphasis lies on length generalization and the reasoning process.
> >
> > (references)
> >
> > [2] Bietti, Alberto, et al. "Birth of a transformer: A memory viewpoint." Advances in Neural Information Processing Systems 36 (2023): 1560-1588.

---

> > > ### Author Response · Authors · 2025-06-07
> > > **Official Comment by Authors**
> > >
> > > We believe we have addressed all of your questions and would be grateful for any additional feedback you may have at your convenience.

---

> > ### Comment · Reviewer_dwGn · 2025-06-09
> >
> > I thank the authors for their detailed reply.  To be clear: I fully understand the value of small and simplified models, and the ability to provide complete understanding (e.g. via formal proof) in a way that cannot be done at scale.  My question about bigrams was not about simplified models in general, but about one particular assumption.  I would still like to hear more discussion about the nature of that assumption and what would be needed to generalize beyond it.  That being said, I am happy to move my score to a 6 in light of this response and the other reviews.

---

### Official Review · Reviewer_SFDP · 2025-05-13

**Rating:** 7
**Confidence:** 3
**Ethics Flag:** 1

**Summary:**

The paper provides a theoretical analysis for understanding in-context learning in transformer LMs, specifically focusing on the associative memory and the induction heads. The authors first prove that a simplified 2-layer transformer, using Relative Position Encodings (RPE) can learn the previous token head without encountering the oversight issue that affects Absolute Position Encodings (APE). The authors also provide a construction for a 2 layer transformer that learns the bigram distribution of the training data through the feed-forward memories and the induction head behavior through the self attention layer. Through their construction, the authors show that the next token prediction is dictated by the bigram distribution as well as the frequencies of possible continuations that follow the current token in the sequence. The authors also provide toy experiments to provide evidence for the theoretical result, where models trained with APE fail to learn the desired associations with previous tokens as the sequence length increases but RPE models do. For global vs. in-context associative memories, the authors provide experiments on the Google Analogy Dataset modified to contain trigger tokens for the in-context associative memories, and find trained models show some evidence supporting their construction.

Overall, the theoretical results of the paper are interesting and build upon the current theoretical understanding of transformer models. The experiments also provide some support for the theoretical findings, but do lack a more thorough exploration, which might be needed to show how much the theoretical constructions agree with what models learn in practice.

**Questions To Authors:**

- What form do you consider for the absolute position encodings $p_t$? Are these learned d-dimensional embeddings with Gaussian entries? If not, do they depend on the value of t?
- In Remark 1, in this line “if the input sequence contains out-of-distribution tokens, the induction head mechanism does not activate”, how are out-of-distribution tokens defined?
- The notation in Lemma 1 is a bit confusing. The $Q$ in $W_Q$ and the $Q$ which is the support of $\pi_Q$ are not the same right?
- In equation 8, what does the prime symbol over the logit symbols denote?
- In the same equation, shouldn’t the difference be between the logarithm of the logits to be equal to the value in the right-hand side?
- I would suggest the authors define induction heads at least informally before talking about them in the introduction.

**Reasons To Accept:**

- The result about transformers with RPE being robust to oversight as compared to APE is very interesting and also a potentially important result in the theoretical understanding of transformers.

- While the constructions are provided for a simplified version of transformers with 2 layers, I appreciate the authors include feed-forward layer in their analysis as well, since those can be harder to construct given the non-linearity. The constructions very clearly show the interplay between the global and in-context associative memories, whereas the constructions in the prior work focused on one or the other but not both.

**Reasons To Reject:**

- While through the experiments the authors show that APE models do not learn the desired associations at later stages of the sequence by calculating the memory recall metric, it would have been interesting to see whether this impacts the performance of models in correctly predicting the next token. In lines 303-310, the authors provide a discussion of how this can influence model prediction but, to my understanding, do not provide any experimental evidence to support it.

- I would have liked stronger empirical evidence for the model's learning behavior consistent with the proposed construction of Definition 2.

---

> ### Author Response · Authors · 2025-06-01
> **Rebuttal by Authors**
>
> Thank you very much for your helpful feedback and support for the paper!
> > Q.1 What form do you consider for the absolute position encodings p_t? Are these learned d-dimensional embeddings with Gaussian entries?  If not, do they depend on the value of t?
>
> **A.1**  No, they are not learned. The position encodings p_t​ are fixed d-dimensional embeddings with Gaussian entries, so they do not depend on the value of t. This setting is what Bietti et al., (2024) used for their analysis. In our experiments, we use fixed embeddings to ensure near-orthogonality, which is necessary for the memory recall metric to function properly.
>
> > Q.2 in this line “if the input sequence contains out-of-distribution tokens, the induction head mechanism does not activate”, how are out-of-distribution tokens defined?
>
> **A.2** Out-of-distribution tokens are defined as tokens in the vocabulary that never appear in any of the training sequences. Since these tokens are never observed during training, the model does not learn their relationship with the relative position embedding r_{-1}. In other words, the associative memory in the first attention layer does not store any information about the pair consisting of such a token and the previous position embedding.
>
> > Q.3 In lemma 1, Q in W_Q and Q in \pi_Q are not the same right?
>
> **A.3** Yes, they are different. W_Q refers to the query matrix in the transformer, which is used to compute query vectors in the attention mechanism. On the other hand, \pi_Q​ refers to the bigram conditional distribution used to generate trigger tokens in our data generation process. While we used the notation \pi_Q​ to be consistent with Bietti et al. (2024), we acknowledge that this may cause confusion due to the overload of the symbol "Q." We plan to revise the notation in the revised version to avoid ambiguity.
>
> > Q.4 In equation 8, what does the prime symbol over the logit symbols denote?
>
> **A.4** The prime symbol denotes the logits before applying the softmax function, as stated in line 249. More specifically, Equation (4) defines the logits  \xi = \sigma(W_U x_t), where \sigma is the softmax function. We define  \xi’ as W_U x_t​, i.e., the raw logits before softmax is applied. By comparing the entries of this vector, we can analyze how the logit difference between two vocabulary items (e.g., v_1​ and v_2 in Proposition 2​) arises.
>
> > Q.5 In equation 8,  shouldn’t the difference be between the logarithm of the logits to be equal to the value in the right-hand side?
>
> **A.5** Yes, the value on the right-hand side should indeed correspond to the difference between the logarithms of the softmax probabilities, which in turn equals the difference between the unnormalized logits before softmax. This is what is explained in Q.4 and A.4.
>
> > Q.6 suggest the authors define induction heads at least informally before talking about them in the introduction.
>
> **A.6** Thank you for the suggestion. We agree that introducing the concept of induction heads earlier in the paper would improve clarity for readers unfamiliar with the term. We will revise the introduction to include a brief explanation of induction heads:
>
> **(before)** *It is known that the induction head mechanism can emerge in two-layer transformers
> 26 (Elhage et al., 2021), and many theoretical studies have focused on analyzing two-layer
> 27 architectures to understand how induction head develops (Edelman et al., 2024; Nichani
> 28 et al., 2024; Chen et al., 2024b)*
>
> **(after)** *It is known that the induction head mechanism, which is a pattern of attention in a transformer that enables the model to copy and reuse information from earlier in the context, can emerge in two-layer transformers
> (Elhage et al., 2021), and many theoretical studies have focused on analyzing two-layer
> architectures to understand how induction head develops (Edelman et al., 2024; Nichani et al., 2024; Chen et al., 2024b)*

---

> > ### Comment · Reviewer_SFDP · 2025-06-03
> > **Acknowledgement**
> >
> > Thank you for answering my questions and providing clarifications. Look forward to the final paper version.

---

### Official Review · Reviewer_1jjv · 2025-05-24

**Rating:** 4
**Confidence:** 4
**Ethics Flag:** 1

**Summary:**

This paper investigates in-context learning from the point of view of associative memory, where the core experiment analyzes how a model balances global knowledge vs knowledge given in context. The paper shows that using relative positional encoding (RPE) in two-layer transformers lets induction heads form more stably avoiding the recall decay seen with absolute positional encoding (APE). The authors motivate this theoretically and confirm empirically (on synthetic bigram data and analogy tasks) that RPE models outperform APE models especially at long context. These results clarify how positional encoding underpins reliable in-context learning.

**Questions To Authors:**

- **Bigram Model**: I am a little bit confused about this bigram model’s description. For this model to represent in context learning, shouldn’t the trigger token-output token pairs mostly be specified by the context? It seems like the trigger tokens are fixed for the whole data.
- **Latter half**: I am quite confused about the mention of the latter half near line 165 and the comparison between sequence position <128 and >=128. Is there a sharp transition at half the sequence length? It seems like the transition is smooth. If there is a sharp transition at half and this is predicted by theory, I might update my score significantly as this would have real generalizable implications!
- **Comparison with NOPE**: While I criticized the comparison of APE vs RPE since APE is seldom used, I think a comparison with no positional encoding (2) might be useful since then one is really comparing whether randomly initialized positional encoding is *worse* than a hard to represent but noise-free encoding.

[2] https://arxiv.org/abs/2404.12224

**Reasons To Accept:**

- **Timely Topic**: This paper investigates a timely topic of global pretrained statistics vs in-context information.
- **Proof of RPE stability**: They provide a p roof showing that RPE forms a more stable induction head irrespective of token position
- **Simple synthetic data**: The paper uses very simple synthetic data to validate their findings, which helps the interpretation of the results
- **Explanation of undertrained token effects**: This paper addresses the recent curiosities of undertrained tokens in LLMs [1].
- **Discussion of limitations**: They acknowledge the limitations: analysis constrained to two-layer transformer and induction heads.

[1] https://arxiv.org/abs/2405.05417

**Reasons To Reject:**

- **Heavily relying on theory of simple architectures**: Theorems proven on a two-layer transformer may not generalize to larger or more realistic models. While the small-scale experiments are helpful, it’s unclear how broadly the proofs apply.
- **Lack of surprisingness**: It isn’t particularly surprising that RPE yields more stable induction heads—RPE explicitly encodes relative distances, which appear as a specific direction in the token representations. It isn’t that surprising that induction heads can naturally pick up this direction.
- **Setting description could be improved**: The paper could more clearly specify exactly how the synthetic training data are constructed and sampled.
- **Presentation of results**: It’s hard to gauge the real impact of the experiments without explicit accuracy-vs-context-length curves. Showing only two curves for first half and second half doesn’t show how the performance decays. Figure 1a in its current form is difficult to interpret, what the main findings are.
- **Practical relevance**: It is unclear how relevant the study of positional encoding’s effect on ICL is when in practice, APE is almost never used compared to RPE.
- **Lack of some critical references**: In general, the paper cites related works throughly, but significant work on global vs in-context correlations and in-context learning’s relation to the training distribution is missing: https://arxiv.org/abs/2312.03002, https://arxiv.org/abs/2401.12973, https://arxiv.org/abs/2412.01003, https://arxiv.org/abs/2306.15063, https://arxiv.org/abs/2404.07129

---

> ### Author Response · Authors · 2025-06-01
> **Rebuttal by Authors**
>
> We appreciate the reviewer’s thorough review. The [FIX]s are for the revised version of our paper and displayed in another comment.
>
> > Q.1  shouldn’t the trigger token-output token pairs mostly be specified by the context?
>
> **A.1** Yes, we sample the trigger token and the output token for each sequence individually. While lines 142–144 mention how these tokens are sampled, what we mean is that the trigger token is sampled from the conditional distribution \pi_q​ for each sequence. Thus, as you correctly point out, the trigger token–output token pairs should indeed be specified by the context. We agree that lines 142–144 may be unclear, so we propose revising them: [FIX1]. Also, we add the following section in the Appendix: [FIX2].
>
>
> > Q.2  quite confused about the mention of the latter half near line 165 and the comparison between sequence position <128 and >=128. Is there a sharp transition at half the sequence length?
>
> **A.2** The transition does not occur exactly at position 128. In Fig. 3 of Appendix H, we visualize the attention pattern in the first layer of a trained transformer. The figure illustrates that the model loses its ability to attend to the previous token around the midpoint of the sequence. This observation motivated our use of the term "latter half" to describe the general region where this degradation becomes apparent. Thus, in the experiment shown in Fig. 1(a), we divide the sequence into two halves at position 128 and measure the memory recall associated with the previous token in each half.
>
>
> > Q.3 Comparison with NOPE: I think a comparison with no positional encoding (2) might be useful.
>
> **A.3** We conducted an experiment to see whether 3-layer transformer with no positional encoding learns previous token head. We examined the 2nd layer attention and the result is in the figure [1]. It attends to previous token for the first appearance of trigger-output tokens but does not learn to always attend to previous token.
>
> [1] https://www.anonfile.la/ad77c6
>
>
> > R.1 Heavily relying on theory of simple architectures: Theorems proven on a two-layer transformer may not generalize to larger or more realistic models.
>
> **C.1** In our opinion, using a two-layer transformer allowed us to analyze the induction head mechanism with greater precision and clarity. We believe that analyzing such small, interpretable models is a necessary step toward understanding the behavior of larger, more complex ones. In fact, we were able to provide a tractable yet meaningful explanation of how relative positional encoding facilitates the implementation of induction heads. This, in turn, offers a theoretical justification for the widespread use of RPE in practice.
>
> > R.2 Lack of surprisingness: It isn’t particularly surprising that RPE yields more stable induction heads
>
> **C.2** We believe the surprising aspect lies in the fact that these behaviors can be theoretically explained in terms of associative memory, while we agree that some of the observed phenomena, such as the superiority of RPE over APE, may be expected or well-known empirically. The core contribution of our work is to formally prove that RPE is better suited for implementing induction heads. We believe this shift from empirical intuition to theoretical understanding is highly meaningful for the community, as it not only justifies a common design choice but also deepens our insight into how transformers operate.
>
> > R.3 Setting description could be improved: The paper could more clearly specify exactly how the synthetic training data are constructed and sampled.
>
> **C.3**  we addressed this concern in A.1. We fully agree that improving readability is important to help readers follow the logic more easily.
>
> > R.4 Presentation of results: Figure 1a in its current form is difficult to interpret, what the main findings are.
>
> **C.4** The main finding illustrated in Fig.1a is that our theoretical insights about the behavior of RPE and APE are empirically validated. This figure plots how the memory recall metric, defined in lines 297–298, evolves throughout training. We will add the paragraph as a supplementary explanation: [FIX3].
>
> > R.5 Practical relevance: It is unclear how relevant the study of positional encoding’s effect on ICL is when in practice, APE is almost never used compared to RPE.
>
> **C.5** Our study helps explain why APE is rarely used in practice and RPE is the default choice in many real-world applications. Our work addresses this gap by providing a theoretical explanation for why APE performs poorly in long-context settings. We believe this insight is practically relevant because it offers interpretability for a widely observed empirical trend. Shedding light on why certain design choices (like RPE) lead to better in-context learning not only justifies current best practices. In that sense, our work contributes to the broader goal of increasing transparency and understanding of in-context learning, a core capability of modern language models.

---

> > ### Author Response · Authors · 2025-06-01
> > **(NOT REQUIRED TO READ) FIXES**
> >
> > Reviewers are not required to read these fixes. However, these change will be included for the revised version of our paper.
> >
> > > [FIX1]
> >
> > **(before)** In this bigram language model over a vocabulary of size V, we first prepare trigger tokens q_k​ sampled from a distribution \pi_q​ and output tokens o_k​ sampled from πo\pi_oπo​ that always follow their corresponding trigger token in every bigram sequence.
> >
> > **(after)** In this bigram language model over a vocabulary of size V, we sample trigger tokens q_k from a distribution \pi_q​ and output tokens o_k​ from \pi_o​ each time we generate a training sequence. Output tokens always follow their corresponding trigger tokens in the generated sequences.
> > > [FIX2]
> >
> > \section{Example of Bigram Sequence Generation with Trigger-Output Pairs}
> >
> > To illustrate the sequence generation procedure in our modified bigram language model, we provide a concrete example with a small vocabulary V = {A, B, C, D}. The generation proceeds in discrete steps as follows:
> >
> > **Step 0: Sample trigger-output pair**
> >
> > A new trigger-output pair (q_k, o_k) is sampled from π_q and π_o. For example, suppose (q_1, o_1) = (A, C).
> >
> > **Step 1: Sample the first token**
> >
> > The first token z_1 is sampled from a predefined distribution over V, e.g., uniformly. Suppose z_1 = B.
> >
> > **Step 2: Trigger check and transition**
> >
> > - If z_1 = q_1, then set z_2 := o_1.
> > - Otherwise, sample z_2 ~ π_b(· | z_1).
> >
> > Since z_1 ≠ q_1, we sample z_2 = A.
> >
> > **Step 3: Trigger condition applies**
> >
> > Since z_2 = q_1 = A, we set z_3 := o_1 = C.
> >
> > **Step 4: Normal bigram transition**
> >
> > Since z_3 ≠ q_1, sample z_4 ~ π_b(· | z_3) and suppose z_4 = D.
> > These steps result in the sequence BACD.
> >
> > **Step 5: Re-sample trigger-output pair**
> >
> > Return to Step 0 and sample a new (q_k, o_k) from π_q and π_o to generate a new sequence.
> >
> > > [FIX3]
> >
> > For example, in the RPE setting, the metric quantifies how strongly each vocabulary token is associated with the relative position embedding r_{-1}​ in the learned associative memory. As training progresses, the recall approaches 1.0, indicating that nearly all vocabulary items are successfully paired with r_{-1}​, confirming the effectiveness of RPE for associative memory formation. In contrast, for APE, we observe that memory recall is substantially higher for positions t < 128 compared to the full positions. This supports our theoretical claim that it is harder to embed the relationship between p_t​ and p_{t-1} ​ into associative memory when t is large, due to the diminishing alignment between position embeddings at distant positions.

---

> > ### Comment · Reviewer_1jjv · 2025-06-06
> >
> > Re: A.2
> > While the attention pattern might look like the performance is abruptly dropping, it is unclear if this actually justifies whether the behavior of the neural network could be segmented into before half and latter half. This especially is the case since transformers can easily compute residuals of two attention patterns, and it has been shown that high magnitude doesn't always means behavioral relevance. Attention sink papers or https://arxiv.org/abs/2404.12224
> >
> > Re: A.3
> > I cannot see the image. is the conclusion the APE actually harms induction head formation?
> >
> > Re: C
> > My concern is that the theory doesn't seem to enhance the insight on how we understand the role of positional encoding in transformers. Is there a prediction of the theory or falsifiable suggested experiment?
> >
> > ---
> >
> > Overall, my judgement is that the current score is adequate unless significant potential practical impact can be demonstrated.

---

> > > ### Author Response · Authors · 2025-06-07
> > > **Official Comment by Authors**
> > >
> > > Thank you for the feedback!
> > >
> > > > While the attention pattern might look like the performance is abruptly dropping, it is unclear if this actually justifies whether the behavior of the neural network could be segmented into before half and latter half.
> > >
> > > Yes, as we previously noted, this behavioral transition happens to appear around the halfway point of the sequence. However, we would like to clarify that our argument is not based solely on the attention pattern.We also use the memory recall metric to assess the function of associative memory, i.e., the model behavior.
> > >
> > > Specifically, in Fig. 1(a), the memory recall measures the proportion of positions that the model attends to their previous position. The APE wk1_acc in the figure is the result of the entire sequence positions (1<t<256). Thus, if we look solely at the before and latter half, it is true that we see clearly different trends.
> > >
> > > > it has been shown that high magnitude doesn't always mean behavioral relevance.
> > >
> > > In our case, high magnitude does relate to behavioral relevance, because the theory explains how the induction head is implemented with such high magnitude, and the actual learned matrices in attention layers align with the theory.
> > > We also conducted an additional experiment, where we generated 10,000 sequences that do not have trigger-output token pair in t < 64 (We set 64 to make sequences having such a constraint appear more easily). Then, the trained transformers with APE and RPE showed clear performance differences as in the table below. This strengthens our claim that high magnitude of attention pattern is related to model behavior, I.e., RPE is superior to APE with regard to accuracy.
> > >
> > > | Metric           | APE    | RPE      |
> > > |------------------|-------------------|-------------------|
> > > | Loss (Mean)      | 2.7441            | 0.4131            |
> > > | Acc (Mean)       | 0.3710            | 0.8027            |
> > > | Acc (Variance)   | 0.0022            | 0.0004            |
> > >
> > > > I cannot see the image.
> > >
> > > Sorry for the inconvenience, we make it visible via the following link.
> > >
> > > https://drive.google.com/file/d/17o_yU2_9AxH581O2B7GyVd8Zxi30GhaF/view?usp=drivesdk
> > >
> > > The figure illustrates the training process of transformers with NoPE using the same training setup as RPE and APE. We see that as the training progresses, the loss decreases and the attention layer directs to previous tokens for positions where  trigger-output pairs first appear. However, as a whole, the attention layer did not learn a previous token head like RPE does, and resembles APE.
> > >
> > > > is the conclusion the APE actually harms induction head formation?
> > >
> > > No, the figure only shows an induction head is learned even without positional encoding. But the above table shows APE harms the performance.
> > >
> > >
> > > > My concern is that the theory doesn't seem to enhance the insight on how we understand the role of positional encoding in transformers.
> > >
> > > First, we would like to point out that the experimental result of NoPE that implies 3-layer NoPE transformers learn an induction head is a new insight that is derived from Proposition.1. We appreciate that the discussion leads to a larger contribution of our work.
> > >
> > > Second, we believe that “knowing” a phenomenon leaves a large gap until one can “explain” its underlying mechanism. Our contribution is to bridge this gap by providing that mechanistic explanation.
> > > There has been a large amount of existing work that examines the behavioral aspects of transformers. This gives us some knowledge about how transformers process data in the real world, but not much effort is dedicated to illuminate the “why” part. We did not know why NoPE transformers can be competitive with transformers with explicit positional encodings, and did not know why transformers with RPE is believed to be better suited for length generalization. This paper does contribute to the analysis of such phenomena.

---

### Decision · Program_Chairs · 2025-07-08

**Decision:**

Accept

**Comment:**

The paper provides a theoretical analysis of how a Transformer would be able to represent associative memory, and how it would balance the two goals of using in-context information vs. using in-weights knowledge. The paper also provides insights into the effect of RPE vs APE for length generalization.

The reviewers acknowledge that the topic considered here is timely; the contributions are interesting and novel.

The major concern that stood out to me was the potential gap to practical relevance (generalizing to multiple layers, beyond bigrams etc.,) and some lack of surprise (of one of the two main theoretical results) raised by one reviewer. However, I believe that the results in the paper are novel and insightful enough in light of what exists in literature --- hopefully future iterations on top of this work can extend these analyses to more interesting settings.

I would argue it makes sense (and even desirable) for the theory to be in a simplified setting without unnecessary complications that make the proofs lengthy with no new insight; however the paper would be strengthened with more experiments verifying their insights on real-world data as suggested by multiple reviews (`SFDP `, `1jjv`, `8sjk`).  I hope the authors will incorporate the feedback about:
- stronger experiments and
- improving the clarity in the description of the setup, and recapping results from Bietti et al 2024.
- any missing citations
for future versions of the paper.

[Automatically added comment] At least one review was discounted during the decision process due to quality]